# PROVABLE DERIVATIVE-FREE INFERENCE WITH SCORE-BASED GENERATIVE PRIORS

## ABSTRACT

A growing trend in solving inverse problems is to use pre-trained score-based generative models (SGMs) as plug-and-play priors. This paradigm retains the generative power of SGMs while allowing adaptation to different forward models without requiring re-training. In parallel, derivative-free posterior sampling algorithms have gained increasing attention for solving inverse problems where the derivative, pseudo-inverse, or full knowledge of the forward model is unavailable or impractical to compute. Despite their success, these methods lack principled foundations and provide no convergence guarantees to the true posterior distribution or to its $\varepsilon$-accurate approximation. We propose *zeroth-order annealed plug-and-play Monte Carlo (ZO-APMC)*, the first principled derivative-free framework for solving general inverse problems that requires only forward-model evaluations and a pre-trained SGM prior. We derive complexity bounds for obtaining samples with $\varepsilon$-relative Fisher information under a non-log-concave likelihood distribution and, under a Poincaré inequality assumption, $\varepsilon$-accuracy in total variation distance, and we establish weak convergence of ZO-APMC to the target posterior. We verify our theory with numerical experiments and demonstrate its performance on both linear and nonlinear inverse problems.

## 1 INTRODUCTION

The use of pre-trained score-based generative models (SGMs) (Song et al., 2020b; Ho et al., 2020) as *plug-ang-play* priors for tackling inverse problems has become increasingly prominent, showing strong effectiveness across diverse domains such as image restoration (Wang et al., 2022; Rout et al., 2023), medical imaging (Song et al., 2021; Sun et al., 2024), and image and music generation (Rout et al., 2024). A primary advantage of this framework is its flexibility. It can be applied to various inverse problems without re-training, while preserving the expressive capacity of SGMs to capture complex, high-dimensional priors. However, there are many practical and scientific applications where the privileged knowledge of the forward model such as its derivative (Song et al., 2023b; Chung et al., 2022), pseudo-inverse (Song et al., 2023a), or its parametrization (Chung et al., 2023) is unavailable or computationally prohibitive. This occurs in a wide range of settings: the forward operator may be defined through large PDE-based simulators whose derivatives or pseudo-inverses are typically inaccessible or undefined (Oliver et al., 2008; Iglesias et al., 2013; Evensen & Van Leeuwen, 1996); simulators may rely on legacy code that cannot be adapted to modern auto-differentiation frameworks (Harbaugh et al., 2000); the forward model may be a proprietary system (closed-source), as in commercial MRI systems (Karakuzu et al., 2025; Cashmore et al., 2021); the underlying physics may involve discontinuities (Moës et al., 1999; Tan et al., 2018; Lopez-Gomez et al., 2022); or the forward model may be implemented as a rule-based expert system (Huang et al., 2024; Rotshtein & Rakytyanska, 2012; Gong et al., 2025).

To address such black-box inverse problems, derivative-free posterior sampling methods with SGM priors have recently gained significant attention (Tang et al., 2024; Huang et al., 2024; Zheng et al., 2025a). Despite their empirical success in reconstructing images, they lack theoretical convergence guarantees to the target posterior distribution or to $\varepsilon$-accurate approximation. In fact, even among posterior sampling methods with gradient access of a forward-model, rigorous guarantees are rare; when provided, they typically assume a linear forward operator, which is an assumption often violated in practice (Daras et al., 2024).

The goal of this work is to develop a theoretically grounded method for solving inverse problems that uses only black-box access to the forward model together with a pre-trained SGM prior. We position this as an important step toward posterior sampling in black-box settings that offers an algorithm with formal convergence guarantees and a solid foundation for future advances. A key challenge in this direction is that, although existing posterior sampling solvers employ principled formulations, they often rely on heuristic, intuition-driven approximations of the forward model's score function (Iglesias et al., 2013; Huang et al., 2024; Tang et al., 2024), which makes rigorous convergence analysis difficult.

To tackle this issue, we develop a *zeroth-order (ZO) Markov chain Monte Carlo (MCMC)* posterior sampling method with Langevin dynamics (Iglesias et al., 2013; Huang et al., 2024; Tang et al., 2024), in which the forward-model score is approximated from noisy function evaluations. In developing this approach, we face two key challenges. First, Langevin methods are known to exhibit mode collapse and slow convergence when sampling high-dimensional multimodal distributions; motivated by annealed importance sampling (Neal, 2001; Sun et al., 2024), we incorporate a weighted annealing scheduler to enhance exploration. Second, conventional ZO methods become computationally expensive in high dimensions. The curse of dimensionality leads to high-variance ZO gradient estimates, which then requires large batch sizes per iteration to obtain an accurate forward-model score. As a result, derivative-free posterior sampling with such estimators becomes impractical. To make our approach practical for high-resolution image reconstruction, we adopt a PAGE-inspired variance-reduction strategy from the optimization literature (Li et al., 2021), which reduces estimator variance and maintains a *fixed per-iteration cost* without sacrificing accuracy. Our main contributions are the following:

- We propose zeroth-order annealed plug-and-play Monte Carlo (ZO-APMC), a derivative-free method with a variance-reduction mechanism that leverages a pre-trained SGM prior for general inverse problems. ZO-APMC operates using only black-box access to the forward model and applies to different forward operators without re-training.

- For general non-log-concave likelihood distributions, we establish that the averaged ZO-APMC algorithm converges to the target posterior in distribution under decaying hyperparameters. We further provide non-asymptotic convergence guarantees showing that it attains $\varepsilon$-relative Fisher information (FI) after $O(1/\varepsilon^4)$ iterations with arbitrary per-iteration cost, and, assuming the target distribution satisfies the Poincaré inequality, $\varepsilon$-accuracy after $O(1/\varepsilon^4)$ iterations in squared total variation (TV) distance as well.

- We substantiate our theoretical findings through comprehensive numerical and statistical evaluations, and further demonstrate that our method outperforms all other derivative-free baselines in MRI reconstruction and black hole imaging, and delivers competitive performance on the Navier–Stokes problem.

## 2 BACKGROUND

**Problem setting.** We consider a general inverse problem modeled as

$$\boldsymbol{y} = \boldsymbol{A}(\boldsymbol{x}) + \xi, \quad \boldsymbol{x} \in \mathbb{R}^d, \quad \boldsymbol{y}, \xi \in \mathbb{R}^m, \tag{1}$$

where we only have black-box access to $\boldsymbol{A}(\cdot)$. The goal is to recover the unknown signal $\boldsymbol{x}$ from noisy measurements $\boldsymbol{y}$. The forward operator $\boldsymbol{A} : \mathbb{R}^d \to \mathbb{R}^m$ characterizes the response of the imaging system, while $\xi \in \mathbb{R}^m$ denotes the measurement noise, typically modeled as Gaussian or Laplacian distribution. In many practical settings, the mapping $\boldsymbol{x} \to \boldsymbol{y}$ is many-to-one, making the reconstruction task an ill-posed inverse problem, since $\boldsymbol{x}$ cannot be uniquely recovered from $\boldsymbol{y}$. In Bayesian framework, one could introduce $p(\boldsymbol{x})$ as the *prior*, and samples from the *posterior* $\pi(\boldsymbol{x}|\boldsymbol{y})$, which is formally established with the Bayes' rule: $\pi(\boldsymbol{x}|\boldsymbol{y}) \propto \ell(\boldsymbol{y}|\boldsymbol{x})p(\boldsymbol{x})$, where $\ell(\boldsymbol{y}|\boldsymbol{x})$ is the *likelihood* distribution induced by (1). In this work, we adopt Langevin MCMC approach to sample from the posterior by leveraging a pre-trained SGM prior for solving inverse problems with black-box forward models.

**Score-based generative models (SGMs).** SGMs have emerged as a powerful deep learning (DL) framework for sampling from complex, high-dimensional distributions. At their core, they learn the perturbed score function $\nabla \log p_\sigma(\boldsymbol{x})$, where $p_\sigma(\boldsymbol{x}) = \int_{\mathbb{R}^d} p(\boldsymbol{z})\phi_\sigma(\boldsymbol{x} - \boldsymbol{z}) \, d\boldsymbol{z}$ and $\phi_\sigma$ is the

probability density function of $\mathcal{N}(0, \sigma^2 I)$. This score is learned using the *score matching* technique (Hyvärinen & Dayan, 2005; Vincent, 2011) and estimated via *Tweedie's* formula (Efron, 2011). The resulting score estimates are then integrated into Langevin MCMC sampler to perform iterative draws for unconditional image generation (Song & Ermon, 2020; 2019). In particular, sampling proceeds via a discretization of the Langevin diffusion process

$$d\mathbf{x}_t = \nabla \log p(\mathbf{x}_t) \, dt + \sqrt{2} \, d\boldsymbol{B}_t, \tag{2}$$

where $\{\boldsymbol{B}_t\}_{t \geq 0}$ denotes an $d$-dimensional Brownian motion and the learned score function $\mathcal{S}_\theta(\boldsymbol{x}_t, \sigma)$ approximates $\nabla \log p(\mathbf{x}_t)$ for sufficiently small $0 < \sigma$. For posterior sampling, applying Bayes' rule and substituting the prior term with its estimate provided by $\mathcal{S}_\theta(\boldsymbol{x}, \sigma)$ yields

$$d\boldsymbol{x}_t = [\nabla \log \ell(\boldsymbol{y}|\boldsymbol{x}_t) - \mathcal{S}_\theta(\boldsymbol{x}_t, \sigma)] \, dt + \sqrt{2} d\boldsymbol{B}_t. \tag{3}$$

Similarly, diffusion models (DMs), a class of SGMs, enable posterior sampling by learning the reverse diffusion process that transforms a simple distribution into $\pi(\boldsymbol{x}|\boldsymbol{y})$ (Yang et al., 2023). This requires approximating the time-dependent score $\nabla \log \ell_t(\boldsymbol{y}|\boldsymbol{x}_t)$ (Chung et al., 2022). Although these methods are empirically attractive due to their short inference time, they generally lack theoretical guarantees on their convergence to the posterior distribution. In addition, a central limitation shared by both MCMC- and DM-based approaches is their reliance on access to gradients, which is unavailable in black-box inverse problems (Knape & De Valpine, 2012; Zheng et al., 2025b).

**Derivative-free diffusion guidance for inverse problems.** Recent studies increasingly explore derivative-free strategies for guiding SGMs in inverse problems. Three DM-based approaches have been proposed to date: Stochastic Control Guidance (SCG) (Huang et al., 2024) and Diffusion Policy Gradient (DPG) (Tang et al., 2024), which cast diffusion guidance in a stochastic control framework and steer the sampling process via an estimated value function, and Ensemble Kalman Guidance (EnKG) (Zheng et al., 2025a), which uses statistical linearization to guide the diffusion process without explicit gradients. Although these methods have shown encouraging empirical results, they face a fundamental trade-off between broad applicability to highly nonlinear, black-box systems and the availability of rigorous convergence guarantees. In fact, even among gradient-based posterior sampling algorithms with SGM prior, only a few offer formal convergence results (Sun et al., 2024). This tension motivates our proposed approach, which seeks to combine the practical scope of derivative-free guidance with strong theoretical foundations.

**Zeroth-order sampling.** A zeroth-order (ZO) gradient estimator of a function $f$ can be obtained using a forward finite difference along a random direction (Nesterov & Spokoiny, 2017):

$$\widetilde{\nabla} f_\mu(\boldsymbol{x}, \boldsymbol{u}) := \frac{f(\boldsymbol{x} + \mu\boldsymbol{u}) - f(\boldsymbol{x})}{\mu} \, \boldsymbol{u}, \quad \boldsymbol{u} \sim \mathcal{N}(0, I), \tag{4}$$

where $\mu > 0$ is a small smoothing parameter. In our setting, $f$ is the negative log-likelihood (or potential) function $f := -\log \ell(\boldsymbol{y}|\boldsymbol{x})$. By discretizing (3) and replacing the negative log-likelihood with its ZO estimator from (4), we obtain a naive ZO MCMC Langevin sampling algorithm with SGM prior. Roy et al. (2022) established convergence guarantees for generating $\varepsilon$-approximate samples in Wasserstein-2 distance under strongly convex and smooth $f$; however, their analysis is purely theoretical, considers only settings without a prior, and assumes strongly log-concave distribution, which is an assumption typically violated for forward models in inverse problems. He et al. (2024) established asymptotic KL convergence but neither demonstrate the method on real-world problems nor consider posterior sampling.

More recently, Sun et al. (2024) proposed annealed plug-and-play Monte Carlo (APMC), the closest work to ours in the literature, and derived an upper bound on the FI, albeit under the assumption of access to forward model gradients. In contrast, we prove convergence to stationary point in $\varepsilon$-relative FI, to the squared TV distance (assuming that the potential function of the forward model satisfies Poincaré inequality), and weak convergence to the target posterior distribution $\pi(\boldsymbol{x}|\boldsymbol{y})$ in black-box setting.

## 3 METHOD

To develop our ZO-APMC method, we first provide an interpretation of annealed Langevin dynamics and intuition behind the variance-reduction mechanism for zeroth-order estimate. Then, we present our algorithm with its convergence guarantees.

**Annealed Langevin dynamics.** As discussed in Section 2, given a SGM prior $\mathcal{S}_\theta(\boldsymbol{x}, \sigma) \approx \nabla \log p(\boldsymbol{x})$, we can discretize the Langevin diffusion in (3) and get the update rule as

$$\boldsymbol{x}_{k+1} := \boldsymbol{x}_k - \gamma \left(\boldsymbol{g}_k - \mathcal{S}_\theta(\boldsymbol{x}_k, \sigma)\right) + \sqrt{2\gamma}\boldsymbol{Z}_k, \tag{5}$$

where $\boldsymbol{Z}_k \sim \mathcal{N}(0, I)$ and $\boldsymbol{g}_k$ denotes a derivative-free estimate of the negative log-likelihood score, defined below. In practice, Langevin algorithms often experience slow convergence and mode collapse when sampling from high-dimensional, multimodal distributions. Inspired by annealed importance sampling (Neal, 2001; Sun et al., 2024), we consider a sequence of posterior distributions for each step

$$\pi_{\sigma_k}^{(\alpha_k)}(\boldsymbol{x}|\boldsymbol{y}) \propto \ell(\boldsymbol{y}|\boldsymbol{x})p_{\sigma_k}^{\alpha_k}(\boldsymbol{x}), \quad p_{\sigma_k}(\boldsymbol{x}) = \int_{\mathbb{R}^d} p(\boldsymbol{z})\phi_{\sigma_k}(\boldsymbol{x} - \boldsymbol{z})\, d\boldsymbol{z}, \tag{6}$$

where $\alpha_0 > \alpha_1 > \ldots > \alpha_K = \cdots = \alpha_{N-1} = 1$, $\sigma_0 > \sigma_1 > \ldots > \sigma_K = \cdots = \sigma_{N-1} \approx 0$, and $\phi_{\sigma_k}$ is the probability density function of $\mathcal{N}(0, \sigma_k^2 I)$. $\{\alpha_k\}_k^{N-1}$ and $\{\sigma_k\}_{k=0}^{N-1}$ are generally initialized with large values in practice and they decay to one and almost zero, respectively. Initially, the weighted posterior, is dominated by a smoothed prior, enabling rapid escape from gradient plateaus where $\nabla \log \pi(\boldsymbol{x}) \approx 0$. As iterations proceed, the likelihood influence grows and the smoothed posterior $\pi_{\sigma_k}^{(\alpha_k)}$ sharpens toward the true posterior distribution $\pi$. This annealing accelerates burn-in by first flattening and then gradually restoring distributional complexity. This process is illustrated with Fig. 5 in Appendix A inspired by (Sun et al., 2024). With the annealing parameters, we can write the new update rule as

$$\boldsymbol{x}_{k+1} := \boldsymbol{x}_k - \gamma \left(\boldsymbol{g}_k - \alpha_k \mathcal{S}_\theta(\boldsymbol{x}_k, \sigma_k)\right) + \sqrt{2\gamma}\boldsymbol{Z}_k. \tag{7}$$

In a black-box setting, the negative likelihood score $\nabla f(\boldsymbol{x}_k)$ is unavailable, so a natural choice is to approximate it using a naive ZO estimator, $\boldsymbol{g}_k = (1/b)\sum_{i=1}^b \tilde{\nabla}f_\mu(\boldsymbol{x}_k, \boldsymbol{u}_i)$. However, ZO estimates are known to exhibit high variance, especially in high-dimensional settings (Nesterov & Spokoiny, 2017), because they approximate a $d$-dimensional gradient using only scalar function evaluations along random directions. This typically requires on the order of $d$ evaluations to obtain an accurate estimate. Consequently, convergence cannot be guaranteed under an arbitrary per-iteration budget; the batch size must increase with the dimension, which quickly becomes computationally prohibitive.

### 3.1 Proposed Method: ZO-APMC

We address this issue by introducing a variance-reduction mechanism for sampling, inspired by PAGE (Li et al., 2021), which enables convergence under a fixed per-iteration budget and thereby makes the algorithm practical. Our proposed ZO-APMC derivative-free sampling algorithm is given in Algorithm 1 and the negative likelihood score estimator with variance-reduction mechanism is defined in line 7. Intuitively, accurate ZO estimates require very large batch sizes per iteration, which is prohibitive, while small batches yield noisy estimates that degrade reconstruction quality. Our method addresses this by mixing both regimes: it computes a high-quality ZO estimate using a large batch $b$ occasionally (with prob. $p \ll 1$); in most of the iterations (with prob. $1-p$), it performs a cheap update using a small batch $b' \ll b$ to refine the existing high-quality estimate $\boldsymbol{g}_k$ carried over from previous iterations, using only the gradient change from $\boldsymbol{x}_k$ to $\boldsymbol{x}_{k+1}$.

While we provide the convergence results in Section 3.3, we now present an upper bound on the estimation error illustrating the variance-reduction mechanism for the proposed ZO-APMC method.

**Assumption 1** *We assume that the log-likelihood $\log \ell(\boldsymbol{x}|\boldsymbol{y})$ is Lipschitz continuous with constant $L_{f_1}$, namely, for any $\boldsymbol{x}_1, \boldsymbol{x}_2 \in \mathbb{R}^n$, $\|\log \ell(\boldsymbol{y}|\boldsymbol{x}_1) - \log \ell(\boldsymbol{y}|\boldsymbol{x}_2)\| \leq L_{f_1}\|\boldsymbol{x}_1 - \boldsymbol{x}_2\|$.*

**Remark 1.** This assumption is used in recent SGM-based inverse problem works (Renaud et al., 2024b;a). For Gaussian noise, if $\boldsymbol{A}(\cdot)$ is linear and the domain is bounded ($\|\boldsymbol{x}\|_2 \leq C$), then Lipschitzness follows directly; this includes practical cases such as images normalized to $[0, 1]^d$. The same argument follows for the Laplace noise (Huang et al., 2017) as well. In Appendix A.4, we present additional toy experiments for both noise models.

---

**Algorithm 1** ZO-APMC

---

**Input:** initial point $\boldsymbol{x}_0$, stepsize $\gamma$, minibatch size $b$, $b' < b$, probability $p \in (0, 1]$, and annealing parameters $\alpha_0 > 0, \sigma_0 > 0$.

1: $g_0 = \frac{1}{b} \sum_{i \in I} \tilde{\nabla} f_\mu(\boldsymbol{x}_0, \boldsymbol{u}_i)$   // $I$ denotes random minibatch samples with $|I| = b$.
2: **for** $k = 0, 1, \ldots, N - 1$ **do**
3:     $\boldsymbol{Z}_k \leftarrow \mathcal{N}(0, I)$
4:     $\sigma_k, \alpha_k \leftarrow \text{WeightedAnnealing}(\sigma_0, \alpha_0, k)$
5:     $\mathcal{G}_k(\boldsymbol{x}_k) \leftarrow \boldsymbol{g}_k - \alpha_k \mathcal{S}_\theta(\boldsymbol{x}_k, \sigma_k)$
6:     $\boldsymbol{x}_{k+1} \leftarrow \boldsymbol{x}_k - \gamma \mathcal{G}_k(\boldsymbol{x}_k) + \sqrt{2\gamma} \boldsymbol{Z}_k$
7:     $\boldsymbol{g}_{k+1} = \begin{cases} \frac{1}{b} \sum_{i \in I} \tilde{\nabla} f_\mu(\boldsymbol{x}_{k+1}, \boldsymbol{u}_i), & \text{with prob. } p \\ \boldsymbol{g}_k + \frac{1}{b'} \sum_{i \in I'} \tilde{\nabla} f_\mu(\boldsymbol{x}_{k+1}, \boldsymbol{u}_i) - \tilde{\nabla} f_\mu(\boldsymbol{x}_k, \boldsymbol{u}_i), & \text{with prob. } 1 - p \end{cases}$
8: **end for**
**Output:** $\boldsymbol{x}_N$

---

**Proposition 1** *Under Assumption 1, let $\{\boldsymbol{x}_k\}_{k=0}^N$ denote the iterates produced by ZO-APMC for $N > 0$ steps. Define the estimation error of the likelihood score as $\boldsymbol{e}_k := \boldsymbol{g}_k - \nabla f_\mu(\boldsymbol{x}_k)$. Then, for each step $k$, the error variance satisfies*

$$\mathbb{E}\big[\|\boldsymbol{e}_k\|^2\big] \leq (1 - p^2) \mathbb{E}\big[\|\boldsymbol{e}_{k-1}\|^2\big] + \frac{4d(1-p)L_{f_1}^2}{b'\mu^2} \mathbb{E}\big[\|\boldsymbol{x}_k - \boldsymbol{x}_{k-1}\|^2\big] + \frac{p \, \mathbb{E}\big[\|\tilde{\nabla} f_\mu(\boldsymbol{x}_k, \boldsymbol{u}_i)\|^2\big]}{b}. \tag{8}$$

The full derivation using the law of total covariance can be found in Appendix A.3.2. Proposition 1 provides an intuition on the benefit of the variance-reduction. When $p = 1$, our method reduces to the posterior sampling with naive ZO estimator in (7), which estimates the negative likelihood score using a batch of $b$ perturbations. In this scenario, the error term is only bounded by the variance term, which is the third term in (8). This variance grows at least linearly with the dimension (see property (c) in Lemma 1), so $b$ must scale with the dimension to keep the error manageable. This is computationally expensive and impractical to perform at every iteration. By choosing $p \ll 1$, we scale the variance by $p$ and obtain the effect of a large batch size, while using a smaller batch $b' \ll b$ in most of the iterations and evaluating the large batch $b$ occasionally. This reduces the variance term significantly while keeping the required average number of function evaluations per iteration smaller than the naive ZO approach. The trade-off is the introduction of two additional error terms from the "$1 - p$ branch" of the estimator. The first is an estimation error propagated from previous iterations decaying with $1 - p^2 < 1$. The second is due to the estimation error of the gradient change between $\mathbf{x}_k$ and $\mathbf{x}_{k+1}$, which depends on the regularity of the forward model (Assumption 1), the ZO smoothing parameter $\mu$, and the discretization error of the Langevin diffusion (inside the expectation). These sources of error can all be mitigated by using a smaller discretization step size.

## 3.2 OPTIMIZATION VIEW OF LANGEVIN DIFFUSION

Consider the minimization of the *Kullback–Leibler (KL)* divergence over the Wasserstein space of probability distributions.

$$\hat{\nu} = \arg\min_\nu \mathrm{KL}(\nu \| \pi) \quad \text{where} \quad \mathrm{KL}(\nu \| \pi) = \int_{\mathbb{R}^d} \nu(\boldsymbol{x}) \log \frac{\nu(\boldsymbol{x})}{\pi(\boldsymbol{x})} d\boldsymbol{x}, \tag{9}$$

where $\nu$ and $\pi$ denote the estimate and desired posterior, respectively. Similar to the gradient concept in Euclidean space, we can write the Wasserstein gradient of $\mathrm{KL}(\nu \| \pi)$ as $\nabla_\nu \mathrm{KL}(\nu \| \pi) = \nabla \log(\nu(\boldsymbol{x})/\pi(\boldsymbol{x}))$ (Ambrosio et al., 2008) and its expected square norm gives us the *relative Fisher information (FI)* $\mathrm{FI}(\nu \| \pi) = \int_{\mathbb{R}^n} \nu(x) \|\nabla \log \nu(x) - \nabla \log \pi(x)\|_2^2 \, dx$. If $\nu_t$ evolves under Langevin diffusion in (3), then $\frac{d}{dt} \mathrm{KL}(\nu_t \| \pi) = -\mathrm{FI}(\nu_t \| \pi)$ (Ambrosio et al., 2008; Villani, 2009), showing that Langevin diffusion is a gradient flow in probability space. From an optimization viewpoint, $\mathrm{FI}(\nu_t \| \pi)$ serves as the analogue of the squared $\ell_2$ gradient norm in $\mathbb{R}^d$ (Balasubramanian et al., 2022). Leveraging this analogy, we analyze the convergence of $\mathrm{FI}(\nu_t \| \pi)$ under a "linear interpolation" of the distributions generated by ZO-APMC, which in turn implies the stationarity of the discrete updates.

### 3.3 General Convergence Results

In this section, we state our main theoretical results establishing the convergence of ZO-APMC. We first state our assumptions.

**Assumption 2** *The log-prior $\log p(\boldsymbol{x})$ is differentiable and $\nabla \log p(\boldsymbol{x})$ is $L_p$-Lipschitz, i.e. $\|\nabla \log p(\boldsymbol{x}_1) - \nabla \log p(\boldsymbol{x}_2)\| \leq L_p\|\boldsymbol{x}_1 - \boldsymbol{x}_2\|$ for all $\boldsymbol{x}_1, \boldsymbol{x}_2 \in \mathbb{R}^n$.*

**Assumption 3** *Let $p_{\sigma_k}(\boldsymbol{x}) = \int_{\mathbb{R}^d} p(\boldsymbol{z})\phi_{\sigma_k}(\boldsymbol{x} - \boldsymbol{z})d\boldsymbol{z}$ denote the smoothed prior, where $\phi_{\sigma_k}$ is the probability density function of $\mathcal{N}(0, \sigma_k^2 I)$. We assume that for any $\sigma_k > 0$ and $\boldsymbol{x} \in \mathbb{R}^d$, $\|\nabla \log p_{\sigma_k}(\boldsymbol{x}) - \nabla \log p(\boldsymbol{x})\| \leq \sigma_k C$.*

**Assumption 4** *We assume that the log-likelihood $\log \ell(\boldsymbol{x}|\boldsymbol{y})$ is differentiable and has a Lipschitz continuous gradient with constant $L_{f_2} > 0$ for any $\boldsymbol{x}_1, \boldsymbol{x}_2 \in \mathbb{R}^n$, that is, $\|\nabla \log \ell(\boldsymbol{y}|\boldsymbol{x}_1) - \nabla \log \ell(\boldsymbol{y}|\boldsymbol{x}_2)\| \leq L_{f_2}\|\boldsymbol{x}_1 - \boldsymbol{x}_2\|$.*

**Assumption 5** *For any $\sigma_k > 0$ and all $\boldsymbol{x} \in \mathbb{R}^d$, the score network satisfies $\|\mathcal{S}_\theta(\boldsymbol{x}, \sigma_k) - \nabla \log p_{\sigma_k}(\boldsymbol{x})\| \leq \varepsilon_{\sigma_k} < \infty$ and $\|\mathcal{S}_\theta(\boldsymbol{x}, \sigma_k)\| \leq \sigma_k^{-1}C$.*

Assumptions 2 and 4 correspond to standard conditions commonly adopted in the non-log-concave sampling literature (He & Zhang, 2025; Guo et al., 2024; Balasubramanian et al., 2022), and Assumption 3 captures the perturbation of the prior as in (Sun et al., 2024). The first part of Assumption 5 states that we have access to a score network with sufficient generalization performance. While Vincent (2011) show that the minimizer of the denoising score matching objective satisfies $\mathcal{S}_{\theta^*}(\boldsymbol{x}, \sigma_k) = \nabla \log p_{\sigma_k}(\boldsymbol{x})$ almost surely, we instead assume a bounded approximation error that depends on $\sigma_k$, following prior work on sampling with SGM priors (Sun et al., 2024; Lee et al., 2023; 2022). We further assume that the score magnitude scales with the noise level, consistent with the empirical observation $|\mathcal{S}_\theta(\boldsymbol{x}, \sigma_k)| \propto 1/\sigma_k$ (Song & Ermon, 2019; 2020). In contrast to related work (Sun et al., 2024; Yang & Wibisono, 2022; Lee et al., 2022), we do not impose Lipschitz continuity on the SGM prior, and we do not require the likelihood distribution to be log-concave.

**Theorem 1** *Let $\{\alpha_k\}_{k=0}^{N-1}$, $\{\sigma_k\}_{k=0}^{N-1}$ be decreasing annealing schedules with $\alpha_{K,...,N-1} = 1$, and let $\{\nu_t\}_{t \geq 0}$ denote the law of the continuous interpolation of $\{\boldsymbol{x}_k\}_{k=0}^N$ produced by ZO-APMC with $N > 0$ iterations under Assumptions 1–5. For any step size $\gamma \in \left(0, 1/(L_m\sqrt{85\phi(\mu)})\right]$, where $\phi(\mu) = 1 + 4(1-p)d/(p\mu^2 b')$ and $L_m = \max\{L_\pi, L_{f_1}\}$, the Fisher information satisfies*

$$\frac{1}{N\gamma}\int_0^{N\gamma} \mathrm{FI}(\nu_t\|\pi)\,dt \leq \frac{C_0}{N\gamma} + 8\gamma L_m^2 d\phi(\mu) + \frac{17d(d+2)L_{f_1}^2}{2b} + \frac{17\mu^2 L_{f_2}^2(d+3)^3}{8} + \bar{\sigma}^2 + \bar{\varepsilon}_\sigma^2 + \bar{\alpha}^2,$$

(10)

*where $\bar{\sigma}^2 = \frac{51C^2}{2N}\sum_{k=0}^{N-1}\sigma_k^2$, $\bar{\varepsilon}_\sigma^2 = \frac{51}{2N}\sum_{k=0}^{N-1}\varepsilon_{\sigma_k}^2$, $\bar{\alpha}^2 = \frac{51C^2}{2N}\sum_{k=0}^{N-1}\frac{(\alpha_k-1)^2}{\sigma_k^2}$, and $C$, $C_0$ are numerical constants. Furthermore, let $\gamma = \sqrt{C_0}/(2L_m\sqrt{Nd\phi(\mu)})$, $b = \lceil 51d(d+2)L_{f_1}^2/\varepsilon\rceil$, $p = 1/b$, $\mu = 2\sqrt{\varepsilon}/\sqrt{51}L_{f_2}(d+3)^{3/2}$ with schedule $\sigma_k = O(k^{-\beta})$ and score network error satisfying $\varepsilon_{\sigma_k} = O(k^{-\beta})$ for any $\beta \geq 1/2$ and for all $k \geq 1$ with initial values $\sigma_0 > 0$ and $\varepsilon_{\sigma_0} > 0$. Then an $\varepsilon$-approximate solution to (10) requires $O\left(d^7 L_m^6/\varepsilon^4\right)$ forward model evaluations.*

Theorem 1 (proof provided in Appendix A.3.3) shows ZO-APMC achieves $\varepsilon$-approximate solution in the Fisher information sense with $O\left(d^7 L_m^6/\varepsilon^4\right)$ forward model evaluations while using a fixed per-iteration budget $pb = O(1)$ on average.

**Remark 2.** In practice, the annealing schedules $\{\alpha_n\}_{n=0}^{N-1}$ and $\{\sigma_n\}_{n=0}^{N-1}$ are typically implemented using geometric decay (Sun et al., 2024; Song & Ermon, 2019), which decreases more rapidly than the polynomial rates selected for our analysis. Moreover, the condition $\varepsilon_{\sigma_k} = O(k^{-\beta})$ with $\beta \geq 1/2$ characterizes the decay of the SGM generalization complexity across each noise level at step $k$, which can be satisfied with large data and minimization of denoising score matching loss (Vincent, 2011). Recent studies on the generalization of SGMs report similar rates as the one used in our analysis (Fu & Lee, 2025; Zhang et al., 2024; Oko et al., 2023).

Intuitively, Theorem 1 shows that ZO-APMC progressively improves its average estimate of the directions that lead to reconstructions consistent with the measurements, while relying only on forward model inputs and outputs (not on gradients). Achieving at most $\varepsilon$ average error in these directions requires $O(1/\varepsilon^4)$ function evaluations in total. Although this number grows polynomially

with dimension, it remains achievable under any fixed per-iteration budget of function evaluations. Moreover, the error bias in the ZO estimates can be reduced arbitrarily by choosing a smaller $\mu$, which in turn requires using a smaller step size $\gamma$ while keeping the number of function evaluations per iteration fixed. Leveraging the results of Theorem 1, we show that if the target posterior $\pi$ further satisfies the Poincaré inequality, ZO-AMPC enjoys stronger sampling guarantees in squared TV distance.

**Assumption 6** *For every smooth, compactly supported function $f : \mathbb{R}^d \to \mathbb{R}$, the posterior distribution $\pi(\boldsymbol{x}|\boldsymbol{y})$ satisfies the Poincaré inequality $\mathrm{Var}_\pi(f) \leq C_{\mathrm{PI}}\mathbb{E}_\pi[\|\nabla f\|^2]$.*

**Corollary 1** *Let $\{\alpha_k\}_{k=0}^{N-1}$, $\{\sigma_k\}_{k=0}^{N-1}$ be decreasing annealing schedules with $\alpha_{K,\dots,N-1} = 1$, and $\{\nu_t\}_{t\geq0}$ denote the law of the continuous interpolation $\{\boldsymbol{x}_k\}_{k=0}^N$ of ZO-APMC, and let the Assumptions Assumptions 1–6 hold. Then, if we choose $\gamma = \sqrt{C_0 C_{\mathrm{PI}}}/2L_m\sqrt{Nd\phi(\mu)}$, we have*

$$\|\bar{\nu}_{N\gamma} - \pi\|_{\mathrm{TV}}^2 \leq 16L_m\sqrt{\frac{C_0 C_{\mathrm{PI}}d\phi(\mu)}{N}} + \frac{34d(d+2)C_{\mathrm{PI}}L_{f_1}^2}{b} + \frac{17}{2}\mu^2 C_{\mathrm{PI}}L_{f_2}^2(d+3)^3$$
$$+ 4C_{\mathrm{PI}}(\bar{\sigma}^2 + \bar{\varepsilon}_\sigma^2 + \bar{\alpha}^2) \quad (11)$$

*where $\bar{\nu}_{N\gamma} \coloneqq (N\gamma)^{-1}\int_0^{N\gamma}\nu_t dt$, and the quantities $\bar{\sigma}$, $\bar{\varepsilon}_\sigma$, and $\bar{\alpha}$ are given in Theorem 1. If we choose $b = \lceil 204d(d+2)C_{\mathrm{PI}}L_{f_1}^2/\varepsilon\rceil$, $p = 1/b$, $\mu = \sqrt{\varepsilon}/L_{f_2}(d+3)^{3/2}\sqrt{51C_{\mathrm{PI}}}$ with schedule $\sigma_k = O(k^{-\beta})$ and score network error satisfying $\varepsilon_{\sigma_k} = O(k^{-\beta})$ for any $\beta \geq 1/2$ and $k \geq 1$ with initial values $\sigma_0 > 0$ and $\varepsilon_{\sigma_0} > 0$, an $\varepsilon$-approximate solution to (11) requires $O(d^7 L_m^6 C_{\mathrm{PI}}^3/\varepsilon^4)$ forward model evaluations.*

Theorem 1 (proof provided in Appendix A.3.4) shows ZO-APMC achieves $\varepsilon$-approximate solution in the stronger squared TV distance sense with $N = O(d^7 L_m^6 C_{\mathrm{PI}}^3/\varepsilon^4)$ forward-model evaluations and using a fixed evaluation budget $pb = O(1)$ per iteration.

**Remark 3.** To generate a sample from $\bar{\nu}_{N\gamma}$, one may proceed as follows. First, draw a time $t \in [0, N\gamma]$ uniformly at random and determine the largest integer $k$ such that $k\gamma \leq t$. Then perform a linear interpolation between the interval $[k\gamma, t]$ to produce $\boldsymbol{x}_t$ according to the update rule in line 7 of ZO-APMC. The resulting $\boldsymbol{x}_t$ is sample from $\bar{\nu}_{N\gamma}$.

Corollary 1 implies that if the likelihood density satisfies the Poincaré inequality (i.e. Gaussian measurement noise with a linear forward model), meaning it has no flat valleys, widely separated modes, or heavy tails, then the samples produced by ZO-APMC are statistically indistinguishable from the images drawn from the true posterior conditioned on measurements, up to an $\varepsilon$ error, after $O(1/\varepsilon^4)$ total function evaluations with fixed per-iteration budget.

**Theorem 2** *Let $\{\nu_t\}_{t\geq0}$ denote the law of the continuous interpolation of the ZO-APMC iterates $\{\boldsymbol{x}_k\}_{k=0}^N$, and assume Assumptions 1–5 hold for $\pi$. Suppose ZO-APMC is initialized at $\nu_0$ with $\mathrm{KL}(\nu_0\|\pi) < \infty$ and uses $\gamma_k = \sqrt{b'}/(kL_m\sqrt{680d})$, $b_k = \lceil k^{1/2}\rceil$, $p_k = k^{-1/2}$, $\mu_k = k^{-1/8}$ with schedule $\sigma_k = O(k^{-\beta})$ and score network error $\varepsilon_{\sigma_k} = O(k^{-\beta})$ for any $\beta > 0$ and $k \geq 1$ with initial values $\sigma_0 > 0$ and $\varepsilon_{\sigma_0} > 0$. Then the time-averaged law $\bar{\nu}_{\tau_n} = \tau_n^{-1}\int_0^{\tau_n}\nu_t\,dt$, where $\tau_n = \sum_{k=1}^n \gamma_k$, converges in distribution to $\pi$ as $k \to \infty$.*

We present the complete proof in Appendix A.3.5. For solving ill-posed inverse problems, the most principled objective is to sample from the true posterior, since many reconstructions are consistent with the measurements. Intuitively, Theorem 2 shows that with very long ZO-APMC iterations under decaying annealing parameters (i.e., geometric annealing) and decaying score-approximation error, the ZO-APMC reconstructions converge to the target posterior. That is, each generated sample corresponds to a plausible underlying image or object consistent with the measured data, and this can be achieved with fixed per-iteration cost $pb = O(1)$. To the best of our knowledge, this is the first work to establish asymptotic convergence of ZO Langevin MCMC sampling for non-log-concave distributions. This result follows directly from Theorem 1 together with the fact that $\mathrm{FI}(\nu\|\pi) = 0$ implies $\nu = \pi$.

**Remark 4**. In inverse problems where the forward model can be implemented numerically via PDEs, automatic differentiation may be possible to calculate gradients but often produces unstable gradients when PDE stability constraints are violated (Zheng et al., 2025b) or when the physics of the model is inherently discrete (Moës et al., 1999). We further theoretically analyze this instability as additive Gaussian noise in Appendix A.1.2. Removing the Lipschitz continuity assumption on the log-likelihood (Assumption 1),

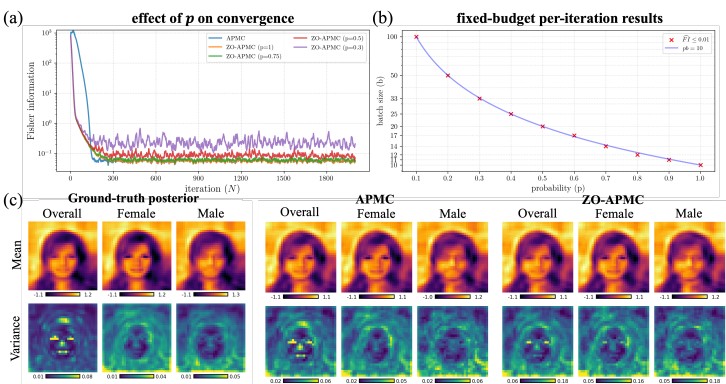

Figure 1: (a) Convergence of ZO-APMC with $b = 10$, $b' = 5$, and $\varepsilon_{k^*} = 2.5$ for various $p$, alongside APMC convergence with gradient access. (b) Convergence results for fixed-budget per-iteration. Each red "x" marks cases where the FI falls below $0.01$ after 2000 iterations for the corresponding $p$ and $b$. (c) Comparison of sample statistics obtained by ZO-APMC and APMC versus the ground-truth posterior.

we show that a lightly modified likelihood-score estimator still achieves FI convergence with a fixed per-iteration budget $pb = O(1)$, requiring $O\left(L_\pi^2 \sigma_{\text{noise}}^4 d^2 / \varepsilon^4\right)$ iterations.

## 4 EXPERIMENTS

**Baselines.** Our primary focus is on gradient-free methods, which assume only black-box access to the forward model. We therefore benchmark against three gradient-free baselines: SCG (Huang et al., 2024), DPG (Tang et al., 2024), and EnKG (Zheng et al., 2025a). We also include the Forward-GSG and Central-GSG baselines, introduced by (Zheng et al., 2025a). These methods resemble Diffusion Posterior Sampling (DPS) (Chung et al., 2022) but approximate the forward-model gradient using Tweedie's formula together with forward and central ZO estimates of the forward score function. For completeness, we also evaluate gradient-based methods in settings where the forward-model gradient is available. Specifically, we compare our algorithm with DPS (Chung et al., 2022), PnPDM (Wu et al., 2024), and APMC (Sun et al., 2024), which is an annealed Langevin MC posterior sampling algorithm with gradient access and the closest approach to ours.

### 4.1 TOY EXPERIMENTS

**Numerical Validation.** We test our theory that ZO-APMC converges in FI with fixed per-iteration cost on a synthetic bimodal 2D Gaussian-mixture prior with random $A$ with $\xi \sim \mathcal{N}(0, I)$.

Using the analytical score with added Gaussian noise $\varepsilon_{k^*} = 2.5$ to mimic SGM error, we generate 1000 samples with ZO-APMC from 20 random initializations and report the mean FI relative to the analytical posterior. Fig. 1a shows that with $b = 10$, $b' = 5$, ZO-APMC converges near zero for $p \in \{1, 0.75, 0.5\}$, matching gradient-based APMC but becoming unstable at $p = 0.3$ due to fixed $b$. Fig. 1b shows that increasing $b$ while keeping $pb = 10$ restores stability and achieves convergence ($FI \le 0.01$), confirming our theoretical results.

**Statistical Validation.** We assess ZO-APMC's ability to sample from multiple modes of a multimodal distribution under black-box conditions. Using the same setup as our previous validation with random $A \in \mathbb{R}^{115 \times 1024}$, we construct a two-mode Gaussian mixture prior from CelebA (Liu et al., 2018) images normalized to $[0, 1]$, where the "male" and "female" attributes define the modes. To ensure clear separation between the modes, we shift them by $+1$ and $-1$, respectively. A shallow SGM is then trained by customizing the U-Net from (Nichol & Dhariwal, 2021) for this data. As shown in Fig. 1c, ZO-APMC with $p = 0.5$, $b = 50$, $b' = 5$, accurately recovers the posterior statistics of both modes, comparable to APMC with gradient access, though with slightly higher variance due to ZO estimation, which can be mitigated by increasing $b$. For extended results and further details of validations, see Appendix A.4.

Table 1: Quantitative comparison with baselines. The best values of each metric for black-box and gradient-access settings are highlighted in **bold** and underline, respectively.

| | PSNR (dB)↑ | SSIM↑ | NRMSE↓ | SD↓ | MSE↓ |
|---|---|---|---|---|---|
| PnPDM | 30.81 | 0.946 | 3.76e-2 | 2.16e-2 | 8.46e-4 |
| DPS | 34.38 | 0.965 | 2.54e-2 | 2.06e-2 | 4.07e-4 |
| APMC | 36.55 | 0.973 | 1.99e-2 | 2.0e-2 | 2.55e-4 |
| Forward-GSG | 27.8 | 0.918 | 5.42e-2 | 3.26e-2 | 19.1e-4 |
| Central-GSG | 27.78 | 0.917 | 5.43e-2 | 3.27e-2 | 19.2e-4 |
| SCG | 7.1 | 0.711 | 7.67 | 1.38 | 0.21 |
| DPG | 32.17 | 0.953 | 5.4e-2 | **2.69e-2** | 6.5e-4 |
| EnKG | 31.32 | 0.934 | 5.72e-2 | 2.92e-2 | 6.72e-4 |
| ZO-APMC (ours) | **35.29** | **0.966** | **2.28e-2** | 2.99e-2 | **3.29e-4** |

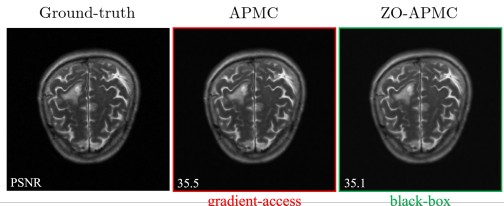

Figure 2: Visual comparison of pathological brain MRI with corresponding PSNR values obtained using APMC with gradient access and ZO-APMC in the black-box setting.

## 4.2 MAGNETIC RESONANCE IMAGING (MRI)

Image inverse problems (i.e., MRI recon.) are widely used benchmarks. Although we focus on more challenging black-box forward models, we also evaluate our method on the linear MRI recon. problem for completeness and demonstrate the capability of our variance-reduction mechanism on high-resolution data.

**Problem Setting** We consider the radial subsampling mask with acceleration factor of $4\times$. For evaluation, we use the SGM prior from Sun et al. (2024), which was pre-trained on the FastMRI brain dataset (Zbontar et al., 1811), and evaluate all algorithms on a separate test set provided in that work to ensure a consistent comparison. We randomly select 40 images at a resolution of $256 \times 256$ pixels and generate 20 reconstructions per algorithm. For each method, we report the mean image-quality metrics along with the average per-pixel standard deviation (SD). In this experiment, we use $p = 0.2$, $b = 10^4$, and $b' = 10^3$ to run ZO-APMC.

**Results** Table 1 shows that ZO-APMC consistently achieve higher reconstruction quality than other black-box baselines in all image quality metrics and closely matches the APMC with gradient access. Fig. 2 further demonstrates that both ZO-APMC and APMC yield visually indistinguishable pathological brain MRI reconstructions, with ZO-APMC accurately capturing fine details without gradient information. Our method yields slightly higher standard deviation than DPG but this can be alleviated by increasing $p$, albeit at increased computational cost.

## 4.3 BLACK-HOLE IMAGING

**Problem Setting** The black-hole interferometric imaging system reconstructs images of black holes from "visibility" measurements collected by Earth-based telescope arrays. We adopt the SGM prior (pre-trained on the GRMHD dataset at $64\times64$ resolution), the highly non-linear forward model, and the 100-sample test set, as provided by the InverseBench benchmark (Zheng et al., 2025b; Wong et al., 2022). For each method, we generate five samples and report their mean results. Since the resolution of the images are low, we use $p = 1$ with $b = 1024$. Evaluation is based on the chi-square errors of the closure phases ($\chi^2_{\text{cph}}$) and closure amplitudes ($\chi^2_{\text{camp}}$), which quantify how well the reconstructions fit the measurements. Because the black-hole imaging system captures only low spatial frequencies, we follow Akiyama et al. (2019) and compute PSNR for both the original and blurred reconstructions at the system's intrinsic resolution.

**Results** Fig. 3 shows two examples of black-hole reconstructions of our ZO-APMC method and other gradient-free baselines against the ground truth. ZO-APMC yields black-hole reconstructions with visual characteristics most closely match the ground truth among other baselines. Table 2 shows the quantitative comparison. ZO-APMC outperforms all baselines across metrics except SD, which can be mitigated by increasing batch size $b$ at additional cost.

## 4.4 NAVIER-STOKES EQUATION

**Problem Setting** The Navier–Stokes equation is a standard fluid-dynamics benchmark (Iglesias et al., 2013), widely used from ocean dynamics to climate modeling, where atmospheric observations calibrate initial conditions for numerical forecasts. Computing forward-model gradients via auto-differentiation is impractical because it requires differentiating through a PDE solver. We evaluate the gradient-free methods on 10 test samples from InverseBench using the SGM prior provided

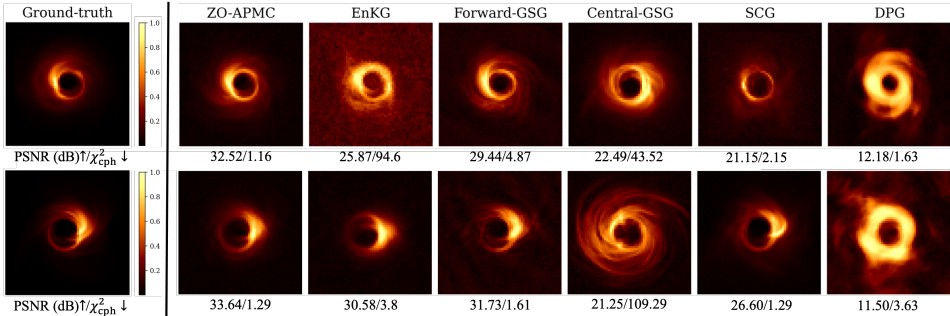

Figure 3: Visualization of samples generated for the black-hole imaging inverse problem. Reconstructions of two examples by gradient-free methods are shown in the top and bottom rows.

Table 2: Quantitative evaluation of reconstructed black-hole images (SD: sample standard deviation).

| | PSNR↑ | Blurred PSNR↑ | $\chi^2_{cph}\downarrow$ | $\chi^2_{camp}\downarrow$ | SD↓ |
|---|---|---|---|---|---|
| PnPDM | 26.48 | 32.31 | 11.48 | 23.54 | 4.5e-2 |
| DPS | 25.61 | 30.84 | 12.39 | 17.72 | 4.32e-2 |
| APMC | 26.23 | 31.32 | 11.78 | 19.23 | 4.34e-2 |
| Forward-GSG | 26.21 | 31.47 | 6.77 | 14.06 | 2.99e-2 |
| Central-GSG | 21.63 | 23.73 | 80.31 | 78.5 | 4.5e-2 |
| SCG | 22.21 | 25.51 | 23.72 | 14.23 | 1.7e-2 |
| DPG | 12.33 | 14.02 | 8.17 | 30.44 | **1.6e-2** |
| EnKG | 22.86 | 27.69 | 64.37 | 33.44 | 0.925 |
| ZO-APMC (Ours) | **26.71** | **32.86** | **5.42** | **11.23** | 3.02e-2 |

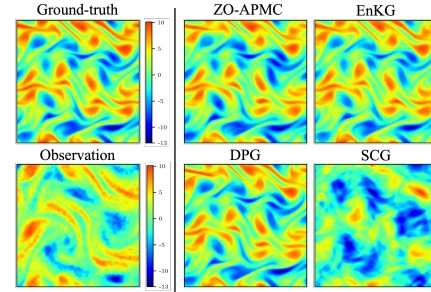

Figure 4: Visualization of results on Navier-Stokes inverse problem.

by the benchmark, generating five reconstructions per method and reporting the mean performance. We report the NRMSE (relative $\ell_2$ error) to evaluate the accuracy of reconstructions with sample SD. For additional details on the experiments, please refer to Appendix A.5.3.

**Results.** Fig. 4 demonstrates that ZO-APMC produces solutions that qualitatively preserve key flow features, comparable to EnKG and DPG, while SCG fails. Moreover, EnKG yields noticeably noisier reconstructions than ZO-APMC. Additional quantitative results and more representative cases, showing our algorithm's performance comparable to the baselines, are provided in Appendix A.5.3.

## 5 CONCLUSION

We proposed ZO-APMC, the first provable derivative-free framework for posterior sampling with a pre-trained SGM prior. It provides non-asymptotic complexity guarantees for reaching an $\varepsilon$-relative Fisher information stationary point and provably converges to the target posterior under mild assumptions using only forward-model evaluations. Toy experiments confirm that our variance-reduction scheme with fixed per-iteration cost ensures convergence in Fisher information across batch sizes, while the annealing mechanism enables accurate sampling from multimodal distributions. On both linear and highly non-linear inverse problems, ZO-APMC matches the performance of state-of-the-art gradient-free methods. The main limitations are higher runtime than gradient-based methods due to Langevin diffusion and the absence of manifold projection as in Chung et al. (2022). Future work includes extending our theoretical analysis with Riemannian zeroth-order derivative estimation (Li et al., 2023) and incorporating faster sampling methods (Yin et al., 2024; Song et al., 2023c).

## 6 REPRODUCIBILITY STATEMENT

The full design of the toy experiments is detailed in Appendix A.4. For image-based experiments, we employed publicly available datasets. For all inverse problems other than brain MRI, we adopted the forward models from the reference implementations provided by the InverseBench benchmark Zheng et al. (2025b), while for brain MRI experiments we followed the implementation of Sun et al. (2024). Furthermore, we use the original implementations of all baseline methods and include our code as supplementary material with the submission. e

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

# A  APPENDIX

## A.1  COMPARISON TO RELATED WORK

In this subsection, we have compared our algorithm with other representative methods in the literature with their limitations and key benefits.

### A.1.1  BLACK-BOX POSTERIOR SAMPLING ALGORITHMS

There are three other prior works that studied posterior sampling with SGM priors for black-box (derivative-free) forward models. The most recent is EnKG, which estimates the likelihood score using the ensemble-based statistical linearization (Zheng et al., 2025a). Unlike our algorithm, EnKG is not designed to sample from the full posterior and therefore offers limited uncertainty quantification. In our experiments, we observed that EnKG also becomes highly inefficient when the evaluation of the forward model is cheaper than computation of the score network, which is the case in MRI reconstruction with linear forward model. In addition, as reflected in our results, in black hole imaging and MRI reconstruction, it produced noticeably lower reconstruction quality than our method. However, it performed better than our method for Navier-Stokes inverse problem because noisy function evaluations of forward model makes the guidance term unstable. In EnKG paper, authors proposed Forward-GSG and Central-GSG baselines, which are originated from DPS algorithm but the guidance term is estimated by mini-batch ZO estimate. Since they have additional approximation error due to heuristic approximation of the guidance term (Chung et al., 2022), they generate samples with larger reconstruction errors, as reflected in our results. However, they require less iteration as they use SDE from diffusion models Song et al. (2020b) as opposed to Langevin sampling, which requires longer iterations.

SCG (Huang et al., 2024) and DPG (Tang et al., 2024), which cast diffusion guidance in a stochastic control framework and steer the sampling process via an estimated value function. Similar to EnKG, neither of the targets to sample from the posterior distribution due to intractable score likelihood term, so uncertainty quantification cannot be done as opposed to our algorithm. Moreover, SCG relies on a threshold parameter that disables the likelihood-based guidance in certain iterations. We found that improper tuning of this threshold leads to severely degraded performance, yet the method provides no theoretical guidance for selecting it. As a result, practitioners must rely on grid search or extensive hyperparameter tuning. However, in our setting, $p$ and $b$ can be adjusted easily once a budget for per-iteration is determined. Although we did not experiment with, since SCG is based on value function (not approximation of gradient), it could perform better than our method for rule-based inverse problems.

Similar to our approach, DPG also relies on Monte Carlo mini-batch estimates of the posterior score. However, unlike our method, DPG does not include any variance-reduction mechanism, and incorporating such mechanisms is nontrivial due to the DDPM (Ho et al., 2020) and DDIM (Song et al., 2020a) sampling procedures it depends on. In addition, their update rule couples the likelihood and prior score terms, preventing the decoupling that enables our variance-reduced design. Empirically, these differences lead to clear performance gaps: in both the black hole and MRI reconstruction tasks, our method achieves substantially better reconstruction quality, whereas on the Navier–Stokes example DPG performs slightly better, which may be due to instability in the ZO score approximations for this specific problem.

### A.1.2  GRADIENT-BASED POSTERIOR SAMPLING ALGORITHMS

There are many gradient-based posterior sampling with SGM priors proposed for solving inverse problems. As can be seen from our MRI reconstruction experiments (Table 1), APMC (Sun et al., 2024), which is the closest work to ours, generated slightly better reconstructions than the proposed ZO-APMC algorithm. This shows that when one can compute the gradient of the forward model exactly, it is more preferable. However, this may not be possible in commercial MRI applications or closed-source systems (Karakuzu et al., 2025), where one can only have access to input and output of the forward operator. In that scenario, our model performs the best among black-box posterior samplers with comparable performance to the gradient-based model. Similar trend can be seen in black hole imaging experiments as well. Similar to our work, the APMC paper also derived an upper bound on the FI. However, beyond establishing an FI bound in the black-box setting, we additionally

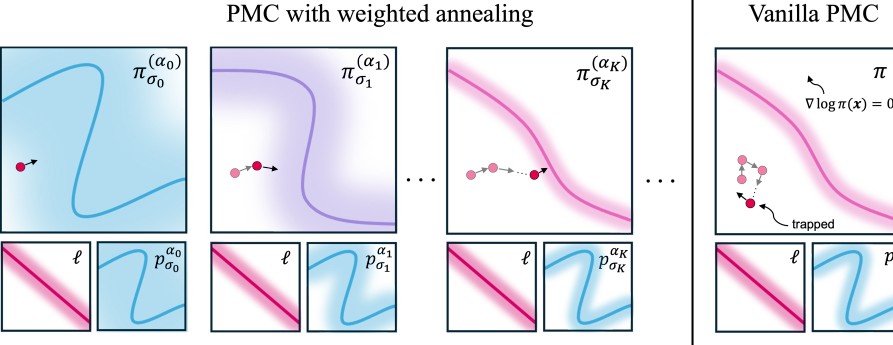

Figure 5: Illustrating how weighted annealing improves the convergence of PMC algorithms by introducing weighted posteriors $\{\pi_{\sigma_k}^{(\alpha_k)}\}$. The solid lines and shaded regions indicate the mean and probability density of the distribution, respectively, while the unshaded area corresponds to $\nabla \log p(\boldsymbol{x}) = 0$. Weighted annealing helps the vanilla PMC algorithm escape plateaus in $\nabla \log p(\boldsymbol{x})$ by gradually reducing the prior's smoothing parameter $\sigma_k$ and its weight w.r.t. the likelihood $\ell$.

prove convergence in squared total variation distance and show weak asymptotic convergence to the target distribution without using Lipschitzness assumption of the score network.

Besides APMC, we used DPS (Chung et al., 2022) and PnP-DM (Wu et al., 2024) posterior sampling algorithms as gradient-based baselines. Although our method treats the forward model as black-box, it performed better than DPS in experiments. We attributed this to the heuristic approximation of the guidance term in DPS, which is called Jensen gap which cannot be controlled explicitly as we control the estimator error via PAGE variance reduction mechanism. PnP-DM provides a principled framework for posterior sampling, but it applies to inverse problems only when implemented within the EDM framework Karras et al. (2022) as opposed to our method. Moreover, although PnP-DM establishes upper bounds on FI, it provides neither asymptotic convergence to the true posterior nor non-asymptotic guarantees in squared total variation distance. Consequently, it remains unclear whether samples produced after many iterations will coincide with those from the exact true posterior.

## A.2 NOISY GRADIENT SETTING

In this section, we consider settings where the likelihood score (data-consistency gradient) is, in principle, accessible, but the forward operator is numerically unstable such as PDE-based simulators or physics-driven models. In these cases, each gradient evaluation is contaminated by solver-induced perturbations, leading to a *noisy* and unreliable estimate of the true likelihood gradient. We model this instability with Gaussian noise as follows

$$\nabla f(\boldsymbol{x}, \boldsymbol{u}) = \nabla f(\boldsymbol{x}) + \sigma_{\text{noise}} \boldsymbol{u}, \quad \boldsymbol{u} \sim \mathcal{N}(0, I), \tag{12}$$

where we recall $f(\boldsymbol{x}) := -\log \ell(\boldsymbol{y}|\boldsymbol{x})$. Our algorithm can still operate effectively in this regime with a small modification, and it retains the key advantage of a fixed per-iteration computational budget as in the black-box ZO setting. We note that the direct application of our estimator in Algorithm 1, with probability $1 - p$, perform the following update:

$$\boldsymbol{g}_{k+1} = \boldsymbol{g}_k + \frac{1}{b'} \sum_{i \in I'} \nabla f(\boldsymbol{x}_{k+1}, \boldsymbol{u}_i) - \nabla f(\boldsymbol{x}_k, \boldsymbol{u}_i), \tag{13}$$

where $\boldsymbol{u}_i$ is source of the noise with the same random seed. Although this modification is straightforward to implement in the pure ZO setting, it becomes impractical once we account for the inherent forward-operator noise $\boldsymbol{u}$, which cannot be controlled or reused across gradient evaluations. Instead, in practice, we have

$$\boldsymbol{g}_{k+1} = \boldsymbol{g}_k + \frac{1}{b'} \sum_{i \in I'} \nabla f(\boldsymbol{x}_{k+1}, \boldsymbol{u}_i) - \nabla f(\boldsymbol{x}_k, \boldsymbol{u}_i'), \tag{14}$$

where $\boldsymbol{u}_i \neq \boldsymbol{u}_i'$. Note that this introduces a variance of $2\sigma_{\text{noise}}^2$, which cannot be reduced without choosing $b' = O(\sigma_{\text{noise}}^2)$ and this grows with dimensionality. This behavior is undesirable because it prevents us from selecting an arbitrary fixed computational budget per iteration and eliminates the variance-reduction advantage. Under the variance-reduction scheme, the benefit is that, with a constant small batch size $b' = O(1)$, we can choose a large batch size $b$ and sampling ratio $p$ such that $pb = O(1)$, while still maintaining convergence in the FI. Intuitively, because we now have access to noisy gradient evaluations rather than ZO estimates, the variance of our large-batch estimate is substantially reduced. Consequently, in this setting the discrepancy between successive gradient evaluations is small given $\boldsymbol{x}_{k+1} \approx \boldsymbol{x}_k$ when using a sufficiently small step size $\gamma$. This justifies modifying our estimator using the following approximation $\nabla f(\boldsymbol{x}_{k+1}) \approx \nabla f(\boldsymbol{x}_k)$.

$$\boldsymbol{g}_k = \begin{cases} \frac{1}{b} \sum_{i \in I} \nabla f(\boldsymbol{x}_k, \boldsymbol{u}_i), & \text{with prob.} \quad p \\ \boldsymbol{g}_{k-1}, & \text{with prob.} \quad (1-p) \end{cases} \tag{15}$$

where $|I| = b$. That is, with probability $p$, we estimate the gradient with large batch size, and with probability $1 - p$, we use previous estimate. Our LMC update structure for posterior sampling remains the same

$$\boldsymbol{x}_{k+1} \coloneqq \boldsymbol{x}_k - \gamma(\boldsymbol{g}_k - \alpha_k \mathcal{S}_\theta(\boldsymbol{x}_k, \sigma_k)) + \sqrt{2\gamma}\boldsymbol{Z}_k, \quad \boldsymbol{Z}_k \sim \mathcal{N}(0, I). \tag{16}$$

We present the corresponding convergence result in FI below.

**Theorem 3** *Let $\{\alpha_k\}_{k=0}^{N-1}$, $\{\sigma_k\}_{k=0}^{N-1}$ be decreasing annealing schedules with $\alpha_{K,\dots,N-1} = 1$, and let $\{\nu_t\}_{t\geq 0}$ denote the law of the continuous interpolation of $\{\boldsymbol{x}_k\}_{k=0}^N$ produced by the update in (16) with estimator in (15) for $N > 0$ iterations under Assumptions 2–5. For any step size $\gamma \in (0, 1/(L_\pi\sqrt{52\phi(p)})]$, where $\phi(p) = 1 + 4(1-p)/p^2$, the Fisher information satisfies*

$$\frac{1}{N\gamma}\int_0^{N\gamma} \text{FI}(\nu_t \| \pi)\, dt \leq \frac{C_0}{N\gamma} + \frac{13\sigma_{noise}^2}{b} + \frac{17}{2}\gamma L_\pi^2 d\phi(p) + \bar{\sigma}^2 + \bar{\varepsilon}_\sigma^2 + \bar{\alpha}^2, \tag{17}$$

*where $\bar{\sigma}^2 = \frac{39C^2}{N}\sum_{k=0}^{N-1}\sigma_k^2$, $\bar{\varepsilon}_\sigma^2 = \frac{39}{N}\sum_{k=0}^{N-1}\varepsilon_{\sigma_k}^2$, $\bar{\alpha}^2 = \frac{51C^2}{2N}\sum_{k=0}^{N-1}(\alpha_k - 1)^2\sigma_k^{-2}$, and $C$, $C_0$ are numerical constants. Furthermore, let $\gamma = \sqrt{2C_0}/(L_\pi\sqrt{17Nd\phi(p)})$, $b = \lceil 65\sigma_{noise}^2/\varepsilon\rceil$, $p = 1/b$ with schedule $\sigma_k = O(k^{-\beta})$ and score network error satisfying $\varepsilon_{\sigma_k} = O(k^{-\beta})$ for any $\beta \geq 1/2$ and for all $k \geq 1$ with initial values $\sigma_0 > 0$ and $\varepsilon_{\sigma_0} > 0$. Then an $\varepsilon$-approximate solution to (17) requires $O\left(L_\pi^2\sigma_{noise}^4 d^2/\varepsilon^4\right)$ gradient calculations.*

The proof is provided in Section A.3.6. In contrast to the ZO sampling analysis, we do *not* require Lipschitz continuity of the noisy gradients. In the original PAGE-style estimator of the proposed ZO-APMC, variance reduction relies on the correction step being sufficiently accurate, even when computed using a small mini-batch with high variance. This accuracy is ensured by a Lipschitzness assumption, which prevents the function from changing too abruptly. In the ZO setting, because the estimator cannot use exact gradients, one must instead assume Lipschitz continuity of the function itself to control the error of the zeroth-order gradient estimator. Here, however, we remove the correction step entirely, and as a result, this regularity assumption is no longer needed. Again, similar to our results in Theorem 1, we achieved convergence in FI with fixed per-iteration budget $pb = O(1)$ in expectation. Since we assume access to gradients although they are noisy, the dimensional dependence is weaker than in Theorem 1. Nevertheless, the rate still retains an explicit $d^2$ dependence due to discretization error, and $d$ also appears implicitly through the noise variance term $\sigma_{\text{noise}}^2$, which grows with dimension.

We validate Theorem 3 and the proposed posterior sampling algorithm in (16), which incorporates the variance-reduction mechanism in (15) for the noisy-gradient setting described above. We refer to this method as **NG-APMC** ("*Noisy-Gradient APMC*"). We present toy experiments that provide numerical and statistical validation of the convergence of NG-APMC in Fisher information and demonstrate the benefit of its variance-reduction mechanism by comparing to the naive NG-APMC approach ($p = 1$).

### A.2.1 NUMERICAL VALIDATION OF NG-APMC

Similar to the numerical experiments in Section 4.1, we conside a synthetic bimodal 2D Gaussian-mixture prior, a randomly chosen forward operator $\boldsymbol{A}$, and additive measurement noise $\xi \sim \mathcal{N}(0, I)$.

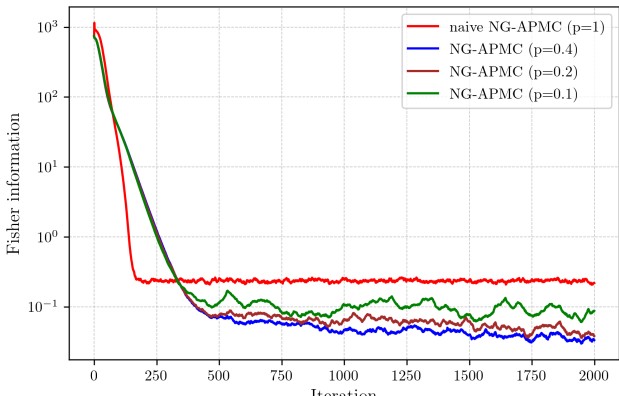

Figure 6: Convergence behavior of the proposed NG-APMC posterior sampling algorithm, evaluated using the Fisher information metric, for different parameter settings $(p, b) \in \{(1.0, 6), (0.4, 15), (0.2, 30), (0.1, 60)\}$. Despite operating under a fixed per-iteration computational budget, $p \ll 1$ and large $b$ yield better convergence, highlighting the effectiveness of the proposed estimator in (15).

We compute the analytical score of the prior and perturb it with bounded Gaussian noise (standard deviation is $\varepsilon_{k^\star} = 2.5 \geq \varepsilon_{\sigma_k}$) to simulate the generalization error of the SGM (i.e. $\mathcal{S}_\theta$). We add noise to the likelihood score to simulate instability in the forward operator (as discussed in the previous section) with standard deviation $\sigma_{\mathrm{noise}}$. From the discussion after Proposition 1, one should keep the batch size $b = O(\sigma_{\mathrm{noise}}^2)$ when $p = 1$ for score likelihood estimation to converge to 0. We note that although the discussion in that section is specific to ZO estimate and says $b = O(d)$, one can easily derive the similar argument for the noisy gradient setting in (12). Thus, we select $p = 1$ as a baseline and refer to it as "naive NG-APMC" approach in the results. Furthermore, Theorem 3 implies the variance term $\sigma_{\mathrm{noise}}^2$ in the FI upper bound can be reduced by using a large batch size $b$ while choosing a small $p \ll 1$, which lowers the expected number of gradient evaluations to roughly $pb$. To illustrate this effect, we fix a small per-iteration budget of 6 with large noise level of $\sigma_{\mathrm{noise}} = 10$. We run NG-APMC for different values of $(p, b) \in \{(1.0, 6), (0.4, 15), (0.2, 30), (0.1, 60)\}$ so that the average computational cost per iteration remains the same. We run 20 experiments for each pair with different random seeds and report the mean of the results. We refer the reader to Appendix A.4 for more information about FI calculation and annealing schedules. From Fig. 6, we observe that the second term in Theorem 3, corresponding to the variance component, becomes the dominant bottleneck and prevents convergence to small $\varepsilon$. When the NG-APMC algorithm employs the variance reduction mechanism (i.e., $p \ll 1$), this bottleneck is mitigated, allowing the Fisher information to converge to smaller $\varepsilon$ values. Additionally, when we set $p = 0.1$, the convergence becomes noticeably more unstable. This behavior agrees with Theorem 3: as $p$ decreases, the discretization error grows (amplified by the smoothness of the log-likelihood) and can become the dominant source of error when $p$ is reduced too aggressively.

### A.2.2 STATISTICAL VALIDATION OF NG-APMC

In addition to the numerical experiments, we demonstrate that the proposed NG-APMC algorithm estimates the posterior mean and variance statistics accurately. We note that, similar to Corollary 1, the convergence of NG-APMC in squared total variation distance can also be established under a Poincaré inequality assumption. For a linear forward model $\boldsymbol{A}$ with Gaussian noise, this assumption is satisfied, which implies that the sample statistics produced by NG-APMC converge to those of the target posterior under noisy unstable forward model gradient calculations. We demonstrate this by generating synthetic prior distribution from CelebA (Liu et al., 2018) with random forward operator $\boldsymbol{A} \in \mathbb{R}^{115 \times 1024}$. The details about the synthetic prior, prior score estimation, and annealing schedule are explained in Appendix A.4. We set the noise level to $\sigma_{\mathrm{noise}} = 15.0$ and run NG-APMC with per-iteration budget $pb = 2$ for different values of $(p, b) \in \{(1.0, 2), (0.4, 5), (0.2, 10), (0.1, 20)\}$.

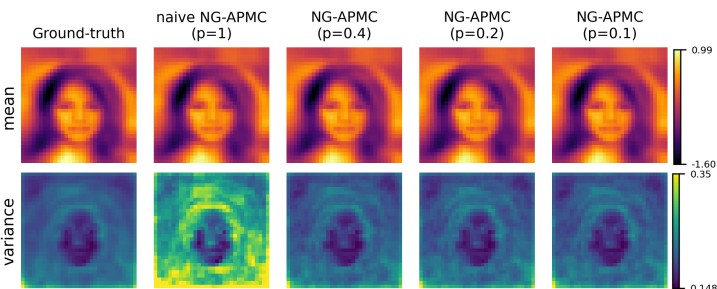

Figure 7: Comparison of the naive NG-APMC ($p = 1$) with NG-APMC using $p \ll 1$, showing that the proposed method estimates the posterior mean accurately and achieves low, accurate variance when $p$ is small. This highlights the benefit of its variance-reduction mechanism.

Fig. 7 shows that the proposed NG-APMC estimates the analytical posterior mean accurately under noisy likelihood scores. Moreover, when the naive approach, $p = 1$, is used, the generated samples have very large variance. If we choose $p < 1$, this variance reduces significantly and the posterior variance is estimated accurately.

## A.3  Proofs

**Notation.** Throughout the proof, we work within the probability space $(\Omega, \mathcal{F}, \mathbb{P})$, where $\Omega$ denotes the sample space, $\mathcal{F}$ the $\sigma$-algebra, and $\mathbb{P}$ the probability measure. For a random variable $X : \Omega \to \mathbb{R}^n$, we write its expectation as

$$\mathbb{E}[X] = \int_{\Omega} \zeta(\omega)\, \mathbb{P}(d\omega).$$

The posterior distribution of interest is of the form

$$\pi(\boldsymbol{x}|\boldsymbol{y}) \propto \ell(\boldsymbol{y}|\boldsymbol{x})p(\boldsymbol{x}),$$

where we define $f(\boldsymbol{x}) = -\log \ell(\boldsymbol{y}|\boldsymbol{x})$ and $h(\boldsymbol{x}) = -\log p(\boldsymbol{x})$. Moreover, the gradient of the perturbed log-prior is denoted by $\nabla h_{\sigma_k}(\boldsymbol{x}) := -\nabla \log p_{\sigma_k}(\boldsymbol{x})$. For simplicity, we omit the explicit dependence on $\boldsymbol{y}$. Recall that

$$-\nabla \log \pi(\boldsymbol{x}) = \nabla f(\boldsymbol{x}) + \nabla h(\boldsymbol{x}). \tag{18}$$

We denote the zeroth-order approximation of the forward model gradient as follows

$$\tilde{\nabla} f_\mu(\boldsymbol{x}_{k\gamma}, \boldsymbol{u}) := \frac{f(\boldsymbol{x}_{k\gamma} + \mu\boldsymbol{u}) - f(\boldsymbol{x}_{k\gamma})}{\mu}\boldsymbol{u}, \tag{19}$$

where $\boldsymbol{u} \sim \mathcal{N}(0, I)$ and $\mu > 0$. The expectation of the zeroth-order approximation is denoted as $\nabla f_\mu(\boldsymbol{x}_{k\gamma}) := \mathbb{E}_{\boldsymbol{u}}\left[ \tilde{\nabla} f_\mu(\boldsymbol{x}_{k\gamma}, \boldsymbol{u}) \right]$. For notational convenience, we also define

$$\Delta_k := \mathbb{E}\left[\|\boldsymbol{x}_{(k+1)\gamma} - \boldsymbol{x}_{k\gamma}\|^2\right], \tag{20}$$

as the expected squared $\ell_2$-distance between consecutive iterates. For convenience, we recall the definition of the Kullback–Leibler (KL) divergence between two probability densities $\nu$ and $\pi$:

$$\mathrm{KL}(\nu\|\pi) = \int_{\mathbb{R}^n} \nu(x) \log \frac{\nu(x)}{\pi(x)}\, dx.$$

The Fisher information (FI) is given by

$$\mathrm{FI}(\nu\|\pi) = \int_{\mathbb{R}^n} \left\|\nabla \log \frac{\nu(x)}{\pi(x)}\right\|_2^2 \nu(x)\, dx = \int_{\mathbb{R}^n} \|\nabla \log \nu(x) - \nabla \log \pi(x)\|_2^2 \nu(x)\, dx.$$

The Total Variation (TV) distance between two probability measures $\mu$ and $\nu$ on a measurable space $(\mathcal{X}, \mathcal{F})$ is given b by

$$\|\mu - \nu\|_{\mathrm{TV}} := \sup_{A \in \mathcal{F}} |\mu(A) - \nu(A)| = \frac{1}{2}\int_{\mathcal{X}} |d\mu - d\nu|.$$

Unless otherwise stated, $\|\cdot\|^2$ denotes the squared $\ell_2$-norm, i.e. $\|\cdot\|_2^2$.

### A.3.1 LEMMAS

We begin by reviewing the key lemmas from the zeroth-order optimization and non-log-concave sampling literature. The following section summarizes the fundamental properties of zeroth-order approximations that will be used in our analysis.

**Lemma 1 ((Lan, 2020))** *Suppose that $f(\boldsymbol{x}) \in C_L^{1,1}$, and let $f_\mu(\boldsymbol{x}) := \mathbb{E}_{\boldsymbol{u}}[f(\boldsymbol{x} + \mu \boldsymbol{u})]$. Then the following statements hold:*

*(a) $f_\mu \in C_\mu^{1,1}(\mathbb{R}^d)$, where $L_\mu \leq L$,*

*(b) $\|\nabla f_\mu(\boldsymbol{x}) - \nabla f(\boldsymbol{x})\| \leq \frac{1}{2}\mu L(d+3)^{\frac{3}{2}}$,*

*(c) $\mathbb{E}_{\boldsymbol{u}}[\|\tilde{\nabla} f_\mu(\boldsymbol{x}, \boldsymbol{u})\|^2] \leq \frac{1}{2}\mu^2 L^2(d+6)^3 + 2(d+4)\|\nabla f(\boldsymbol{x})\|_2^2$,*

*where $\tilde{\nabla} f_\mu(\boldsymbol{x}, \boldsymbol{u}) := \frac{f(\boldsymbol{x} + \mu \boldsymbol{u}) - f(\boldsymbol{x})}{\mu} \boldsymbol{u}$ for $\boldsymbol{u} \sim \mathcal{N}(\boldsymbol{0}, \boldsymbol{I}_d)$ and any $\boldsymbol{x} \in \mathbb{R}^d$, $\mu > 0$.*

The following lemma concerns the density evolution of an interpolated diffusion process.

**Lemma 2 ((Balasubramanian et al., 2022))** *Consider the stochastic process defined by*

$$\boldsymbol{x}_t := \boldsymbol{x}_0 - t g_0 + \sqrt{2} \boldsymbol{B}_t, \quad \text{with } g_0 = g(\boldsymbol{x}_0), \quad \boldsymbol{x}_0 \sim \nu_0 \tag{21}$$

*where $g_0$ is integrable and $\{B_t\}_{t \geq 0}$ is a standard Brownian motion in $\mathbb{R}^d$ independent of $(\boldsymbol{x}_0, g_0)$. Then, writing $\nu_t$ for the probability density of $\boldsymbol{x}_t$, we have*

$$\frac{d}{dt} \mathrm{KL}(\nu_t \| \pi) \leq -\frac{3}{4} \mathrm{FI}(\nu_t \| \pi) + \mathbb{E}\left[\|\nabla f(\boldsymbol{x}_t) - g_0\|^2\right], \tag{22}$$

*where we recall that $\pi \propto e^{-f}$, and the expectation in the last term is with respect to $x_0 \sim \nu_0$ and $x_t \sim \nu_t$.*

We also used the following lemma to bound the Fisher information, which is taken from (Chewi et al., 2024).

**Lemma 3 ((Chewi et al., 2024))** *Assume that $\nabla \log \pi(\boldsymbol{x})$ is $L_\pi$-Lipschitz. For any probability measure $\nu$, it holds that*

$$\mathbb{E}_\mu\left[\|\nabla \log \pi(\boldsymbol{x})\|^2\right] \leq \mathrm{FI}(\nu \| \pi) + 2d L_\pi. \tag{23}$$

We use the following lemma to derive an upper bound on Total Variation (TV) distance.

**Lemma 4 ((Guillin et al., 2009))** *If $\pi$ satisfies a Poincaré inequality, i.e. for every smooth, compactly supported $f : \mathbb{R}^d \to \mathbb{R}$,*

$$\mathrm{Var}_\pi(f) \leq C_{\mathrm{PI}} \mathbb{E}_\pi[\|\nabla f\|^2],$$

*then for any probability measure $\mu$,*

$$\|\mu - \pi\|_{\mathrm{TV}}^2 \leq 4 C_{\mathrm{PI}} \mathrm{FI}(\mu \| \pi).$$

### A.3.2 PROOF OF PROPOSITION 1

For simplicity, let our PAGE estimator be defined as

$$\boldsymbol{g}_k := \begin{cases} \dfrac{1}{b} \sum_{i \in I} \tilde{\nabla} f_\mu(\boldsymbol{x}_k, \boldsymbol{u}_i), & \boldsymbol{B}_k = \mathrm{ref}, \\[2mm] \boldsymbol{g}_{k-1} + \dfrac{1}{b'} \sum_{i \in I'} \left(\tilde{\nabla} f_\mu(\boldsymbol{x}_k, \boldsymbol{u}_i) - \tilde{\nabla} f_\mu(\boldsymbol{x}_{k-1}, \boldsymbol{u}_i)\right), & \boldsymbol{B}_k = \mathrm{corr}, \end{cases} \tag{24}$$

where "ref" and "corr" denote the "refresh" and "correction" branches of the estimate and $\boldsymbol{B}_k \in \{\mathrm{corr}, \mathrm{ref}\}$ is a random variable such that $\mathbb{P}(\boldsymbol{B}_k = \mathrm{ref}) = p$ and $\mathbb{P}(\boldsymbol{B}_k = \mathrm{corr}) = 1 - p$. Additionally, define the mini-batch estimators as

$$\tilde{\boldsymbol{v}}_b(\boldsymbol{x}_k) = \frac{1}{b} \sum_{i \in I} \tilde{\nabla} f_\mu(\boldsymbol{x}_k, \boldsymbol{u}_i) \quad \text{and} \quad \boldsymbol{\delta}_k = \frac{1}{b'} \sum_{i \in I'} \left(\tilde{\nabla} f_\mu(\boldsymbol{x}_k, \boldsymbol{u}_i) - \tilde{\nabla} f_\mu(\boldsymbol{x}_{k-1}, \boldsymbol{u}_i)\right), \tag{25}$$

where $|I| = b$ and $|I'| = b'$. Then, $\boldsymbol{g}_k$ can be written as

$$\boldsymbol{g}_k := \begin{cases} \tilde{\boldsymbol{v}}_b(\boldsymbol{x}_k), & \boldsymbol{B}_k = \text{ref}, \\ \boldsymbol{g}_{k-1} + \boldsymbol{\delta}_k, & \boldsymbol{B}_k = \text{corr}. \end{cases} \tag{26}$$

Let $\mathcal{F}_k := \sigma(\boldsymbol{x}_0, \boldsymbol{B}_1, \boldsymbol{Z}_1, \boldsymbol{u}_1, \ldots, \boldsymbol{x}_k, \boldsymbol{B}_k, \boldsymbol{Z}_k, \boldsymbol{u}_k)$ be the sigma-algebra generated by all the random variables revealed up to the end of iteration $k$. From (6), recall that $\boldsymbol{Z}_k$ is due to the discretization of the Langevin diffusion. Then, conditioning on the history $\mathcal{F}_{k-1}$ and on $\boldsymbol{x}_k$, we have

$$\mathbb{E}[\tilde{\boldsymbol{v}}_b(\boldsymbol{x}_k)|\mathcal{F}_{k-1}, \boldsymbol{x}_k] = \nabla f_\mu(\boldsymbol{x}_k), \quad \mathbb{E}[\boldsymbol{\delta}_k|\mathcal{F}_{k-1}, \boldsymbol{x}_k] = \nabla f_\mu(\boldsymbol{x}_k) - \nabla f_\mu(\boldsymbol{x}_{k-1}). \tag{27}$$

Then, we have $\mathbb{E}[\boldsymbol{g}_k|\mathcal{F}_{k-1}, \boldsymbol{x}_k] = \nabla f_\mu(\boldsymbol{x}_k)$. Using this property inductively, we can obtain $\mathbb{E}[\boldsymbol{g}_k] = \mathbb{E}[\nabla f_\mu(\boldsymbol{x}_k)]$ after taking the expectation of both sides. Therefore, if we define the error as $\boldsymbol{e}_k := \boldsymbol{g}_k - \nabla f_\mu(\boldsymbol{x}_k)$, then $\mathbb{E}[\boldsymbol{e}_k] = 0$, which implies that $\boldsymbol{g}_k$ is unbiased estimate of $\nabla f_\mu(\boldsymbol{x}_k)$. Let's consider the error propagation at each branch separately.

For the correction branch, assume $\boldsymbol{B}_k = \text{corr}$ is true. We can define zero-mean fluctuation at step $k$ as

$$\tilde{\boldsymbol{\delta}}_k := \boldsymbol{\delta}_k - \mathbb{E}[\boldsymbol{\delta}_k|\mathcal{F}_{k-1}, \boldsymbol{x}_k], \tag{28}$$

where $\mathbb{E}[\tilde{\boldsymbol{\delta}}_k|\mathcal{F}_{k-1}|\boldsymbol{x}_k] = 0$. Then,

$$\boldsymbol{e}_k = \boldsymbol{g}_k - \nabla f_\mu(\boldsymbol{x}_k) = \underbrace{\boldsymbol{g}_{k-1} - \nabla f_\mu(\boldsymbol{x}_{k-1})}_{\boldsymbol{e}_{k-1}} + \boldsymbol{\delta}_k - (\nabla f_\mu(\boldsymbol{x}_k) - \nabla f_\mu(\boldsymbol{x}_{k-1}))$$

$$= \boldsymbol{e}_{k-1} + \tilde{\boldsymbol{\delta}}_k. \tag{29}$$

Note that $\boldsymbol{u}_i \sim \mathcal{N}(0, I)$ random vectors selected at step $k$ are independent of the ones selected at step $k-1$. Therefore, $\tilde{\boldsymbol{\delta}}_k$ is conditionally independent of $\boldsymbol{e}_{k-1}$, so we can write

$$\text{Cov}(\boldsymbol{e}_k|\mathcal{F}_{k-1}, \boldsymbol{x}_k, \boldsymbol{B}_k = \text{corr}) = \text{Cov}(\boldsymbol{e}_{k-1}|\mathcal{F}_{k-1}) + \text{Cov}(\tilde{\boldsymbol{\delta}}_k|\mathcal{F}_{k-1}, \boldsymbol{x}_k). \tag{30}$$

In the refresh branch, we have $\boldsymbol{B}_k = \text{ref}$. Then, the error term can be written as $\boldsymbol{e}_k = \tilde{\boldsymbol{v}}_b(\boldsymbol{x}_k) - \nabla f_\mu(\boldsymbol{x}_k)$ and its covariance is

$$\text{Cov}(\boldsymbol{e}_k|\mathcal{F}_{k-1}, \boldsymbol{x}_k, \boldsymbol{B}_k = \text{ref}) = \Sigma_b(\boldsymbol{x}_k), \tag{31}$$

where $\Sigma_b(\boldsymbol{x}_k) := \frac{1}{b}\text{Cov}(\tilde{\nabla} f_\mu(\boldsymbol{x}_k, \boldsymbol{u}_i))$. Furthermore, using the definition of $\boldsymbol{e}_k$, we have

$$\mathbb{E}[\boldsymbol{e}_k|\mathcal{F}_{k-1}, \boldsymbol{x}_k, \boldsymbol{B}_k = \text{corr}] = \boldsymbol{e}_{k-1} \quad \text{and} \quad \mathbb{E}[\boldsymbol{e}_k|\mathcal{F}_{k-1}, \boldsymbol{x}_k, \boldsymbol{B}_k = \text{ref}] = 0. \tag{32}$$

Using the law of total variance, we can write the covariance matrix conditioned on the history $\mathcal{F}_{k-1}$ and $\boldsymbol{x}_k$ as

$$\text{Cov}(\boldsymbol{e}_k|\mathcal{F}_{k-1}, \boldsymbol{x}_k) = \mathbb{E}[\text{Cov}(\boldsymbol{e}_k|\mathcal{F}_{k-1}, \boldsymbol{x}_k, \boldsymbol{B}_k)] + \text{Cov}(\mathbb{E}[\boldsymbol{e}_k|\mathcal{F}_{k-1}, \boldsymbol{x}_k, \boldsymbol{B}_k]). \tag{33}$$

If we plug (30) and (31) with the conditional means in (32), we get

$$\text{Cov}(\boldsymbol{e}_k|\mathcal{F}_{k-1}, \boldsymbol{x}_k) = (1-p)\left(\text{Cov}(\boldsymbol{e}_{k-1}|\mathcal{F}_{k-1}) + \text{Cov}(\tilde{\boldsymbol{\delta}}_k|\mathcal{F}_{k-1}, \boldsymbol{x}_k)\right) + p\Sigma_b(\boldsymbol{x}_k) + p(1-p)\boldsymbol{e}_{k-1}\boldsymbol{e}_{k-1}^T. \tag{34}$$

Taking the expectation of both sides, we get

$$\text{Cov}(\boldsymbol{e}_k) = (1 - p^2)\text{Cov}(\boldsymbol{e}_{k-1}) + (1 - p)\mathbb{E}[\text{Cov}(\tilde{\boldsymbol{\delta}}_k)] + p\mathbb{E}[\Sigma_b(\boldsymbol{x}_k)]. \tag{35}$$

The factor $(1 - p^2)$ is the contraction on the previous error covariance in expectation. Note that $\boldsymbol{u}_i$ are i.i.d. and recalling the definition of $\tilde{\boldsymbol{\delta}}_k$, we have

$$\text{Cov}(\tilde{\boldsymbol{\delta}}_k) \preceq \text{Cov}(\boldsymbol{\delta}_k) \preceq \frac{4dL_{f_2}^2}{b'\mu^2}\|\boldsymbol{x}_k - \boldsymbol{x}_{k-1}\|^2 I, \tag{36}$$

where we use the Assumption 1 to get the second inequality. This shows that the correction-step noise is small when the iterate moves only a little between steps. Taking the trace of both sides in (35) and plugging (36), we get

$$\text{Cov}(\boldsymbol{e}_k) \preceq (1 - p^2)\text{Cov}(\boldsymbol{e}_{k-1}) + \frac{4d(1-p)L_{f_1}^2}{b'\mu^2}\mathbb{E}[\|\boldsymbol{x}_k - \boldsymbol{x}_{k-1}\|^2]I + p\mathbb{E}[\Sigma_b(\boldsymbol{x}_k)]. \tag{37}$$

Equivalently, this can be written as

$$\mathbb{E}[\|\boldsymbol{e}_k\|^2] \leq (1 - p^2)\mathbb{E}[\|\boldsymbol{e}_{k-1}\|^2] + \frac{4d(1-p)L_{f_1}^2}{b'\mu^2}\mathbb{E}[\|\boldsymbol{x}_k - \boldsymbol{x}_{k-1}\|^2] + \frac{p}{b}\sigma^2, \tag{38}$$

where $\sigma_g^2 \leq \sigma^2$ and $\sigma_g^2(\boldsymbol{x}) := \text{Tr}\left(\text{Cov}(\tilde{\nabla} f_\mu(\boldsymbol{x}, \boldsymbol{u}_i))\right)$.

### A.3.3 PROOF OF THEOREM 1

We can construct the following interpolation for ZO-APMC

$$\boldsymbol{x}_t = \boldsymbol{x}_{k\gamma} - (t - k\gamma)\mathcal{G}(\boldsymbol{x}_{k\gamma}) + \sqrt{2}(\boldsymbol{B}_t - \boldsymbol{B}_{k\gamma}) \quad \text{for} \quad t \in [k\gamma, (k+1)\gamma] \tag{39}$$

where $\mathcal{G}(\boldsymbol{x}_{k\gamma}) = \boldsymbol{g}_k - \alpha_k \mathcal{S}_\theta(\boldsymbol{x}_{k\gamma})$, $\boldsymbol{g}_k$ is an estimate of the forward model gradient with zeroth-order approximation and variance-reduction mechanism, $\alpha_k$ and $\sigma_k$ are annealing parameters. By Assumption 2 and 4 and triangle inequality, we know that the target posterior score function $\nabla \log \pi(\boldsymbol{x})$ is Lipschitz continuous with Lipschitz constant $L_\pi = L_p + L_{f_2}$. Furthermore, by Assumptions 3 and 5, the error between the prior score function and the score estimate scaled by annealing parameter can be bounded by

$$\|\nabla h(\boldsymbol{x}_{k\gamma}) + \alpha_k \mathcal{S}_\theta(\boldsymbol{x}_{k\gamma})\| \le \sigma_k C + \varepsilon_{\sigma_k} + (\alpha_k - 1)\sigma_k^{-1} C \tag{40}$$

where we recall $\nabla h(\boldsymbol{x}_{k\gamma}) = -\nabla \log p(\boldsymbol{x}_{k\gamma})$. Note that we add and subtract $\nabla h_{\sigma_k}(\boldsymbol{x}_{k\gamma})$ and use triangle inequality. Now we can provide the proof for Theorem 1. Combining Lemma 2 with the interpolation argument in (39), it follows that for every $t \in [k\gamma, (k+1)\gamma]$,

$$\frac{d}{dt}\mathrm{KL}(\nu_t\|\pi) \le -\frac{3}{4}\mathrm{FI}(\nu_t\|\pi) + \mathbb{E}\left[\|\nabla \log \pi(\boldsymbol{x}_t) + \boldsymbol{g}_k - \alpha_k \mathcal{S}_\theta(\boldsymbol{x}_{k\gamma}, \sigma_k)\|^2\right]. \tag{41}$$

Adding and subtracting the following values $\nabla f_\mu(\boldsymbol{x}_t), \nabla f_\mu(\boldsymbol{x}_{k\gamma}), \nabla h(\boldsymbol{x}_{k\gamma}), \nabla h_{\sigma_k}(\boldsymbol{x}_{k\gamma})$ inside the expectation and using the convexity of $\ell_2$ norm with the upper bound in (40), we get

$$\frac{d}{dt}\mathrm{KL}(\nu_t\|\pi) \le -\frac{3}{4}\mathrm{FI}(\nu_t\|\pi) + 4\mathbb{E}\left[\|\boldsymbol{g}_k - \nabla f_\mu(\boldsymbol{x}_{k\gamma})\|^2\right] + 4L_\pi^2 \mathbb{E}\left[\|\boldsymbol{x}_t - \boldsymbol{x}_{k\gamma}\|^2\right]$$
$$+ \mu^2 L_{f_2}^2 (d+3)^3 + 4(\sigma_k C + \varepsilon_{\sigma_k} + (\alpha_k - 1)\sigma_k^{-1} C)^2. \tag{42}$$

Let $e_k^2 := \mathbb{E}\left[\|\boldsymbol{g}_k - \nabla f_\mu(\mathbf{x}_{k\gamma})\|^2\right]$, which quantifies the squared error between the zeroth-order estimate $\boldsymbol{g}_k$ and the true score $\nabla f_\mu(\mathbf{x}_{k\gamma})$ of the $\mu$-perturbed forward model. Here the expectation is taken with respect to both the randomness of the zeroth-order approximation and the measure $\mathcal{F}_k$ associated with the data $\boldsymbol{x}_{k\gamma}$. Note that the bias term due to the zeroth-order approximation appears as the fourth term of the previous inequality. Using the definition of $\boldsymbol{g}_k$, we can expand the error term as

$$e_{k+1}^2 = p\,\mathbb{E}\left[\left\|\nabla f_\mu(\boldsymbol{x}_{(k+1)\gamma}) - \frac{1}{b}\sum_{i=1}^{b}\tilde{\nabla} f_\mu(\boldsymbol{x}_{(k+1)\gamma}, \boldsymbol{u}_i)\right\|^2\right]$$

$$+ (1-p)\,\mathbb{E}\left[\left\|\nabla f_\mu(\boldsymbol{x}_{(k+1)\gamma}) - \boldsymbol{g}_k - \frac{1}{b'}\sum_{i=1}^{b'}\left(\tilde{\nabla} f_\mu(\boldsymbol{x}_{(k+1)\gamma}, \boldsymbol{u}_i) - \tilde{\nabla} f_\mu(\boldsymbol{x}_{k\gamma}, \boldsymbol{u}_i)\right)\right\|^2\right] \tag{43}$$

where $b'$, $b$ denote the small and large batch sizes, respectively, and $\boldsymbol{u}_i \sim \mathcal{N}(0, I)$ in $\mathbb{R}^d$. We can upper bound the first expectation as

$$\mathbb{E}\left[\left\|\nabla f_\mu(\boldsymbol{x}_{(k+1)\gamma}) - \frac{1}{b}\sum_{i=1}^{b}\tilde{\nabla} f_\mu(\boldsymbol{x}_{(k+1)\gamma}, \boldsymbol{u}_i)\right\|^2\right] \le \frac{1}{b}\mathbb{E}\left[\left\|\nabla f_\mu(\boldsymbol{x}_{(k+1)\gamma}) - \tilde{\nabla} f_\mu(\boldsymbol{x}_{(k+1)\gamma}, \boldsymbol{u}_i)\right\|^2\right] \tag{44}$$

$$\le \frac{1}{b}\mathbb{E}\left[\left\|\tilde{\nabla} f_\mu(\boldsymbol{x}_{(k+1)\gamma}, \boldsymbol{u}_i)\right\|^2\right] \tag{45}$$

$$\le \frac{L_{f_1}^2}{b}\mathbb{E}\left[\|\boldsymbol{u}_i\|^4\right] \tag{46}$$

$$= \frac{d(d+2)L_{f_1}^2}{b}. \tag{47}$$

In (44), we use the fact that the random variables $\boldsymbol{u}_i$ are i.i.d. In (45), we use the second-moment bound on the variance. Finally, in (46), we use the zeroth-order definition of $\tilde{\nabla} f_\mu(\boldsymbol{x}_{k\gamma}, \boldsymbol{u}_i)$ with the

Assumption 1 and evaluate the expectation under $\boldsymbol{u}_i \sim \mathcal{N}(0, I)$ to get (47). Plugging this upper bound into (43), we get

$$
e_{k+1}^2 \leq \frac{pd(d+2)L_{f_1}^2}{b} + (1-p)\mathbb{E}\left[\left\|\nabla f_\mu(\boldsymbol{x}_{(k+1)\gamma}) - g_k\right.\right.
$$

$$
\left.\left. - \frac{1}{b'}\sum_{i=1}^{b'}\left(\tilde{\nabla}f_\mu(\boldsymbol{x}_{(k+1)\gamma}, \boldsymbol{u}_i) - \tilde{\nabla}f_\mu(\boldsymbol{x}_{k\gamma}, \boldsymbol{u}_i)\right)\right\|^2\right] \quad (48)
$$

$$
= \frac{pd(d+2)L_{f_1}^2}{b} + (1-p)\mathbb{E}\left[\left\|\nabla f_\mu(\boldsymbol{x}_{k\gamma}) - g_k + \nabla f_\mu(\boldsymbol{x}_{(k+1)\gamma}) - \nabla f_\mu(\boldsymbol{x}_{k\gamma})\right.\right.
$$

$$
\left.\left. - \frac{1}{b'}\sum_{i=1}^{b'}\left(\tilde{\nabla}f_\mu(\boldsymbol{x}_{(k+1)\gamma}, \boldsymbol{u}_i) - \tilde{\nabla}f_\mu(\boldsymbol{x}_{k\gamma}, \boldsymbol{u}_i)\right)\right\|^2\right] \quad (49)
$$

$$
= \frac{pd(d+2)L_{f_1}^2}{b} + (1-p)e_k^2 + \frac{1}{b'}\mathbb{E}\left[\left\|\nabla f_\mu(\boldsymbol{x}_{(k+1)\gamma}) - \nabla f_\mu(\boldsymbol{x}_{k\gamma})\right.\right.
$$

$$
\left.\left. - \left(\tilde{\nabla}f_\mu(\boldsymbol{x}_{(k+1)\gamma}, \boldsymbol{u}_i) - \tilde{\nabla}f_\mu(\boldsymbol{x}_{k\gamma}, \boldsymbol{u}_i)\right)\right\|^2\right]
$$

$$
\quad (50)
$$

$$
\leq \frac{pd(d+2)L_{f_1}^2}{b} + (1-p)e_k^2 + \frac{1}{b'}\mathbb{E}\left[\left\|\tilde{\nabla}f_\mu(\boldsymbol{x}_{(k+1)\gamma}, \boldsymbol{u}_i) - \tilde{\nabla}f_\mu(\boldsymbol{x}_{k\gamma}, \boldsymbol{u}_i)\right\|^2\right] \quad (51)
$$

$$
= \frac{pd(d+2)L_{f_1}^2}{b} + (1-p)e_k^2 + \frac{1}{\mu^2 b'}\mathbb{E}\left[\left\|\left(f(\boldsymbol{x}_{(k+1)\gamma} + \mu\boldsymbol{u}_i) - f(\boldsymbol{x}_{k\gamma} + \mu\boldsymbol{u}_i)\right)\right.\right.
$$

$$
\left.\left. - \left(f(\boldsymbol{x}_{(k+1)\gamma}) - f(\boldsymbol{x}_{k\gamma})\right)\right\|^2 \|\boldsymbol{u}_i\|^2\right]
$$

$$
\quad (52)
$$

$$
\leq \frac{pd(d+2)L_{f_1}^2}{b} + (1-p)e_k^2 + \frac{4L_{f_1}^2\Delta_k}{\mu^2 b'}\mathbb{E}\left[\|\boldsymbol{u}_i\|^2\right] \quad (53)
$$

$$
= \frac{pd(d+2)L_{f_1}^2}{b} + (1-p)e_k^2 + \frac{4dL_{f_1}^2\Delta_k}{\mu^2 b'} \quad (54)
$$

where $\Delta_k := \mathbb{E}\left[\|\boldsymbol{x}_{(k+1)\gamma} - \boldsymbol{x}_{k\gamma}\|^2\right]$. Note that we add and subtract $\nabla f_\mu(\boldsymbol{x}_{k\gamma})$ in (49). To get (50), we use the fact that random variables $\boldsymbol{u}_i \sim \mathcal{N}(0, I)$ are i.i.d., and calculate conditional expectation conditioned with respect to $\mathcal{F}_k$ and then use the definition of $e_k^2$. We use second-moment bound on variance in (51) and use the zeroth-order definition to get (52). Following that, we first apply Assumption 1 and then exploit the independence between $\boldsymbol{x}_{k\gamma}$, $\boldsymbol{x}_{(k+1)\gamma}$, and $\boldsymbol{u}_i$ to obtain (53). Dividing both sides by $p$ and rearranging the terms, we get an upper bound on the error term

$$
e_k^2 \leq \frac{d(d+2)L_{f_1}^2}{b} + \left(\frac{1-p}{p}\right)\frac{4dL_{f_1}^2}{\mu^2 b'}\Delta_k - \frac{1}{p}(e_{k+1}^2 - e_k^2) \quad (55)
$$

Plugging this upper bound into (42), we get

$$
\frac{d}{dt}\mathrm{KL}(\nu_t\|\pi) \leq -\frac{3}{4}\mathrm{FI}(\nu_t\|\pi) + 4L_\pi^2\mathbb{E}\left[\|\boldsymbol{x}_t - \boldsymbol{x}_{k\gamma}\|^2\right] + \mu^2 L_{f_2}^2(d+3)^2 + \frac{4d(d+2)L_{f_1}^2}{b}
$$

$$
+ \left(\frac{1-p}{p}\right)\frac{16dL_{f_1}^2}{\mu^2 b'}\mathbb{E}\left[\|\boldsymbol{x}_{(k+1)\gamma} - \boldsymbol{x}_{k\gamma}\|^2\right] - \frac{4}{p}(e_{k+1}^2 - e_k^2)
$$

$$
+ 4(\sigma_k C + \varepsilon_{\sigma_k} + (\alpha_k - 1)\sigma_k^{-1}C)^2 \quad (56)
$$

By the interpolation argument in (39), we have

$$\mathbb{E}[\|\boldsymbol{x}_t - \boldsymbol{x}_{k\gamma}\|^2] = (t - k\gamma)^2 \mathbb{E}\left[\|\mathcal{G}(x_{k\gamma})\|^2\right] + 2(t - k\gamma)d \tag{57}$$

$$\leq \gamma^2 \mathbb{E}\left[\|\mathcal{G}(x_{k\gamma})\|^2\right] + 2\gamma d \tag{58}$$

$$= \mathbb{E}[\|\boldsymbol{x}_{(k+1)\gamma} - \boldsymbol{x}_{k\gamma}\|^2] \tag{59}$$

for $t \in [k\gamma, (k+1)\gamma]$ because dimensionality of vectors $d > 0$ and $\mathbb{E}\left[\|\mathcal{G}(x_{k\gamma})\|^2\right] \geq 0$. We use the bound on (59) in (56) and get

$$\frac{d}{dt}\mathrm{KL}(\nu_t\|\pi) \leq -\frac{3}{4}\mathrm{FI}(\nu_t\|\pi) + \mu^2 L_{f_2}^2(d+3)^2 + \frac{4d(d+2)L_{f_1}^2}{b} - \frac{4}{p}(e_{k+1}^2 - e_k^2)$$

$$+ 4L_m^2\left[1 + \left(\frac{1-p}{p}\right)\frac{4d}{\mu^2 b'}\right]\Delta_k + 4(\sigma_k C + \varepsilon_{\sigma_k} + (\alpha_k - 1)\sigma_k^{-1}C)^2 \tag{60}$$

where $L_m := \max\{L_{f_1}, L_\pi\}$. We use the interpolation argument given in (39) to put an upper bound on $\Delta_k$ as

$$\Delta_k = \gamma^2 \mathbb{E}[\|\boldsymbol{g}_k - \alpha_k \mathcal{S}_\theta(\boldsymbol{x}_{k\gamma}, \sigma_k)\|^2] + 2\gamma d \tag{61}$$

$$\leq 5\gamma^2 e_k^2 + \frac{5\gamma^2\mu^2 L_{f_2}^2(d+3)^3}{4} + 5\gamma^2 L_\pi^2 \Delta_k + 5\gamma^2 \mathbb{E}\left[\|\nabla \log \pi(\boldsymbol{x}_t)\|^2\right]$$

$$+ 5\gamma^2(\sigma_k C + \varepsilon_{\sigma_k} + (\alpha_k - 1)\sigma_k^{-1}C)^2 + 2\gamma d \tag{62}$$

where we add and subtract $\nabla f_\mu(\mathbf{x}_{k\gamma})$, $\nabla f(\mathbf{x}_{k\gamma})$, $\nabla f(\mathbf{x}_t)$, $\nabla h(\mathbf{x}_t)$, $\nabla h(\mathbf{x}_{k\gamma})$, and $\nabla h_{\sigma_k}(\mathbf{x}_{k\gamma})$ inside the expectation, and then apply the convexity of the $\ell_2$ norm together with the part (b) of Assumption 5 to obtain (62). Using the bound on $e_k^2$ in (55), we get

$$\Delta_k \leq 5\gamma^2 L_m^2\left[1 + \left(\frac{1-p}{p}\right)\frac{4d}{\mu^2 b'}\right]\Delta_k + \frac{5\gamma^2 d(d+2)L_{f_1}^2}{b} + \frac{5\gamma^2\mu^2 L_{f_2}^2(d+3)^3}{4}$$

$$+ 5\gamma^2 \mathbb{E}\left[\|\nabla \log \pi(\boldsymbol{x}_t)\|^2\right] - \frac{5\gamma^2}{p}\left(e_{k+1}^2 - e_k^2\right) + 5\gamma^2(\sigma_k C + \varepsilon_{\sigma_k} + (\alpha_k - 1)\sigma_k^{-1}C)^2 + 2\gamma d \tag{63}$$

Assume that $\gamma^2 \leq \left[85L_m^2\left(1 + \left(\frac{1-p}{p}\right)\frac{4d}{\mu^2 b'}\right)\right]^{-1}$, then we have

$$\frac{16}{17}\Delta_k \leq \frac{5\gamma^2 d(d+2)L_{f_1}^2}{b} + \frac{5\gamma^2\mu^2 L_{f_2}^2(d+3)^3}{4} + 5\gamma^2 \mathbb{E}\left[\|\nabla \log \pi(\boldsymbol{x}_t)\|^2\right] - \frac{5\gamma^2}{p}\left(e_{k+1}^2 - e_k^2\right)$$

$$+ 5\gamma^2(\sigma_k C + \varepsilon_{\sigma_k} + (\alpha_k - 1)\sigma_k^{-1}C)^2 + 2\gamma d \tag{64}$$

Multiplying both sides by $\frac{17}{16}$, we get

$$\Delta_k \leq \frac{85\gamma^2 d(d+2)L_{f_1}^2}{16b} + \frac{85\gamma^2\mu^2 L_{f_2}^2(d+3)^3}{64} + \frac{85}{16}\gamma^2 \mathbb{E}\left[\|\nabla \log \pi(\boldsymbol{x}_t)\|^2\right]$$

$$- \frac{85\gamma^2}{16p}\left(e_{k+1}^2 - e_k^2\right) + \frac{85}{16}\gamma^2(\sigma_k C + \varepsilon_{\sigma_k} + (\alpha_k - 1)\sigma_k^{-1}C)^2 + \frac{17}{8}\gamma d \tag{65}$$

We can use Lemma 3 to put an upper bound to the third term

$$\Delta_k \leq \frac{85\gamma^2 d(d+2)L_{f_1}^2}{16b} + \frac{85\gamma^2\mu^2 L_{f_2}^2(d+3)^3}{64} + \frac{85}{16}\gamma^2 \mathrm{FI}(\nu_t\|\pi) + \frac{85}{8}\gamma^2 L_\pi d$$

$$- \frac{85\gamma^2}{16p}\left(e_{k+1}^2 - e_k^2\right) + \frac{85}{16}\gamma^2(\sigma_k C + \varepsilon_{\sigma_k} + (\alpha_k - 1)\sigma_k^{-1}C)^2 + \frac{17}{8}\gamma d. \tag{66}$$

We can combine the fourth and the last term by using $\gamma \leq \left[85L_m^2\left(1 + \left(\frac{1-p}{p}\right)\frac{4d}{\mu^2 b'}\right)\right]^{-\frac{1}{2}}$ and concavity of the square root. Note that

$$\gamma \leq \frac{1}{\sqrt{85L_m^2\left(1 + \left(\frac{1-p}{p}\right)\frac{4d}{\mu^2 b'}\right)}} \leq \frac{2}{L_m\sqrt{170} + \sqrt{\left(\frac{1-p}{p}\right)\frac{8d}{\mu^2 b'}}}. \tag{67}$$

We can use this and get

$$\frac{85}{8}\gamma^2 dL_\pi \leq \frac{85\gamma d}{4}\left(\frac{L_m}{L_m\sqrt{170} + \sqrt{\left(\frac{1-p}{p}\right)\frac{8d}{\mu^2 b'}}}\right) \leq \frac{85}{4\sqrt{170}}\gamma d. \tag{68}$$

Finally, we get the upper bound for $\Delta_k$ as

$$\Delta_k \leq \frac{85\gamma^2 d(d+2)L_{f_1}^2}{16b} + \frac{85\gamma^2\mu^2 L_{f_2}^2(d+3)^3}{64} + \frac{85}{16}\gamma^2\mathrm{FI}(\nu_t\|\pi) + 4\gamma d$$

$$- \frac{85\gamma^2}{16p}\left(e_{k+1}^2 - e_k^2\right) + \frac{85}{16}\gamma^2(\sigma_k C + \varepsilon_{\sigma_k} + (\alpha_k - 1)\sigma_k^{-1}C)^2. \tag{69}$$

We define $\phi(\mu) := 1 + \left(\frac{1-p}{p}\right)\frac{4d}{\mu^2 b'}$ for convenience. Plugging (69) into (60), we get

$$\frac{d}{dt}\mathrm{KL}(\nu_t\|\pi) \leq \left(-\frac{3}{4} + \frac{85\gamma^2 L_m^2\phi(\mu)^2}{4}\right)\mathrm{FI}(\nu_t\|\pi) + \left(1 + \frac{85\gamma^2 L_m^2\phi(\mu)}{16}\right)\frac{4d(d+2)L_{f_1}^2}{b}$$

$$+ \left(1 + \frac{85\gamma^2 L_m^2\phi(\mu)}{16}\right)\mu^2 L_{f_2}^2(d+3)^3 - \frac{4}{p}\left(1 + \frac{85\gamma^2 L_m^2\phi(\mu)}{16}\right)(e_{k+1}^2 - e_k^2)$$

$$+ 4\left(1 + \frac{85\gamma^2 L_m^2\phi(\mu)}{16}\right)(\sigma_k C + \varepsilon_{\sigma_k} + (\alpha_k - 1)\sigma_k^{-1}C)^2 + 4\gamma dL_m^2\phi(\mu) \tag{70}$$

$$\leq -\frac{1}{2}\mathrm{FI}(\nu_t\|\pi) + \frac{17d(d+2)L_{f_1}^2}{4b} + \frac{17}{16}\mu^2 L_{f_2}^2(d+3)^3 + 4\gamma dL_m^2\phi(\mu)$$

$$- \frac{4}{p}\left(1 + \frac{85\gamma^2 L_m^2\phi(\mu)}{16}\right)(e_{k+1}^2 - e_k^2) + \frac{17}{4}(\sigma_k C + \varepsilon_{\sigma_k} + (\alpha_k - 1)\sigma_k^{-1}C)^2, \tag{71}$$

where we use the fact that $85\gamma^2 L_m^2\phi(\mu) \leq 1$ to get (71). Integrating both sides between $[k\gamma, (k+1)\gamma]$, we get

$$\mathrm{KL}(\nu_{(k+1)\gamma}\|\pi) - \mathrm{KL}(\nu_{k\gamma}\|\pi) \leq -\frac{1}{2}\int_{k\gamma}^{(k+1)\gamma}\mathrm{FI}(\nu_t\|\pi)dt + \frac{17\gamma d(d+2)L_{f_1}^2}{4b} + \frac{17}{16}\gamma\mu^2 L_{f_2}^2(d+3)^3$$

$$+ 4\gamma^2 dL_m^2\phi(\mu) - \frac{\gamma}{p}\left(4 + \frac{85\gamma^2 L_m^2\phi(\mu)}{16}\right)(e_{k+1}^2 - e_k^2)$$

$$+ \frac{51\gamma}{4}(\sigma_k^2 C^2 + \varepsilon_{\sigma_k}^2 + (\alpha_k - 1)^2\sigma_k^{-2}C^2) \tag{72}$$

Note that for the last term, we use Jensen's inequality. Let $\mathcal{L}_k := \mathrm{KL}(\nu_{k\gamma}\|\pi) + \frac{\gamma}{p}\left[4 + \frac{85}{16}\gamma^2 L_m^2\phi(\mu)\right]e_k^2$, iterating for $k = 0, \ldots, N-1$, multiplying both sides by $\frac{2}{N\gamma}$, rearranging the terms and using the fact that $\mathcal{L}_k \geq 0$, we get

$$\frac{1}{N\gamma}\int_0^{N\gamma}\mathrm{FI}(\nu_t\|\pi)dt \leq \frac{2\mathcal{L}_0}{N\gamma} + \frac{17d(d+2)L_{f_1}^2}{2b} + \frac{17}{8}\mu^2 L_{f_2}^2(d+3)^3 + 8\gamma L_m^2 d\phi(\mu)$$

$$+ \bar{\sigma}^2 + \bar{\varepsilon}_\sigma^2 + \bar{\alpha}^2, \tag{73}$$

where $\bar{\sigma}^2 := \frac{51C}{2N}\sum_{k=0}^{N-1}\sigma_k^2$, $\bar{\varepsilon}_\sigma^2 := \frac{51}{2N}\sum_{k=0}^{N-1}\varepsilon_{\sigma_k}^2$, and $\bar{\alpha} := \frac{51C^2}{2N}\sum_{k=0}^{N-1}\frac{(\alpha_k-1)^2}{\sigma_k^2}$. Since $\gamma^2 L_m^2\phi(\mu) \leq \frac{1}{85}$, we have

$$\mathcal{L}_0 = \mathrm{KL}(\nu_0\|\pi) + \frac{4\gamma}{p}\left(1 + \frac{85}{64}\gamma^2 L_m^2\phi(\mu)\right)e_0^2 \leq \mathrm{KL}(\nu_0\|\pi) + \frac{5\gamma e_0^2}{p}. \tag{74}$$

We can let $C_0 = 2\mathrm{KL}(\nu_0\|\pi) + \frac{10\gamma e_0^2}{p}$ be a constant and this completes the proof of the first statement. To find an $\varepsilon$-approximate solution, we choose the step size as

$$\gamma = \frac{1}{2L_m}\sqrt{\frac{C_0}{Nd\phi(\mu)}} \tag{75}$$

so that we get

$$\frac{1}{N\gamma}\int_0^{N\gamma}\mathrm{FI}(\nu_t\|\pi)dt \le 6L_m\sqrt{\frac{C_0 d\phi(\mu)}{N}} + \frac{17d(d+2)L_{f_1}^2}{2b} + \frac{17}{8}\mu^2 L_{f_2}^2(d+3)^3 + \bar{\sigma}^2 + \bar{\varepsilon}_\sigma^2 + \bar{\alpha}^2. \tag{76}$$

We know that $\{\alpha\}_{k=0}^{N-1}$ and $\{\sigma\}_{k=0}^{N-1}$ are nonincreasing sequences such that

$$\alpha_0 > \alpha_1 > \ldots > \alpha_K = \cdots = \alpha_{N-1} = 1 \quad \text{and} \quad \sigma_0 > \sigma_1 > \ldots > \sigma_K = \cdots = \sigma_{N-1} \approx 0. \tag{77}$$

Therefore, we may choose an index $K < N-1$ such that $\alpha_K = \alpha_{K+1} = \cdots = \alpha_{N-1} = 1$. In practice, one typically selects $K \ll N-1$; see, for example, the geometric annealing schedule in (Sun et al., 2024). Thus, for $\bar{\alpha}^2$, we have

$$\bar{\alpha}^2 = \frac{51C^2}{2N}\sum_{k=0}^{N-1}\frac{(\alpha_k-1)^2}{\sigma_k^2} = \frac{51C^2}{2N}\sum_{k=0}^{K}\frac{(\alpha_k-1)^2}{\sigma_k^2} = O\left(\frac{1}{N}\right). \tag{78}$$

In addition, if the training error is $\varepsilon_{\sigma_k} = O(k^{-\beta})$ for $\beta \ge 1/2$, that is, $\varepsilon_{\sigma_k}^2 \le \frac{C'}{k}$, then

$$\bar{\varepsilon}_\sigma^2 = \frac{51}{2N}\sum_{k=0}^{N-1}\varepsilon_{\sigma_k}^2 \le \frac{51\varepsilon_{\sigma_0}^2}{2N} + \frac{51C'}{2N}\sum_{k=1}^{N}\frac{1}{k} \le \frac{51\varepsilon_{\sigma_0}^2}{2N} + \frac{51C'}{2N} + \frac{51C'\log N}{2N} = O\left(\frac{\log N}{N}\right). \tag{79}$$

Similarly, we can choose the perturbation or smoothing schedule as $\sigma_k = O(k^{-\beta})$ for $\beta \ge 1/2$ noting that the geometric decay schedule used in practice (Sun et al., 2024) decays faster than this and therefore we get $\bar{\sigma}^2 = O\left(\frac{\log N}{N}\right)$. For the convergence of other terms, we can choose a sufficiently large batch size $b$ and sufficiently small smoothing parameter $\mu$ for the zeroth-order estimate as

$$b = \left\lceil\frac{51d(d+2)L_{f_1}^2}{\varepsilon}\right\rceil \quad \text{and} \quad \mu^2 = \frac{4\varepsilon}{51L_{f_2}^2(d+3)^3} \tag{80}$$

where $\lceil.\rceil$ rounds the value to the larger closest integer. Using these values and assuming that $O(\frac{\log N}{N}) \le O(\varepsilon)$ we get

$$\frac{1}{N\gamma}\int_0^{N\gamma}\mathrm{FI}(\nu_t\|\pi)dt \le 6L_m\sqrt{\frac{C_0 d\phi(\mu)}{N}} + \frac{5\varepsilon}{6}. \tag{81}$$

To make the per-iteration complexity constant, we can choose $p = \frac{1}{b}$ so that we have $pb = O(1)$. We can find the lower bound on $N$ by using the following inequality

$$6L_m\sqrt{\frac{C_0 d\left(1 + \left(\frac{1-p}{p}\right)\frac{4d}{\mu^2 b'}\right)}{N}} \le 6L_m\sqrt{\frac{C_0 d\left(1 + \frac{4d}{p\mu^2 b'}\right)}{N}} \le \frac{\varepsilon}{6}. \tag{82}$$

Plugging the chosen values for $p$ and $\mu$, we get

$$N \ge \frac{C_1 d L_m^2}{\varepsilon^2} + \frac{C_2(d+2)^7 L_m^6}{\varepsilon^4}, \tag{83}$$

where $C_1 = 1296\mathrm{KL}(\nu_0\|\pi) + \frac{6480\gamma}{p}$ and $C_2 = 36^2 \times 51^2$ are numerical constants. That implies

$$N = O\left(\frac{d^7 L_m^6}{\varepsilon^4}\right) \tag{84}$$

number of iteration complexity, which is also equivalent to number forward model evaluations because per-iteration complexity is fixed $pb = O(1)$. Plugging the iteration complexity in $O\left(\frac{\log N}{N}\right)$, it is clear that $O\left(\frac{\log N}{N}\right) \le O(\varepsilon)$. That proves the $\varepsilon-$approximate bound on (10) and concludes the proof. □

### A.3.4 PROOF OF COROLLARY 1

Recall that from Theorem 1 proof, we have inequality (73) as

$$\frac{1}{N\gamma}\int_0^{N\gamma}\mathrm{FI}(\nu_t\|\pi)dt \le \frac{C_0}{N\gamma} + \frac{17d(d+2)L_{f_1}^2}{2b} + \frac{17}{8}\mu^2 L_{f_2}^2(d+3)^3 + 8\gamma L_m^2 d\phi(\mu)$$
$$+ \bar{\sigma}^2 + \bar{\varepsilon}_\sigma^2 + \bar{\alpha}^2, \tag{85}$$

where $C_0 = 2\mathrm{KL}(\nu_0\|\pi) + \frac{10\gamma e_0^2}{p}$. By the convexity of the Fisher information, we have

$$\mathrm{FI}(\bar{\nu}\|\pi) \le \frac{1}{N\gamma}\int_0^{N\gamma}\mathrm{FI}(\nu_t\|\pi)dt$$
$$\le \frac{C_0}{N\gamma} + \frac{17d(d+2)L_{f_1}^2}{2b} + \frac{17}{8}\mu^2 L_{f_2}^2(d+3)^3 + 8\gamma L_m^2 d\phi(\mu)$$
$$+ \bar{\sigma}^2 + \bar{\varepsilon}_\sigma^2 + \bar{\alpha}^2 \tag{86}$$

where $\bar{\nu}_{N\gamma} := \frac{1}{N\gamma}\int_0^{N\gamma}\nu_t dt$. By Assumption 6, we can invoke Lemma 4 and get

$$\|\bar{\nu}_{N\gamma} - \pi\|_{\mathrm{TV}}^2 \le 4C_{\mathrm{PI}}\mathrm{FI}(\bar{\nu}\|\pi)$$
$$\le \frac{4C_0 C_{\mathrm{PI}}}{N\gamma} + \frac{34d(d+2)C_{\mathrm{PI}}L_{f_1}^2}{b} + \frac{17}{2}\mu^2 C_{\mathrm{PI}}L_{f_2}^2(d+3)^3 + 32\gamma C_{\mathrm{PI}}L_m^2 d\phi(\mu)$$
$$+ 4C_{\mathrm{PI}}(\bar{\sigma}^2 + \bar{\varepsilon}_\sigma^2 + \bar{\alpha}^2). \tag{87}$$

If we let

$$\gamma = \frac{1}{2L_m}\sqrt{\frac{C_0 C_{\mathrm{PI}}}{Nd(1 + \left(\frac{1-p}{p}\right)\frac{4d}{\mu^2 b'})}} \quad b = \left\lceil\frac{204d(d+2)C_{\mathrm{PI}}L_{f_1}^2}{\varepsilon}\right\rceil, \quad \mu = \sqrt{\frac{\varepsilon}{51C_{\mathrm{PI}}L_{f_2}^2(d+3)^3}}$$

$$p = \frac{1}{b} \quad \text{and} \quad \bar{\sigma}^2 + \bar{\varepsilon}_\sigma^2 + \bar{\alpha}^2 \le \varepsilon/8C_{\mathrm{PI}},$$

plugging these values in (87), we get

$$\|\bar{\nu}_{N\gamma} - \pi\|_{\mathrm{TV}}^2 \le 24L_m\sqrt{\frac{C_p C_{\mathrm{PI}}d}{N}\left(1 + \left(\frac{1-p}{p}\right)\frac{4d}{\mu^2 b'}\right)} + \frac{5\varepsilon}{6}. \tag{88}$$

We can bound the first term with $\frac{\varepsilon}{6}$ if we choose the number of iteration as

$$N \ge \frac{C_1 C_{\mathrm{PI}}L_m^2 d}{\varepsilon^2} + \frac{C_2 C_{\mathrm{PI}}^3 L_m^6(d+3)^7}{\varepsilon^4}, \tag{89}$$

where $C_1 = 36\times 16^2 C_0$ and $C_2 = 36\times 16^2\times 24\times 34\times 51 C_0$ are numerical constants. This yields a forward model complexity of

$$N = O\left(\frac{d^7 L_m^6 C_{\mathrm{PI}}^3}{\varepsilon^4}\right),$$

with a fixed per-iteration cost $pb = O(1)$ to achieve $\|\bar{\nu}_{N\gamma} - \pi\|_{\mathrm{TV}}^2 \le \varepsilon$. That proves Corollary 1. □

### A.3.5 PROOF OF THEOREM 2

Given step–dependent parameters $\gamma_n, b_n, p_n, \mu_n$, define

$$\tau_n := \sum_{k=1}^n \gamma_k, \qquad \bar{\nu}_{\tau_n} := \frac{1}{\tau_n}\int_0^{\tau_n}\nu_t\,dt,$$

where $\nu_t$ denotes the law of the process $\boldsymbol{x}_t$ specified by

$$\boldsymbol{x}_t := \boldsymbol{x}_{\tau_{n-1}} - (t-\tau_{n-1})\,\mathcal{G}\left(\boldsymbol{x}_{\tau_{n-1}}\right) + \sqrt{2}\left(\boldsymbol{B}_t - \boldsymbol{B}_{\tau_{n-1}}\right), \qquad t\in[\tau_{n-1},\tau_n]. \tag{90}$$

Here, $\gamma_n$ denotes the step size used at iteration $n$, while $\tau_n$ denotes the cumulative time elapsed up to iteration $n$. We note that the steps up to (72) in the proof of Theorem 1 hold for $t \in [\tau_{n-1}, \tau_n]$ with step-dependent parameters. Then, we can write the one-step recursion of ZO-APMC as

$$
\mathrm{KL}(\nu_{\tau_n}\|\pi) - \mathrm{KL}(\nu_{\tau_{n-1}}\|\pi) \leq -\frac{1}{2}\int_{\tau_{n-1}}^{\tau_n} \mathrm{FI}(\nu_t\|\pi)dt + \frac{17\gamma_n d(d+2)L_{f_1}^2}{4b} + \frac{17}{16}\gamma_n\mu_n^2 L_{f_2}^2(d+3)^3
$$

$$
+ 4\gamma_n^2 dL_m^2\phi(\mu) - \frac{\gamma_n}{p_n}\left(4 + \frac{85\gamma_n^2 L_m^2\phi_n(\mu_n)}{16}\right)\left(e_n^2 - e_{n-1}^2\right)
$$

$$
+ \frac{51\gamma}{4}(\sigma_n^2 C^2 + \varepsilon_{\sigma_n}^2 + (\alpha_n - 1)^2\sigma_n^{-2}C^2) \tag{91}
$$

for $t \in [\tau_{n-1}, \tau_n]$. We choose the parameters as follows

$$
\gamma_n = \frac{1}{n}\sqrt{\frac{b'}{680L_m^2 d}}, \quad b = \lceil n^{\frac{1}{2}}\rceil, \quad p = n^{-\frac{1}{2}}, \quad \mu_n = n^{-\frac{1}{8}}. \tag{92}
$$

We emphasize that the chosen step size for $\gamma_n$, namely $\gamma_n \in \left(0,\, 1/\left(85\,L_m^2\phi_n(\mu_n)\right)\right]$, satisfies this condition for all $n \in \mathbb{N}^+$. Plugging these selected values into (91), we get

$$
\mathrm{KL}(\nu_{\tau_n}\|\pi) - \mathrm{KL}(\nu_{\tau_{n-1}}\|\pi) \leq -\frac{1}{2}\int_{\tau_{n-1}}^{\tau_n} \mathrm{FI}(\nu_t\|\pi)dt + \frac{A_1}{n^{\frac{3}{2}}} + \frac{A_2}{n^{\frac{5}{4}}} + \frac{A_3}{n^2}\left(1 + \frac{4dn^{\frac{3}{4}}}{b'}\right)
$$

$$
- c_n\left(e_n^2 - e_{n-1}^2\right) + \frac{51\gamma_n}{4}(\sigma_n^2 C^2 + \varepsilon_{\sigma_n}^2 + (\alpha_n - 1)^2\sigma_n^{-2}C^2), \tag{93}
$$

where

$$
A_1 = \frac{17d(d+2)L_{f_1}^2}{4}\sqrt{\frac{b'}{680L_m^2 d}}, \quad A_2 = \frac{17L_{f_2}^2(d+3)^3}{16}\sqrt{\frac{b'}{680L_m^2 d}}, \quad A_3 = 4dL_m^2\sqrt{\frac{b'}{680L_m^2 d}}, \tag{94}
$$

and all of them are constant. Moreover, we define

$$
c_n := \frac{\gamma_n}{p_n}\left(4 + \frac{85\gamma_n^2 L_m^2\phi_n(\mu_n)}{16}\right)\left(e_n^2 - e_{n-1}^2\right), \tag{95}
$$

for $n \in \mathbb{N}^+$. Iterating the bound in (93), we obtain

$$
\mathrm{KL}(\nu_{\tau_n}\|\pi) \leq \mathrm{KL}(\nu_0\|\pi) - \frac{1}{2}\int_0^{\tau_n} \mathrm{FI}(\nu_t\|\pi)dt + A_1 S_1 + A_2 S_2 + A_3 S_3 - \sum_{k=1}^n c_k\left(e_k^2 - e_{k-1}^2\right)
$$

$$
+ \frac{51}{4}\sum_{k=1}^n \gamma_k(\sigma_k^2 C^2 + \varepsilon_{\sigma_k}^2 + (\alpha_k - 1)^2\sigma_k^{-2}C^2), \tag{96}
$$

where we have

$$
S_1 = \sum_{k=1}^n k^{-3/2} \leq \sum_{k=1}^\infty k^{-3/2} < \infty, \quad S_2 = \sum_{k=1}^n k^{-5/4} \leq \sum_{k=1}^\infty k^{-5/4} < \infty.
$$

$$
S_3 = \sum_{k=1}^n\left(k^{-2} + \frac{4dk^{-5/4}}{b'}\right) \leq \sum_{k=1}^\infty\left(k^{-2} + \frac{4dk^{-5/4}}{b'}\right) < \infty.
$$

Thus, $A_1 S_1$, $A_2 S_2$, and $A_3 S_3$ are bounded constants and are independent of $n$. Furthermore, if we assume that $c_n$ is nonnegative and nonincreasing sequence (i.e. $0 \leq c_{n+1} \leq c_n$), we can bound the summation of difference of the error terms in (96) as follows

$$
-\sum_{k=1}^n c_k\left(e_k^2 - e_{k-1}^2\right) = c_1 e_0^2 + \sum_{k=1}^{n_1}(c_{k+1} - c_k)e_k^2 - c_n e_n^2 \leq c_1 e_0. \tag{97}
$$

Thus, we need to prove that $c_n$ is nonnegative and nonincreasing sequence. Plugging the variables in (92) into the definition of $c_n$, we get

$$c_n = \frac{1}{n^{1/2}} \sqrt{\frac{b'}{680 L_m^2 d}} \left[4 + \frac{85}{12 n^2} \frac{b'}{680 d}\right] + \frac{1}{n^{9/4}} \left(1 - \frac{1}{n^{1/2}}\right) \sqrt{\frac{b'}{680 L_m^2 d}} \frac{85 b'}{12 \times 680 d}. \tag{98}$$

Therefore, $c_n \geq 0$ for any $n \in \mathbb{N}^+$ and we have $c_n = O(n^{-1/2})$ so it's a nonincreasing sequence. Using this upper bound on the differences of error terms in (93), we get

$$\mathrm{KL}(\nu_{\tau_n} \| \pi) \leq \mathrm{KL}(\nu_0 \| \pi) - \frac{1}{2} \int_0^{\tau_n} \mathrm{FI}(\nu_t \| \pi) dt + A_1 S_1 + A_2 S_2 + A_3 S_3 + c_1 e_0^2$$

$$+ \frac{51}{4} \sum_{k=1}^n \gamma_k (\sigma_k^2 C^2 + \varepsilon_{\sigma_k}^2 + (\alpha_k - 1)^2 \sigma_k^{-2} C^2) \tag{99}$$

We need to upper bound the last term related to the annealing parameters and the score network. Note that $\{\alpha_k\}_{k=0}^{N-1}$ is a decreasing sequence with $\alpha_{K,\ldots,N-1} = 1$ for some $K$. Hence, we have

$$\frac{51}{4} \sum_{k=1}^n \gamma_k (\alpha_k - 1)^2 \sigma_k^{-2} C^2 \leq \frac{51 C^2}{4} \sum_{k=1}^\infty \gamma_k (\alpha_k - 1)^2 \sigma_k^{-2} = \frac{51 C^2}{4} \sum_{k=1}^K \gamma_k (\alpha_k - 1)^2 \sigma_k^{-2} = C_\alpha, \tag{100}$$

where $C_\alpha < \infty$ is bounded constant. Additionally, we can choose a schedule for $\sigma_k$ such that $\sigma_k = O(k^{-\beta})$ for $\beta > 0$. For example, in practice, one often uses an exponentially decaying schedule $\sigma_k = \sigma_0 \xi^k$ (Sun et al., 2024; Song & Ermon, 2019), where $\xi$ denotes the decay rate. Then, we get

$$\frac{51}{4} \sum_{k=1}^n \gamma_k \sigma_k^2 C^2 \leq \frac{51}{4} \sum_{k=1}^\infty \gamma_k \sigma_k^2 C^2 = C_\sigma < \infty. \tag{101}$$

Lastly, if we assume that $\varepsilon_{\sigma_k} = O(k^{-\beta})$ for some $\beta > 0$, then

$$\frac{51}{4} \sum_{k=1}^n \gamma_k \varepsilon_{\sigma_k}^2 \leq \frac{51}{4} \sum_{k=1}^\infty \gamma_k \varepsilon_{\sigma_k}^2 = C_{\varepsilon_\sigma} < \infty. \tag{102}$$

This assumption implies that, as the noise level decreases, the test error of the score network forms a monotonically decreasing sequence of order $O(n^{-\beta})$ with $\beta > 0$. Such a decay pattern is commonly observed in diffusion models, where networks trained across noise scales exhibit progressively lower errors at finer (less noisy) levels (Song et al., 2020b; Ho et al., 2020). Combining (100), (101), and (102) in (99), we get

$$\mathrm{KL}(\nu_{\tau_n} \| \pi) \leq \mathrm{KL}(\nu_0 \| \pi) - \frac{1}{2} \int_0^{\tau_n} \mathrm{FI}(\nu_t \| \pi) dt + A_1 S_1 + A_2 S_2 + A_3 S_3 + c_1 e_0^2$$

$$+ C_\alpha + C_{\varepsilon_\sigma} + C_\sigma \tag{103}$$

Rearranging the terms, using the convexity of Fisher information and dividing both sides by $\tau_n$, we get

$$\mathrm{FI}(\bar{\nu}_{\tau_n} \| \pi) \leq \frac{1}{\tau_n} \int_0^{\tau_n} \mathrm{FI}(\nu_t \| \pi) \, dt$$

$$\leq \frac{2 \mathrm{KL}(\nu_0 \| \pi)}{\tau_n} + \frac{2}{\tau_n} \left(A_1 S_1 + A_2 S_2 + A_3 S_3 + c_1 e_0^2 + C_\alpha + C_{\varepsilon_\sigma} + C_\sigma\right) \tag{104}$$

where $A_1 S_1 + A_2 S_2 + A_3 S_3 + c_1 e_0^2 + C_\alpha + C_{\varepsilon_\sigma} + C_\sigma < \infty$. Alternatively, if $t \in [\tau_n, \tau_{n+1}]$, integrating (71) between $\tau_n$ and $t$ and dropping the negative integral over the Fisher information give us

$$\mathrm{KL}(\nu_t \| \pi) \leq \mathrm{KL}(\tau_n \| \pi) + \frac{17(t - \tau_n) d(d+2) L_{f_1}^2}{4b} + \frac{17}{16}(t - \tau_n) \mu^2 L_{f_2}^2 (d+3)^3$$

$$+ 4(t - \tau_n) \gamma d L_m^2 \phi(\mu) - \frac{4(t - \tau_n)}{p} \left(1 + \frac{85 \gamma^2 L_m^2 \phi(\mu)}{16}\right) e_{n+1}^2$$

$$+ \frac{17(t - \tau_n)}{4} (\sigma_n C + \varepsilon_{\sigma_n} + (\alpha_n - 1) \sigma_n^{-2} C^2)^2 \tag{105}$$

$$\leq \mathrm{KL}(\nu_0 \| \pi) + 2 A_1 S_1 + 2 A_2 S_2 + 2 A_3 S_3 + 2 C_\alpha + 2 C_{\varepsilon_\sigma} + 2 C_\sigma + c_1 e_0^2. \tag{106}$$

where, in the second inequality, for positive terms, we use the fact that $\frac{1}{n+1} \leq \sum_{k=1}^{n} \frac{1}{k^\beta}$ for $\beta > 1$. With (106), we show that $\{\mathrm{KL}(\nu_t\|\pi)\}_{t \geq 0}$ is bounded. By the convexity of the KL divergence, this implies that the sequence $\{\mathrm{KL}(\bar{\nu}_{\tau_n}\|\pi)\}_{n \in \mathbb{N}}$ is uniformly bounded as well. Since the sublevel sets of $\mathrm{KL}(\cdot\|\pi)$ are weakly compact, $(\bar{\nu}_{\tau_n})_{n \in \mathbb{N}}$ is tight. To establish that $\bar{\nu}_{\tau_n} \rightharpoonup \pi$ weakly, it suffices to verify that every cluster point of $(\bar{\nu}_{\tau_n})_{n \in \mathbb{N}}$ equal to $\pi$.

Take a subsequence $(\bar{\nu}_{\tau_n})_{n \in \mathbb{N}}$ converging to some limit $\bar{\nu}$. Sending $n \to \infty$ in (104) and using $\tau_n \to \infty$ gives $\mathrm{FI}(\bar{\nu}_{\tau_n}\|\pi) \to 0$, so the same holds along the subsequence. By the weak lower semicontinuity of the Fisher information along the subsequence, we have $\mathrm{FI}(\bar{\nu}\|\pi) = 0$. Writing $\psi := \frac{d\bar{\nu}}{d\pi}$, this means $\sqrt{\psi} \in \mathrm{dom}\,\mathcal{E}$ and $\mathcal{E}(\sqrt{\psi}) = 0$. Since $\nabla \log \pi$ is Lipschitz by Assumption 1, $\pi$ has a continuous and strictly positive density on $\mathbb{R}^d$, so $\mathcal{E}(\sqrt{\psi}) = 0$, which implies that $\psi$ must be a constant $\pi$-a.e., hence $\bar{\nu} = \pi$. $\qquad\square$

### A.3.6 PROOF OF THEOREM 3

We can construct the following interpolation using the proposed estimator in 15 and update rule 16

$$\boldsymbol{x}_t := \boldsymbol{x}_{k\gamma} - (t - k\gamma)\mathcal{G}(\boldsymbol{x}_{k\gamma}) + \sqrt{2}(\boldsymbol{B}_t - \boldsymbol{B}_{k\gamma}), \quad \text{for} \quad t \in [k\gamma, (k+1)\gamma], \qquad (107)$$

where $\mathcal{G}(\boldsymbol{x}_{k\gamma}) = \boldsymbol{g}_k - \alpha_k \mathcal{S}_\theta(\boldsymbol{x}_{k\gamma})$ and $\boldsymbol{g}_k$ is the modified estimator in (15). We can use Lemma 2 with the interpolation argument and get

$$\frac{d}{dt}\mathrm{KL}(\nu_t\|\pi) \leq -\frac{3}{4}\mathrm{FI}(\nu_t\|\pi) + \mathbb{E}[\|\nabla \log \pi(\boldsymbol{x}_t) + \boldsymbol{g}_k - \alpha_k\|^2]. \qquad (108)$$

We can follow the same approach in the analysis of Theorem 1 and get

$$\frac{d}{dt}\mathrm{KL}(\nu_t\|\pi) \leq -\frac{3}{4}\mathrm{FI}(\nu_t\|\pi) + 3L_\pi^2\Delta_k + 3e_k^2 + 3(\sigma_k C + \varepsilon_{\sigma_k} + (\alpha_k - 1)\sigma_k^{-1}C)^2, \qquad (109)$$

where $e_k^2 := \mathbb{E}[\|\boldsymbol{g}_k - \nabla f(\boldsymbol{x}_k)\|^2]$ and $\Delta_k := \mathbb{E}[\|\boldsymbol{x}_{k+1} - \boldsymbol{x}_k\|^2]$. First, we upper bound the error term using the definition of $\boldsymbol{g}_k$ as follows

$$e_{k+1}^2 = \mathbb{E}[\|\boldsymbol{g}_{k+1} - \nabla f(\boldsymbol{x}_{k+1})\|^2] \qquad (110)$$

$$= \frac{p\sigma_{\mathrm{noise}}^2}{b} + (1-p)\mathbb{E}[\|\boldsymbol{g}_k - \nabla f(\boldsymbol{x}_{k+1})\|^2] \qquad (111)$$

$$= \frac{p\sigma_{\mathrm{noise}}^2}{b} + (1-p)\mathbb{E}[\|\boldsymbol{g}_k - \nabla f(\boldsymbol{x}_k) + \nabla f(\boldsymbol{x}_k) - \nabla f(\boldsymbol{x}_{k+1})\|^2] \qquad (112)$$

$$\leq \frac{p\sigma_{\mathrm{noise}}^2}{b} + (1-p)\left(1 + \frac{1}{\rho_0}\right)L_{f_2}^2\Delta_k + (1-p)(1+\rho_0)e_k^2 \qquad (113)$$

$$\leq \frac{p\sigma_{\mathrm{noise}}^2}{b} + \frac{2(1-p)}{p}L_{f_2}^2\Delta_k + \left(1 - \frac{p}{2}\right)e_k^2. \qquad (114)$$

To obtain (113), we apply the Cauchy and Young's inequalities. Then, by choosing $\rho_0 = \frac{p}{2(1-p)}$, we arrive at (114). Rearranging the terms, we get

$$e_k^2 \leq -\frac{2}{p}(e_{k+1}^2 - e_k^2) + \frac{4(1-p)}{p^2}L_{f_2}^2\Delta_k + \frac{2\sigma_{\mathrm{noise}}^2}{b}. \qquad (115)$$

Plugging this in (109), we get

$$\frac{d}{dt}\mathrm{KL}(\nu_t\|\pi) \leq -\frac{3}{4}\mathrm{FI}(\nu_t\|\pi) + 3L_\pi^2\left(1 + \frac{4(1-p)}{p^2}\right)\Delta_k + \frac{6\sigma_{\mathrm{noise}}^2}{b} - \frac{6}{p}(e_{k+1}^2 - e_k^2)$$

$$+ 3(\sigma_k C + \varepsilon_{\sigma_k} + (\alpha_k - 1)\sigma_k^{-1}C)^2. \qquad (116)$$

Following the similar analysis in Theorem 1, we can get the following upper bound on $\Delta_k$ using the interpolation argument in (107)

$$\Delta_k \leq 4\gamma^2 e_k^2 + 4\gamma^2 L_\pi^2\Delta_k + 4\gamma^2\mathbb{E}[\|\nabla \log \pi(\boldsymbol{x}_t)\|^2] + 2\gamma d \qquad (117)$$

$$+ 4\gamma^2(\sigma_k C + \varepsilon_{\sigma_k} + (\alpha_k - 1)\sigma_k^{-1}C)^2. \qquad (118)$$

$$\qquad (119)$$

Using the upper bound on $e_k^2$ derived in (115) with the fact that $L_{f_2} \leq L_\pi$, we get

$$\Delta_k \leq 4\gamma^2 L_\pi^2 \phi(p) \Delta_k + \frac{8\gamma^2 \sigma_{\text{noise}}^2}{b} - \frac{8\gamma^2}{p} \left(e_{k+1}^2 - e_k^2\right) + 4\gamma^2 \mathbb{E}\left[\|\nabla \log \pi(\boldsymbol{x}_t)\|^2\right]$$
$$+ 2\gamma d + 4\gamma^2 \left(\sigma_k C + \varepsilon_{\sigma_k} + (\alpha_k - 1)\sigma_k^{-1} C\right)^2, \tag{120}$$

where $\phi(p) := 1 + \frac{4(1-p)}{p^2}$. Rearranging the terms, we have

$$\left(1 - 4\gamma^2 L_\pi^2 \phi(p)\right) \Delta_k \leq \frac{8\gamma^2 \sigma_{\text{noise}}^2}{b} + 4\gamma^2 \mathbb{E}[\|\nabla \log \pi(\boldsymbol{x}_t)\|^2] - \frac{8\gamma^2}{p} \left(e_{k+1}^2 - e_k^2\right) \tag{121}$$
$$+ 4\gamma^2 \left(\sigma_k C + \varepsilon_{\sigma_k} + (\alpha_k - 1)\sigma_k^{-1} C\right)^2 + 2\gamma d. \tag{122}$$

Let $\gamma^2 L_\pi^2 \leq \frac{1}{52\phi(p)}$, then multiplying both sides by $\frac{13}{12}$, we get

$$\Delta_k \leq \frac{26}{3} \frac{\gamma^2 \sigma_{\text{noise}}^2}{b} + \frac{13}{3}\gamma^2 \mathbb{E}[\|\nabla \log \pi(\boldsymbol{x}_t)\|^2] - \frac{26}{3}\gamma^2 (e_{k+1}^2 - e_k^2)$$
$$+ \frac{13}{3}(\sigma_k C + \varepsilon_{\sigma_k} + (\alpha_k - 1)\sigma_k^{-1} C)^2 + \frac{13}{6}\gamma d. \tag{123}$$

Plugging this into (116), we get

$$\frac{d}{dt}\text{KL}(\nu_t \| \pi) \leq -\frac{3}{4}\text{FI}(\nu_t \| \pi) + \left(6 + 26\gamma^2 L_\pi^2 \phi(p)\right) \frac{\sigma_{\text{noise}}^2}{b} - \frac{6}{p}\left(1 + \frac{13}{3}\gamma^2 L_\pi^2 \phi(p)\right)(e_{k+1}^2 - e_k^2)$$
$$+ 13\gamma^2 L_\pi^2 \phi(p)\mathbb{E}[\|\nabla \log \pi(\boldsymbol{x}_t)\|^2] + \frac{13}{2}\gamma L_\pi^2 \phi(p) d$$
$$+ 3\left(1 + \frac{13}{3}\gamma^2 L_\pi^2 \phi(p)\right)(\sigma_k C + \varepsilon_{\sigma_k} + (\alpha_k - 1)\sigma_k^{-1} C)^2. \tag{124}$$

Using the fact that $\gamma^2 L_\pi^2 \phi(p) \leq \frac{1}{52}$, we can do the following simplifications

$$1 + \frac{13}{3}\gamma^2 L_\pi^2 \phi(p) \leq \frac{13}{12} \quad \text{and} \quad 6 + 26\gamma^2 L_\pi^2 \phi(p) \leq \frac{13}{12}. \tag{125}$$

Then, we get

$$\frac{d}{dt}\text{KL}(\nu_t \| \pi) \leq -\frac{3}{4}\text{FI}(\nu_t \| \pi) + \frac{13}{2}\frac{\sigma_{\text{noise}}^2}{b} - \frac{6}{p}\left(1 + \frac{13}{3}\gamma^2 L_\pi^2 \phi(p)\right)(e_{k+1}^2 - e_k^2)$$
$$+ 13\gamma^2 L_\pi^2 \phi(p)\mathbb{E}[\|\nabla \log \pi(\boldsymbol{x}_t)\|^2] + \frac{13}{2}\gamma L_\pi^2 \phi(p) d$$
$$+ \frac{13}{4}(\sigma_k C + \varepsilon_{\sigma_k} + (\alpha_k - 1)\sigma_k^{-1} C)^2. \tag{126}$$

Using Lemma 3, we get

$$\frac{d}{dt}\text{KL}(\nu_t \| \pi) \leq -\frac{1}{2}\text{FI}(\nu_t \| \pi) + \frac{13}{2}\frac{\sigma_{\text{noise}}^2}{b} - \frac{6}{p}\left(1 + \frac{13}{3}\gamma^2 L_\pi^2 \phi(p)\right)(e_{k+1}^2 - e_k^2)$$
$$+ \left(\frac{13}{2} + 26\gamma L_\pi\right)\gamma L_\pi^2 \phi(p) d + \frac{13}{4}(\sigma_k C + \varepsilon_{\sigma_k} + (\alpha_k - 1)\sigma_k^{-1} C)^2. \tag{127}$$

Note that

$$\frac{13}{2} + 26\gamma L_\pi \leq \frac{17}{2}, \tag{128}$$

so we get

$$\frac{d}{dt}\text{KL}(\nu_t \| \pi) \leq -\frac{1}{2}\text{FI}(\nu_t \| \pi) + \frac{13}{2}\frac{\sigma_{\text{noise}}^2}{b} - \frac{6}{p}\left(1 + \frac{13}{3}\gamma^2 L_\pi^2 \phi(p)\right)(e_{k+1}^2 - e_k^2)$$
$$+ \frac{17}{2}\gamma L_\pi^2 \phi(p) d + \frac{13}{4}(\sigma_k C + \varepsilon_{\sigma_k} + (\alpha_k - 1)\sigma_k^{-1} C)^2. \tag{129}$$

Integrating from $k\gamma$ to $(k+1)\gamma$ and let $\mathcal{L}_k := \mathrm{KL}(\nu_{k\gamma\|\pi}) + \frac{6\gamma}{p}(1 + \frac{13}{3}\gamma^2 L_\pi^2\phi(p))e_k^2$,

$$\mathcal{L}_{k+1} - \mathcal{L}_k \leq -\frac{1}{2}\int_{k\gamma}^{(k+1)\gamma} \mathrm{FI}(\nu_t\|\pi)\,dt + \frac{13}{2}\frac{\gamma\sigma_{\mathrm{noise}}^2}{b} + \frac{17}{2}\gamma^2 L_\pi^2\phi(p)d$$

$$+ \frac{13}{2}(\sigma_k C + \varepsilon_{\sigma_k} + (\alpha_k - 1)\sigma_k^{-1}C)^2. \tag{130}$$

Integrating for $k = 0, 1, \ldots, N-1$, and rearranging the terms, we get

$$\frac{1}{N\gamma}\int_0^{N\gamma} \mathrm{FI}(\nu_t\|\pi)\,dt \leq \frac{2\mathcal{L}_0}{N\gamma} + \frac{13\sigma_{\mathrm{noise}}^2}{b} + \frac{17}{2}\gamma L_\pi^2 d\left(1 + \frac{4(1-p)}{p^2}\right) + \bar{\sigma}^2 + \bar{\varepsilon}_\sigma + \bar{\alpha}, \tag{131}$$

where

$$\bar{\sigma}^2 := \frac{39C^2}{N}\sum_{k=0}^{N-1}\sigma_k^2, \quad \bar{\varepsilon}_\sigma^2 := \frac{39}{N}\sum_{k=0}^{N-1}\varepsilon_{\sigma_k}^2, \quad \text{and} \quad \bar{\alpha}^2 := \frac{39C^2}{N}\sum_{k=0}^{N-1}\frac{(\alpha_k - 1)^2}{\sigma_k^2}, \tag{132}$$

and $\mathcal{L}_0$ was defined as

$$2\mathcal{L}_0 = 2\mathrm{KL}(\nu_0\|\pi) + \frac{12\gamma}{p}\left(1 + \frac{13}{3}\gamma^2 L_\pi^2\phi(p)\right)e_0^2 \leq 2\mathrm{KL}(\nu_0\|\pi) + \frac{13\gamma}{p}\mathbb{E}[\|\boldsymbol{g}_0 - \nabla f(\boldsymbol{x}_0)\|^2] = C_0, \tag{133}$$

where $C_0$ is a numerical constant. This concludes the first statement of the theorem. Furthermore, if we let

$$\gamma = \frac{1}{L_\pi}\sqrt{\frac{2C_0}{17d\phi(p)N}}, \quad \text{and} \quad b = \frac{65\sigma_{\mathrm{noise}}^2}{\varepsilon}, \tag{134}$$

we get

$$\frac{1}{N\gamma}\int_0^{N\gamma} \mathrm{FI}(\nu_t\|\pi)\,dt \leq 2L_\pi\sqrt{\frac{17C_0 d\phi(p)}{N}} + \frac{\varepsilon}{5} + \bar{\sigma}^2 + \bar{\varepsilon}_\sigma + \bar{\alpha}. \tag{135}$$

Similar to Theorem 1, if we let $\sigma_k = O(k^{-\beta})$, $\varepsilon_{\sigma_k} = O(k^{-\beta})$ for any $\beta \geq 1/2$ and all $k \geq 1$ with initial values $\sigma_0 > 0$ and $\varepsilon_{\sigma_0} > 0$, we get

$$\frac{1}{N\gamma}\int_0^{N\gamma} \mathrm{FI}(\nu_t\|\pi)\,dt \leq 2L_\pi\sqrt{\frac{17C_0 d\phi(p)}{N}} + \frac{4\varepsilon}{5}. \tag{136}$$

Since we want convergence for fixed per-iteration cost $pb = O(1)$, we choose $p = 1/b$. Thus, we need to do $N = O\left(L_\pi^2\sigma_{\mathrm{noise}}^4 d^2/\varepsilon^4\right)$ number of iterations or total gradient calculations for $\varepsilon$-approximate convergence in FI. That concludes the proof $\qquad\square$.

## A.4 Extended Experimental Results

### A.4.1 Numerical Validation

In this section, we give more details on our numerical validation experiments. Similar to the related works (Sun et al., 2024; Song & Ermon, 2020), we run ZO-APMC with an exponential annealing schedule:

$$\sigma_k := \max\{\sigma_0 r^k, \sigma_{\min}\}, \quad \alpha_k = \max\{\alpha_0\sigma_k^2, 1\}, \tag{137}$$

where $r$ is the decay factor and $k$ is the step number. We always choose $\alpha_0 \leq 1/\sigma_{\min}^2$ so that $\{\alpha_k\}$ converges to 1. For our numerical validation results, we set $\sigma_0 = 10$, $\alpha_0 = 10$, $r = 0.975$, $\sigma_{\min} = 0$ and $\gamma = 0.1$. For ZO estimator, we choose the smoothing parameter as $\mu = 10^{-4}$. We run ZO-APMC with 1000 sample points initialized with uniform distribution $\mathrm{U}[-50, 50]^2$ on $[-50, 50]^2$ grid for $N = 2000$ iterations. At each step, we use a Gaussian mixture model (GMM) to fit a distribution to the samples at intermediate steps, which allows us to compute the probability of an arbitrary value on $[-50, 50]^2$ grid. Then, for each intermediate distribution, we calculate the empirical Fisher information relative to target posterior whose analytical posterior can be calculated. We discretize the grid to $1000 \times 1000$ unit areas in $[-50, 50]^2$ and calculate the Fisher information for each unit area. The total sum over the grid gives us the approximate relative Fisher information. We provide additional experiments illustrating the effect of $p$ on FI convergence in Fig. 8.

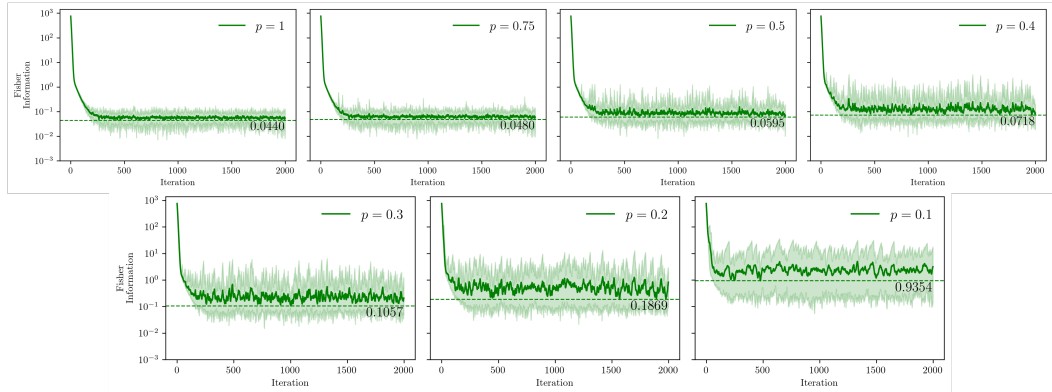

Figure 8: Effect of $p$ on the convergence of ZO-APMC to the true posterior distribution in terms of relative Fisher information. The solid lines show the mean values and shaded areas show the minimum and maximum ranges.

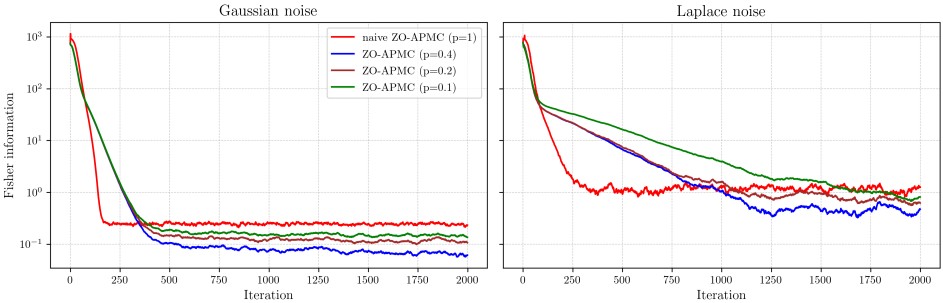

Figure 9: Comparison of naive ZO-APMC ($p = 1$) and ZO-APMC with $p < 1$ for Fisher information convergence under Gaussian (left) and Laplace (right) measurement noise types in an inverse problem. Each plot performs the same number of forward model evaluations per iteration on average.

At each iteration, ZO-APMC performs a zero-order estimate with a large batch size $b = 10$ with probability $p$, while with probability $1 - p$ it uses a smaller batch size $b' = 1$, whose gradient estimate is aggregated with the previous step's update.

To show the benefit of the variance reduction mechanism, we compare naive ZO-APMC approach ($p = 1$) discussed in after Proposition 1 with ZO-APMC with $p < 1$. We use the same inverse problem setting and analytical prior explained in Section 4.1. We note that we approximate FI accurately because our generated samples live in $\mathbb{R}^2$. In higher dimensions, this approximation quickly becomes unreliable and expensive. Importantly, in $\mathbb{R}^2$, ZO estimators do not suffer from the curse of dimensionality, which is a primary motivation for our variance-reduction mechanism. To better mimic high-dimensional behavior and induce higher variance in ZO estimates, we inject synthetic noise into the likelihood score estimates with standard deviation $\sigma_{\text{noise}} = 10$. Given a fixed budget of 6 score likelihood approximations on average, which corresponds to 12 forward model evaluations, we run ZO-APMC with different pairs of $(p, b) \in \{(1, 6), (0.4, 9), (0.2, 14), (0.1, 24)\}$. For each pair, we choose $b'$ such that $(1 - p)b' + pb \leq 6$ and the algorithm remains within the per-iteration budget. This ensures a fair comparison between ZO-APMC with $p < 1$ and the naive ZO-APMC ($p = 1$) approach. In addition, in Assumption 1, we mention that one can assume Laplace noise modeling instead of Gaussian modeling so we run our experiments for both type of noise modeling. We present the results in Fig. 9. They verify our theoretical results in Theorem 1. When $p = 1$, ZO-APMC converges to the largest point. This suggests that when $b$ is not large enough, the upper bound in Theorem 1 becomes suboptimal and it can be made optimal only by increasing $b$. In contrast, when $p < 1$, we can increase $b$ without changing the per-iteration budget. Doing so improves the convergence of the FI because ZO-APMC with different values of $p < 1$

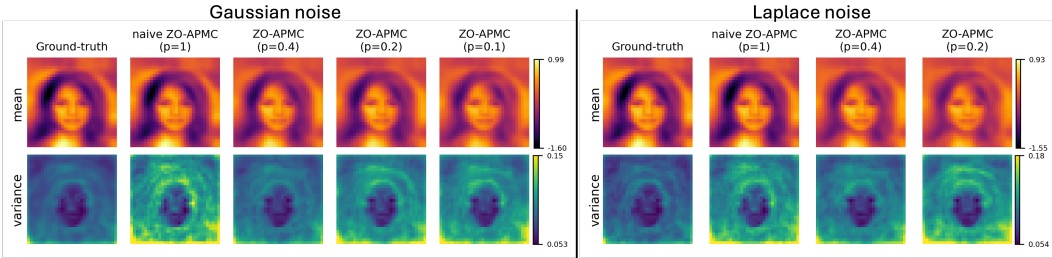

Figure 10: Comparison of naive ZO-APMC ($p = 1$) and ZO-APMC with $p < 1$ for estimating the ground truth posterior mean and variance statistics under Gaussian (left) and Laplace (right) measurement noise types. First row shows the posterior means and the second row shows the posterior variance. The colorbar for each statistic is shown at the end of its corresponding row. Each column shows the statistics estimated by ZO-APMC with specified probability $p$.

converge to smaller values. We further note that the convergence for $p = 0.1$ is worse than $p = 0.4$. This is consistent with Theorem 1 because as $p$ reduces, it increases the discretization error (second term) at the upper bound. We present a similar comparison for statistical validation in the subsequent section.

### A.4.2 STATISTICAL VALIDATION

The SGM used in this experiment is U-Net taken from (Nichol & Dhariwal, 2021) with some of its layers removed to process the $32 \times 32$ images, which are taken from CelebA (Liu et al., 2018) dataset. Each image is normalized to $[-1, 1]$ and downscaled to $32 \times 32$ pixels for simplicity. The forward operator is generated as random Gaussian matrix and for each test image, we inject a Gaussian noise with variance 0.01 as a measurement noise. We construct a bimodal distribution by selecting male and female images from the CelebA dataset and fitting a Gaussian mixture model (GMM) to the combined data. To ensure adequate separation, the two modes are shifted by $+1$ and $-1$. The SGM prior is then trained on samples drawn from this synthetic multimodal distribution. Because the synthetic Gaussian images lack the structural richness of natural images, the score network's results on this dataset should not be taken as representative of its performance on real-world data. For comparison, we compute the target modes and posterior statistics using the statistics derived from male and female images in the CelebA dataset.

In addition to the statistical validation experiments in Section 4.1, we compare naive ZO-APMC ($p = 1$) with ZO-APMC for $p < 1$ under Gaussian and Laplace noise modeling. We run each ZO-APMC algorithm for $N = 5000$ iterations for 1000 samples with different values of $(p, b) \in \{(1, 2), (0.4, 4), (0.2, 7), (0.1, 12)\}$ and $b'$ is selected such that the number of gradient approximations per-iteration satisfies $(1 - p)b' + pb \leq 2$. Because the Laplace likelihood and Gaussian prior are not a conjugate pair, closed-form posterior statistics are not available. Therefore, when modeling Laplace noise, we estimate the ground truth posterior statistics numerically by running standard Langevin Monte Carlo and get an empirical estimate of mean and variance. In this procedure, we use the likelihood score directly and do not apply any annealing schedule to the sampling parameters. We present the results in Fig. 10. Under Gaussian measurement noise, the naive ZO-APMC has significantly higher variance than the ground truth variance, which would require increasing the batch size and therefore the number of forward evaluations per iteration. Instead, our ZO-APMC with $p < 1$ reduces this variance by using a larger batch size $b$ together with a small $p$ as explained previously. Fig. 10 shows that choosing $p = 0.4$ reduces the sample variance significantly and it looks very similar to the ground truth variance. However, the variance reduction is not monotonic in $p$. For $p = 0.2$ and $p = 0.1$, the variance increases slightly, although it remains lower than the variance at $p = 1$. This behavior arises because, in Theorem 1, the discretization error (second term) grows as $p$ decreases and eventually becomes dominant, which prevents the samples from converging to the target statistics. We observe similar variance reduction for $p = 0.4$ under Laplace noise modeling in Fig. 10 and it increases when $p$ is decreased too much to $p = 0.2$. Lastly, across all the experiments, we keep the ZO smoothing parameter fixed $\mu$ to isolate its effect. This results in increasing bias as $p$ decreases, and we observe this effect at $p = 0.1$ in estimated posterior mean for

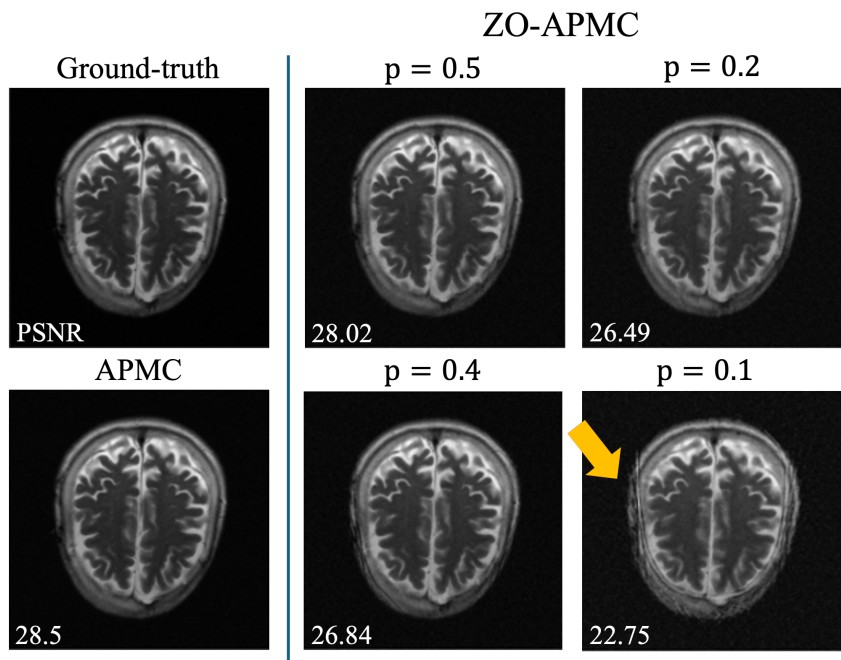

Figure 11: Comparison of the ground-truth brain MRI with APMC and ZO-APMC reconstructions for various probabilities $p \in \{0.1, 0.2, 0.4, 0.5\}$, using a large batch size of $b = 10^4$ and a small batch size of $b' = 10^3$. PSNR values for each reconstruction are displayed in the lower-left corner of the corresponding image.

both noise types. The effect is stronger under Laplace noise because, according to Theorem 1, the bias term scales with the smoothness of the log-likelihood. Note that $\|y - Ax\|_2^2$ is globally smooth with a smaller smoothness constant, whereas $\|y - Ax\|_1$ is smooth almost everywhere and has a larger effective smoothness constant. Thus, ZO-APMC with Laplace noise modeling has more bias but it can be mitigated by choosing smaller $\mu$.

## A.5 ADDITIONAL DETAILS FOR INVERSE PROBLEM EXPERIMENTS

### A.5.1 MRI EXPERIMENT DETAILS

We evaluate the reconstruction quality of the samples generated by ZO-APMC and other baselines methods by using peak signal to noise (PSNR) ratio, structural similarity index measure (SSIM), normalized root mean square (NRMSE), and standard deviation (SD). Given an estimate $\hat{x} \in \mathbb{R}^d$ and the ground truth $x_{GT} \in \mathbb{R}^d$, we define the error metrics as

$$\text{MSE}(\hat{x}, x_{GT}) = \frac{1}{d}\|\hat{x} - x_{GT}\|_2^2, \quad \text{NRMSE}(\hat{x}, x_{GT}) = \frac{\|\hat{x} - x_{GT}\|_2}{\|x_{GT}\|_2},$$

$$\text{PSNR}(\hat{x}, x_{GT}) = 10 \log_{10}\left(\frac{\max(x_{GT})^2}{\text{MSE}(\hat{x}, x_{GT})}\right).$$

where $d$ is the number of elements in $x$, and $\max$ denotes the maximum possible value of the signal (e.g., 1 for normalized data or 255 for 8-bit images).

**Ablation Study.** Among the inverse problems considered in this work, MRI reconstruction involves the largest image size ($256 \times 256$), which necessitates a larger batch size in our ZO estimator to accurately compute the forward-model gradient. To identify the optimal value of $p$, we subsample examples from the validation set of FastMRI (Zbontar et al., 1811) and evaluate reconstruction quality across different values, $p \in \{0.1, 0.2, 0.4, 0.5\}$, as illustrated in Fig. 11. As indicated by the orange arrow, reducing $p$ excessively while keeping $b$ fixed produces visible artifacts in the

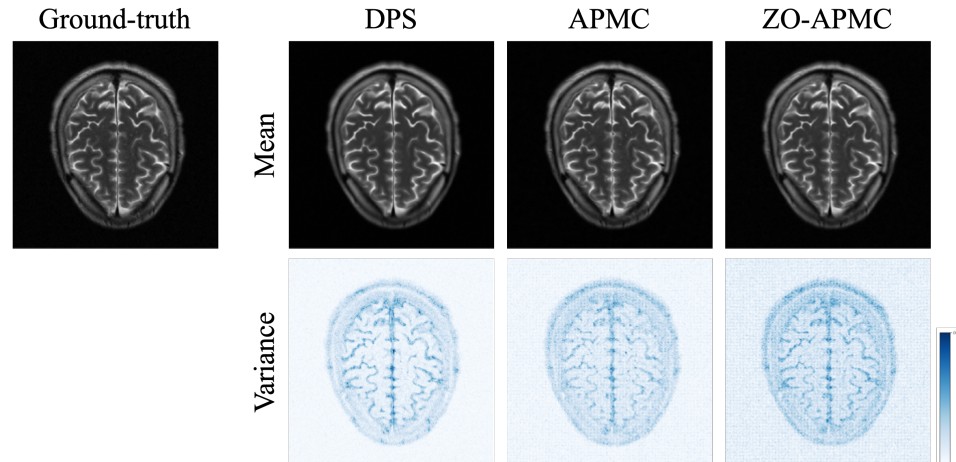

Figure 12: Comparison of the ground-truth brain MRI with reconstructions from ZO-APMC and the gradient-based approaches DPS and APMC. Each method generates 20 samples from the same measurements; the first row shows the mean reconstructions and the second row shows the corresponding variance maps. Owing to its variance-reduction mechanism, ZO-APMC produces variance maps comparable to those of the gradient-based algorithms despite relying on noisy evaluations of the forward model.

generated samples. ZO-APMC maintains reasonable reconstruction quality down to $p = 0.2$. Even when using a smaller batch size of $b' = 10^3$, an order of magnitude lower than $b$, in about half of the iterations on average, ZO-APMC maintains high reconstruction quality that is very close both visually and quantitatively to the reconstruction of APMC. As opposed to APMC, ZO-APMC achieves this without any gradient information and uses only forward model function evaluations. Because the performance gain beyond $p = 0.2$ is not significant and the gap between $p = 0.5$ and $p = 0.2$ can be further reduced by averaging multiple parallel outputs, we set $p = 0.2$ for our brain MRI inverse problem experiments.

Moreover, ZO estimators are widely recognized in the literature for exhibiting high variance in high-dimensional settings, as they rely on first-order approximations of the function along random directions. To evaluate our proposed variance-reduction mechanism, we compare the reconstructions of our method with DPS and APMC, which do not assume black-box setting and have access to gradients of the forward model. Results in Fig. 12 show that although our proposed method ZO-APMC assumes black-box setting and uses noisy forward model evaluations to approximate the gradient of the forward model, it has similar variance compared to DPS and APMC, which assumes access to the gradients, thanks to our proposed variance-reduction mechanism.

### A.5.2 BLACK-HOLE IMAGING EXPERIMENT DETAILS

In black-hole imaging, very long baseline interferometry (VLBI) uses an array of ground-based telescopes. Each telescope pair $(a, b)$ at time $t$ produces a complex visibility $V_t^{a,b}$. To mitigate atmospheric and thermal phase errors, visibilities are combined into noise-robust *closure* measurements (Chael et al., 2018): closure phases $\boldsymbol{y}_{t,(a,b,c)}^{\text{cph}}$ and log-closure amplitudes $\boldsymbol{y}_{t,(a,b,c,d)}^{\text{camp}}$. Following Sun & Bouman (2021); Zheng et al. (2025a), we use the following likelihood model:

$$\ell(\boldsymbol{y} \mid \boldsymbol{x}) = \sum_t \frac{\left\|\mathcal{A}_t^{\text{cph}}(\boldsymbol{x}) - \boldsymbol{y}_t^{\text{cph}}\right\|_2^2}{2\beta_{\text{cph}}^2} + \sum_t \frac{\left\|\mathcal{A}_t^{\text{camp}}(\boldsymbol{x}) - \boldsymbol{y}_t^{\text{camp}}\right\|_2^2}{2\beta_{\text{camp}}^2} + \frac{\rho}{2}\left\|\sum_i x_i - y^{\text{flux}}\right\|_2^2. \quad (138)$$

Here, $\mathcal{A}_t^{\text{cph}}$ and $\mathcal{A}_t^{\text{camp}}$ map an image $\boldsymbol{x}$ to predicted closure phases and log-closure amplitudes, respectively; $\beta_{\text{cph}}$ and $\beta_{\text{camp}}$ are instrument-specific noise scales. The first two sums act as chi-squared penalties for the closure measurements, while the final term enforces the total-flux constraint with weight $\rho$ and target flux $y^{\text{flux}}$. For our experiments, we adopted the dataset, pre-trained SGM prior, forward model implementation, and baseline methods provided by Zheng et al. (2025b). For

EnKG, we adopt the hyperparameters recommended by Zheng et al. (2025a), and for the baseline methods we use the hyperparameters provided by Zheng et al. (2025b).

### A.5.3 NAVIER-STOKES EQUATION EXPERIMENT DETAILS

In our experiments, we study the two-dimensional Navier–Stokes equations for a viscous, incompressible fluid in vorticity form on a torus. Let $u \in C([0, T]; H^r_{\text{per}}((0, 2\pi)^2, \mathbb{R}^2))$ for any $r > 0$ denote the velocity field, and let $w = \nabla \times u$ be the vorticity. The initial vorticity is $w_0 \in L^2_{\text{per}}((0, 2\pi)^2; \mathbb{R})$, the viscosity coefficient is $\nu \in \mathbb{R}_+$, and the forcing term is $f \in L^2_{\text{per}}((0, 2\pi)^2; \mathbb{R})$. The solution operator $\mathcal{G}$ maps the initial vorticity to the vorticity at time $T$, i.e. $\mathcal{G} : w_0 \mapsto w_T$. In our experiments, we implement $\mathcal{G}$ using a pseudo-spectral solver following He & Sun (2007):

$$\partial_t w(x, t) + u(x, t) \cdot \nabla w(x, t) = \nu \Delta w(x, t) + f(x), \qquad x \in (0, 2\pi)^2, \, t \in (0, T], \qquad (139)$$

$$\nabla \cdot u(x, t) = 0, \qquad x \in (0, 2\pi)^2, \, t \in (0, T], \qquad (140)$$

$$w(x, 0) = w_0(x), \qquad x \in (0, 2\pi)^2. \qquad (141)$$

The task is to infer the initial vorticity field from noisy and sparsely observed vorticity data at time $T = 1$. Since Eq. (21) admits no closed-form solution, the corresponding derivative of the solution operator is also unavailable. Furthermore, the computation of accurate numerical derivatives via automatic differentiation through the solve is challenging since the extensive computation graph can span thousands of discrete time steps.

We follow the approach in Zheng et al. (2025a;b) and first solve the equation up to time $T = 5$ starting from random Gaussian initial conditions, which are highly nontrivial due to the nonlinearity of the Navier–Stokes equations. We use the SGM-prior, which was pre-trained over 20,000 vorticity fields, and use the test set consisting of 10 samples from InverseBench. For EnKG, we use the hyperparameters recommended by Zheng et al. (2025a), and for the baseline methods we adopt the hyperparameters provided by Zheng et al. (2025b). Quantitative results are presented in Fig. 3. Our method demonstrates a performance comparable to most black-box posterior samplers, while distinctively providing rigorous theoretical guarantees of convergence to the target posterior guarantees, which is not established for the baseline methods.

Table 3: Quantitative results for the Navier–Stokes equation benchmark. For each case, the best-performing method is shown in **bold**. Baseline results are taken from Zheng et al. (2025a).

|  | NRMSE ($\sigma_{\text{noise}} = 0$)↓ | NRMSE ($\sigma_{\text{noise}} = 1$)↓ | NRMSE ($\sigma_{\text{noise}} = 2$)↓ |
|---|---|---|---|
| Forward-GSG | 1.687 | 1.612 | 1.454 |
| Central-GSG | 2.203 | 2.117 | 1.746 |
| SCG | 0.908 | 0.928 | 0.966 |
| DPG | 0.325 | 0.408 | 0.466 |
| EnKG | **0.120** | **0.191** | **0.294** |
| ZO-APMC (Ours) | 0.459 | 0.463 | 0.472 |

