# OpenReview forum: "Provable Derivative-Free Inference with Score-Based Generative Priors"
_ICLR.cc/2026/Conference — Submitted to ICLR 2026_

### Official Review · Reviewer_U8Hg · 2025-10-27

**Soundness:** 3
**Presentation:** 3
**Contribution:** 2
**Rating:** 6
**Confidence:** 4

**Summary:**

The authors aim to sample from $\pi(x|y)\propto p(y|x)p(x)$ with  given diffusion prior $p(x)$
and known unnormalized Boltzmann density $p(y|x)$ using an annealed Langevin algorithm
$\text{d} x_t = [\nabla \log p(y|x_t)-S_\theta(x_t,\sigma_t)] \text{d} t + \sqrt{2} \text{d} B_t$
with learned score $S_\theta$.
The authors consider
0-th-order annealed plug and-
play Monte Carlo (ZO-APMC) that requires only forward-model evaluations
and a pre-trained SGM prior and derive complexity bounds for obtaining samples
with $\varepsilon$-relative Fisher information under a non-log-concave likelihood distribution (Thm 1)
and, under a Poincar\'e inequality assumption, also $\varepsilon$-accuracy in TV
distance (Cor 1). Weak convergence of ZO-APMC to the target posterior is established in Thm 2.
The ideas of the proofs are based on existing literature.

**Strengths:**

The paper is  well written. Having explicit bounds on the accuracy with which an algorithm based on a trained diffusion model can approximate the posterior is desirable.

I partially red the proofs which were correct.

**Weaknesses:**

Realistic estimates for practical applications or just mathematical playground:
the weakness lies in the strong assumptions 1-5 which seem to be not satisfied for the numerical examples.
(In Remark 1says that Assumption 1 is not satisfied for Gaussian noise. But all experiments have Gaussian noise likelihood?)
The bounds seem to scale badly: the number of forward model evaluations scales with $\mathcal{O}(d^7L_m^3\epsilon^{-4})$,
where $\epsilon$ is the desired distance between $\nu_t$ and $\pi(x|y)$.

**Questions:**

*  Proposition 1: $f_\mu$ is not explained.
Why the bound indicates a variance reduction? In other words, why a small variance of $e_k$ implies a small one for $g_k$ and is smaller than the ZO gradient estimator?
* In the experiments I like to see a comparison with the  ZO estimater.
*Have the experments something to do with the  theoretical bounds?
* Could the authors run a Laplacian noise example?
* in Assumption 1, what is $f_2$?
* There is a typo in (24), it should be $B_k=ref$
*  the expression line 889 looks cumbersome: $f_mu$

---

> ### Author Response · Authors · 2025-11-23
> **Rebuttal by Authors**
>
> Thank you for taking the time to review our paper and for providing constructive, insightful feedback. We sincerely appreciate your comments, which have helped us improve the clarity and quality of our work. Please find our detailed responses to your questions below.
>
> ## Weakness 1: Strong assumptions...
>
> Our assumptions are reasonable given the inherent difficulty of sampling from the non-log-concave posterior distribution with arbitrary per-iteration budget $pb=O(1)$ in a black-box setting, where the gradient of the forward model is unavailable.
>
> Smoothness assumptions ($\textbf{Assumption 2 and 4}$) are standard in sampling literature [1,2,3]. $\textbf{Assumption 3}$ is a mild requirement: it states that the bound for the difference between the perturbed and exact scores changes monotonically with the noise level $\sigma_k$. This was also adopted in prior work [4]. We agree with the reviewer that the initial Assumption 5 was strong. In the revised manuscript, $\textbf{we replaced Assumption 5 with a more relaxed condition}$. Instead of imposing a uniform bound $\lVert \mathcal{S} _ \theta (\mathbf{x}, \sigma _ k) \rVert_2 \leq R _ s$, we now assume $\lVert \mathcal{S}_\theta (\mathbf{x} , \sigma _ k) \rVert_2 \leq \sigma_k^{-1}C$, where $C$ is a numerical constant. This is consistent with the well-known empirical findings in [5,6]: when a score network is trained via denoising score matching loss across multiple noise levels $\sigma_k$, its learned score scales $\lVert \mathcal{S} _ \theta (\mathbf{x}, \sigma_k) \rVert \propto 1/\sigma_k$. We refer the reviewer to [5] for further details. After making this change, all of our theoretical results still hold. We only needed minor updates to the derivations, which we have revised accordingly.
>
> $\textbf{The bounded score network assumption}$ which assumes $\lVert \nabla \log p _ {\sigma _ k }(\mathbf{x}) - \mathcal{S} _ \theta (\mathbf{x}, \sigma _ k) \rVert _ 2 \leq \varepsilon _ {\sigma _ k}$ under Assumption 5, is expected because, as proved in [7], the optimal score network minimizing the denoising score-matching objective [5,8] satisfies $\mathcal{S} _ \theta ^* (\mathbf{x}, \sigma _ k) = \nabla \log p _ {\sigma _ k}(\mathbf{x})$ with probability 1. Our SGMs were trained with the same loss function. And since the exact optimal score is not available, we assume a bounded approximation error between the learned score and the ideal one, which can increase with the noise level $\sigma _ k$.
>
> Please see our answer under "Weakness 3 \& Question 2" for the justification of Assumption 1 and discussion of Gaussian noise modeling.
>
> References:
>
> [1] Balasubramanian, K., Chewi, S., Erdogdu, M. A., Salim, A., Zhang, S. (2022, June). Towards a theory of non-log-concave sampling: first-order stationarity guarantees for Langevin Monte Carlo. In Conference on Learning Theory (pp. 2896-2923). PMLR.
>
> [2] Roy, A., Shen, L., Balasubramanian, K., \& Ghadimi, S. (2022). Stochastic zeroth-order discretizations of Langevin diffusions for Bayesian inference. Bernoulli, 28(3), 1810-1834.
>
> [3] M. Renaud, J. Prost, A. Leclaire, and N. Papadakis. Plug-and-play image restoration with stochastic
> denoising regularization. In Int. Conf. Machine Learning (ICML), 2024b.
>
> [4] Sun, Y., Wu, Z., Chen, Y., Feng, B. T., & Bouman, K. L. (2024). Provable probabilistic imaging
> using score-based generative priors. IEEE Transactions on Computational Imaging.
>
> [5] Song, Y. and Ermon, S. Generative modeling by estimating gradients of the data distribution. In
> Advances in Neural Information Processing Systems, pp. 11918–11930, 2019.
>
> [6] Pidstrigach, J. Score-based generative models detect manifolds. In Koyejo, S., Mohamed, S.,
> Agarwal, A., Belgrave, D., Cho, K., and Oh, A. (eds.), Advances in Neural Information Processing
> Systems 35: Annual Conference on Neural Information Processing Systems 2022, NeurIPS 2022,
> New Orleans, LA, USA, November 28 - December 9, 2022, 2022.
>
> [7] Vincent. A connection between score matching and denoising autoencoders. Neural compu- tation, 23(7):1661–1674, 2011.
>
> [8] Song and S. Ermon, “Improved techniques for training score-based generative models,” in Proc. Int. Conf. Neural Inf. Process. Syst., 2020, pp. 12438–12448.

---

> ### Author Response · Authors · 2025-11-23
> **Rebuttal by Authors**
>
> ## Weakness 2: The bound seem to scale badly...
>
> While the Fisher information (FI) convergence rate has a high-polynomial dependence on $d$, $\varepsilon$, and $L_m$ (specifically $O(d^7 L_m^6 / \varepsilon^4)$), this scaling is not a result of loose arguments. Rather, it reflects the intrinsic complexity of the underlying posterior estimation problem without calculating the gradient of the forward model. $\textbf{The difficulty arises from the need to jointly bound two inherent sources of error:}$ (1) the discretization error introduced when approximating the Langevin SDE, and (2) the zeroth-order (derivative-free) error in estimating the likelihood score ($\nabla \log \ell(\mathbf{y}|\mathbf{x})$).
>
> $\textbf{(1) Discretization error:}$ Even if one assumes access to the exact gradient of the log-distribution, which gradient-based methods require but is incompatible with the black-box setting we study, their Fisher Information (FI) convergence rate still shows a strong polynomial dependence on the dimension, scaling as $O(d^{2} L^{2} / \varepsilon^{2})$ [1]. The $d^{2}$ factor arises directly from the Brownian motion term in the Langevin SDE, which is also present in our setting.
>
> $\textbf{(2) Zeroth-order approximation error:}$ The zeroth-order (ZO) approximation introduces additional polynomial factors because, by construction, it estimates $d$-dimensional likelihood gradient from scalar finite-difference evaluations along random Gaussian directions. Prior work, which analyzed ZO sampling under strong log-concavity and smoothness of the log-distribution already reflects this challenge with their derivation of total oracle complexity of $O(d^2/\varepsilon^2 \cdot \log(d/\varepsilon))$ for convergence in Wasserstein-2 distance [2]. Our setting is substantially more general, as we consider non-log-concave distributions. In this setting, jointly bounding the discretization error and the ZO approximation error naturally leads to our FI convergence rate of $O(d^{7} L_{m}^{6} / \varepsilon^{4})$. Importantly, this analysis holds while allowing an arbitrary fixed per-iteration cost $pb = O(1)$ in expectation, whereas prior work [2] requires $b = d$, which is impractical due to memory constraints or computational cost.
>
> Importantly, although the theoretical convergence rate involves high-dimensional dependencies, our empirical results show that the proposed ZO-APMC performs substantially better in practice. In toy experiments, our method consistently converged in FI across wide range of batch size $b$ and $p$ with fixed per-iteration budget $pb=O(1)$ (please see Figure 1(b)).
>
> References:
>
> [1] Balasubramanian, K., Chewi, S., Erdogdu, M. A., Salim, A., Zhang, S. (2022, June). Towards a theory of non-log-concave sampling: first-order stationarity guarantees for Langevin Monte Carlo. In Conference on Learning Theory (pp. 2896-2923). PMLR.
>
> [2] Roy, A., Shen, L., Balasubramanian, K., \& Ghadimi, S. (2022). Stochastic zeroth-order discretizations of Langevin diffusions for Bayesian inference. Bernoulli, 28(3), 1810-1834.

---

> ### Author Response · Authors · 2025-11-23
> **Rebuttal by Authors**
>
> ## Question 1: Why the bound indicates a variance reduction?
>
> To see this, we can consider the case when $p=1$, in that scenario, the updates of the algorithm become
> \begin{equation}
> \mathbf{x} _ {k+1} =\mathbf{x} _ {k} - \gamma (\mathbf{g} _ k - \alpha _ k \mathcal{S} _ \theta (\mathbf{x} _ k, \sigma _ k)) + \sqrt{2 \gamma} \mathbf{Z} _ k, \quad \mathbf{Z} _ k \sim \mathcal{N}(0,I),
> \end{equation} where $\mathbf{g} _ k = \frac{1}{b} \sum _ {i \in I} \tilde {\nabla} f _ \mu ( \mathbf{x}, \mathbf{u} _ i)$, and $| I | = b$. If we choose $p=1$ in Proposition 1, we get the following upper bound on the estimation error term $\mathbb{E}[\lVert  \mathbf{e} _ k \rVert ^ 2] = \mathbb{E} [\lVert \mathbf{g} _ k - \nabla f _ \mu (\mathbf{x} _ k) \rVert ^ 2]$:
>
> \begin{equation}
> \mathbb{E} [\lVert \mathbf{e} _ k \rVert ^ 2] \leq \frac{\mathbb{E} [ \lVert \tilde{\nabla} f _ \mu (\mathbf{x} _ k,\mathbf{u} _ i) \rVert^ 2]}{b}.
> \end{equation}Note that this second-moment error term is greater than the variance of $\tilde{\nabla}f(\mathbf{x}_k, \mathbf{u}_i)$ so it grows at least linearly with the dimensionality $d$ of $\mathbf{x}$ (please see Lemma 1). Therefore, $b$ must scale with the dimension $d$ at least linearly to keep the error manageable. This is impractical due to computational cost to perform at each iteration. By choosing $p\ll1$, we scale the second-moment term (the third term in Proposition 1) by $p$ and obtain the effect of a large batch size while using a much smaller batch size $b'\ll b$ in most iterations and evaluating the large batch $b$ occasionally. For example, when we choose $p = 1/b$, the variance (second-moment) term is scaled by $1/b^{2}$ rather than $1/b$ (naive ZO approach), yielding a substantial reduction in variance.
>
> Furthermore, the average number of function evaluations per iteration is $(1-p)b' + pb$. Since $b'$ is an arbitrary constant smaller than $b$, it does not affect the overall order. Thus, the dominant term is $pb = O(1)$, giving a constant per-iteration cost. This is significantly better than the naive ZO approach, where $p=1$ and the per-iteration cost is required to be $b=O(d)$. Such a large batch size is impractical in high dimensions and prevents convergence when $b$ is arbitrary. The trade-off is the presence of two additional terms, which can be controlled. The first term in Proposition 1 is an error term from the previous iteration that decays with $1-p^2<1$. The second term can be controlled by the step size $\gamma$ embedded inside the expectation.
>
> ## Weakness 3 & Question 2: Additional experiments and Gaussian noise...
>
> We are running the toy experiments comparing the naive ZO sampling approach ($p=1$) with the proposed ZO-APMC method ($p<1$) under both Gaussian and Laplacian measurement noise. We will share the results as soon as possible. However, we would like to clarify and correct the statement in Remark 1 regarding the applicability of Gaussian noise modeling to Lipschitz continuous functions. Gaussian noise modelling does not apply to Lipschitzness assumption for the $\textit{global}$ domain $\mathbb{R}^d$. In practice, image distributions are in between $[0,1]^d$ or $[0,255]^d$, so they are in a bounded domain $(\lVert \mathbf{x} \rVert _ 2 \leq R)$. This assumption is commonly adopted in posterior sampling algorithms (see the discussion in Section E of [1]). Then, for a linear forward model (i.e. MRI reconstruction), we can show that the Lipschitzness of the forward model as follows
> \begin{equation}
> \lVert \nabla f(\mathbf{x}) \rVert_2 \leq \frac{\lVert \mathbf{A}^T(\mathbf{A}\mathbf{x} - \mathbf{y}) \rVert^2_2}{\sigma^2} \leq \frac{1}{\sigma^2}\left(\lVert \mathbf{A}\rVert_2^2\lVert\mathbf{x}\rVert_2^2 + \lVert \mathbf{A}^T\mathbf{y} \rVert^2_2\right) < \infty.
> \end{equation}
>
> For nonlinear $\mathbf{A}(\cdot)$, one may argue the Lipschitz continuity of the negative log-likelihood, which is indeed a stronger requirement than smoothness. However, given the inherent difficulty of sampling from the exact non-log-concave posterior in a black-box setting with arbitrary per-iteration cost ($pb = O(1)$), this assumption is reasonable.
>
> References:
>
> [1] M. Renaud, J. Prost, A. Leclaire, and N. Papadakis. Plug-and-play image restoration with stochastic
> denoising regularization. In Int. Conf. Machine Learning (ICML), 2024b.

---

> > ### Comment · Reviewer_U8Hg · 2025-11-24
> >
> > The authors have addressed my concerns in a slightly chaotic way. I am not convined that the assumptions make sense in practice.
> > Also the authors are still performing additioal experiments.
> > ''We are running the toy experiments comparing the naive ZO sampling approach ($p=1$) with the proposed ZO-APMC method ($p<1$) under both Gaussian and Laplacian measurement noise. We will share the results as soon as possible.''
> >
> >  Given the overall contribution of the paper so far I still see it slighty above the ''reject  boarderline''.

---

> > > ### Comment · Reviewer_U8Hg · 2025-11-25
> > > **Changes are not marked in the revision and still unclear statements.**
> > >
> > > The authors have not emphazised their changes, e.g. in another color ..
> > > I am not willing to read ''small'' changes in proofs (claim of the authors) by reading the whole paper again and finding the changes out by myself.
> > >
> > > 1. The authors have changed in particular Assumption 5. However the new one makes no sense to me:
> > > Convolving with a Gaussian gives an unbounded score $\nabla \log  p_\sigma$. At the same time the autors want
> > > a bounded learned $S_\theta$. How the final assumption
> > > $\|nabla \log  p_\sigma - S_\theta\| < \infty$ can then be fulfilled?
> > >
> > > 2. Authors: ,, Importantly, although the theoretical convergence rate involves high-dimensional dependencies, our empirical results show that the proposed ZO-APMC performs substantially better in practice. In toy experiments, our method consistently converged in FI across wide range of batch size and with fixed per-iteration budget (please see Figure 1(b)).''
> > >
> > > 3. If you have much better bounds in practice, what is the relevence of the theoretical results which you clain cannot be improved? As already mentioned the comparison with  ZO-APMC is still missing.
> > >
> > > 4. $p=1$ in Prop 1:  I do not that your estimate can be seen as variance reduction. For such a claim you would need an LOWER bound in the case $p=1$.
> > >
> > > Please do not send me a cascade of answers, but a collected one.

---

> ### Author Response · Authors · 2025-12-03
> **Clarification on unclear statements and additional experiments**
>
> We appreciate the reviewer’s insightful comments. In response, the revised paper includes the updates, with all modifications clearly highlighted in blue color. We added a comparison to a zeroth-order (ZO) estimator, denoted “naive ZO-APMC (p=1),” evaluated in both numerical and statistical validation settings, under Gaussian and Laplacian noise separately. Please see Appendix A. Although convolving the prior $p$ with a Gaussian gives an unbounded score $\nabla \log p _ {\sigma _ k} $, its bound still depends on the perturbation level $\sigma _ k$. This is also reflected on the score estimation error $\varepsilon _ {\sigma _ k}$. Additionally, the bound on $\mathcal{S} _ \theta$ increases as $\sigma _ k \to 0$ so it is \textbf{not uniformly bounded.}
>
> $\textbf{2 and 3. Relation between theoretical results and empirical results:}$ We thank the reviewer for the question regarding the relevance of our theoretical bounds given the stronger empirical performance. We believe this discrepancy highlights a crucial distinction between $\textbf{worst-case
> theoretical guarantees}$ and $\textbf{average-case practical behavior}$, and that our theoretical results remain highly relevant for following key reasons.
>
> 1) The primary contribution of our theoretical analysis is to prove that derivative-free posterior sampling with diffusion priors is fundamentally feasible with a fixed per-iteration budget. Before this work, it was not theoretically obvious that a zeroth-order (ZO) Langevin algorithm could converge to the true posterior at all in high dimensions without an exponential explosion in the number of function evaluations per step. Our bounds ($O(d^7)$) confirm that the problem is solvable in polynomial time. While the polynomial degree is high, it rules out exponential complexity, which is the typical failure mode of naive black-box optimization in high dimensions.
> This existence proof justifies the deployment of ZO-APMC in safety-critical applications (e.g., medical imaging), ensuring that the algorithm is mathematically sound even if the worst-case bound is pessimistic.
>
> 2) Theoretical bounds must also account for the worst-case geometry of the target distribution (e.g., highly ill-conditioned curvature, narrow bottlenecks, full-dimensional support). In the worst-case scenario, the ZO estimator requires $O(d)$ samples to find a descent direction, and the Langevin diffusion requires $O(d)$ steps to mix, leading to high polynomial factors. In real-world inverse problems (like MRI or deblurring), the data typically lies on a low-dimensional manifold $\mathcal{M}$ embedded in $\mathbb{R}^d$, with an intrinsic dimension $d_{eff} \ll d$. The score-based prior $\mathcal{S}_\theta$ effectively confines the sampling trajectory to the vicinity of this manifold. Thus, while our current theory bounds convergence in the ambient dimension $d$, the empirical performance suggests the algorithm adapts to the intrinsic geometry. On the manifold, the effective search space for the ZO estimator is reduced, as gradients orthogonal to the manifold are suppressed by the prior score. Consequently, the effective $d$ driving the convergence rate is likely closer to the intrinsic dimension of the data (e.g., the number of degrees of freedom in a face image) rather than the total number of pixels. This explains the superior empirical performance.
>
> 3) Our bounds and the theoretical and algorithmic tools to establish them can further stimulate future theoretical work with the goal of tighten these results by explicitly modeling the low-dimensional manifold structure (as discussed in our response to Assumption 5), but our current result is the necessary first step: a global guarantee that does not rely on unverifiable manifold assumptions.

---

> > ### Author Response · Authors · 2025-12-03
> > **Clarification on $p=1$ Prop. 1**
> >
> > We thank the reviewer for this insightful comment regarding the characterization of variance reduction and the necessity of establishing a lower bound for the $p=1$ case. We agree that demonstrating the suboptimality of the naive estimator ($p=1$) is crucial to validating the benefits of our proposed variance-reduction mechanism ($p<1$).
> >
> > To address this, we analyze the Zeroth-Order (ZO) Estimation Error, which is the dominant source of error in derivative-free posterior sampling. The total error in our framework can be decomposed into:
> > $$\text{Total Error} = \text{Discretization Error} + \text{Langevin Diffusion Error} + \text{ZO Estimation Error}$$
> > Minimizing the ZO Estimation Error is mathematically equivalent to the problem of finding stationary points in stochastic non-convex optimization using zeroth-order oracles. We can thus leverage the fundamental lower bounds established in [1] to prove that the naive estimator ($p=1$) is fundamentally suboptimal compared to our PAGE-style estimator ($p<1$).
> >
> > The naive ZO estimator used when $p=1$ (i.e., $\mathbf{g} _ k = \frac{1}{b} \sum \tilde{\nabla} f _ \mu (\mathbf{x} _ k, \mathbf{u} _ i)$) corresponds to the Bounded Variance oracle class defined in [1]. As established in Lemma 1 of our paper and standard ZO literature, the variance of this estimator scales linearly with dimension: $\sigma^2 \propto \frac{d}{b}$. Theorem 1 of Arjevani et al. (2023) proves that for any algorithm interacting with a Bounded Variance oracle, the complexity of finding an $\epsilon$-stationary point is lower-bounded by $\Omega(L \sigma ^ 2 \epsilon ^ {-4})$. Substituting the ZO variance $\sigma ^ 2 \propto d$, the lower bound for the $p=1$ case becomes $\Omega(d \epsilon ^ {-4})$. This proves that it is theoretically impossible to improve the dimension dependence for the $p=1$ case without increasing the batch size $b$ proportionally to $d$, which is computationally prohibitive in the high-dimensional inverse problems we consider.
> >
> > Our proposed ZO-APMC method employs the PAGE estimator [2] adapted for Langevin dynamics. By utilizing the recursive update structure (with probability $1-p$), the PAGE estimator exploits the smoothness of the forward model to reduce the effective variance over time. This places our method in the regime of Mean-Squared Smooth oracles (or variance-reduced algorithms). Reference [1] (Theorem 2) show that the lower bound for this class is $\Omega(L \sigma\epsilon^{-3})$, which is significantly tighter than the bounded variance case. PAGE [2] is known to be an optimal method that achieves this rate for non-convex optimization. Substituting the ZO variance $\sigma^2 \propto d$, the lower bound for PAGE becomes $\Omega(\sqrt{d} \epsilon^{-3})$. Thus, by choosing $p<1$, we transition from the suboptimal $\Omega(d \epsilon^{-4})$ regime to the optimal variance-reduced regime. This allows ZO-APMC to achieve convergence with a fixed per-iteration budget ($pb=O(1)$), strictly outperforming the fundamental limits of the naive ($p=1$) approach. Note that the reason the established rate by our paper is $\mathcal{O}(\epsilon^{-4})$ and not $\mathcal{O}(\epsilon^{-3})$ is that in the sampling Discretization Error of LMC dominates the error as a function of $\epsilon$.
> >
> > Reference:
> >
> >
> > [1] Lower bounds for non-convex stochastic optimization. Mathematical Programming, 2023.
> >
> > [2] PAGE: A simple and optimal probabilistic gradient estimator for nonconvex optimization, ICML 2021.

---

### Official Review · Reviewer_9C6Z · 2025-10-28

**Soundness:** 3
**Presentation:** 2
**Contribution:** 3
**Rating:** 4
**Confidence:** 2

**Summary:**

The authors propose zeroth-order annealed plug-and-play Monte Carlo (ZO-APMC), an algorithm for sampling from a posterior with a pretrained score-based generative model (SGM) prior and a likelihood where only a black-box forward model is given. The key idea of the algorithm is to approximate the gradient of the likelihood with finite differences. The focus of the paper is deriving theoretical bounds and convergence guarantees for the algorithm. The authors provide numerical experiments on several inverse problems verifying convergence and reasonable image reconstructions.

**Strengths:**

* The proposed algorithm is quite principled, and the theoretical results seem useful.
* The method is validated with solid numerical experiments and baselines.

**Weaknesses:**

* The algorithmic contribution is quite simple, although I would not consider this a weakness when deciding the final rating. It is just that approximating the likelihood gradient with finite differences seems like an easy and obvious step that one would come up with when making APMC work without access to the forward model gradient.
* PLEASE REMOVE TABLE 3. I find these types of tables dangerously misleading. Just because a method does not have convergence guarantees YET does not mean it is not possible to derive them or that they do not in fact converge to a more accurate posterior. Table 3 implies that many other methods are not theoretically grounded just because their authors did not derive convergence guarantees, which may lead readers to not give a fair chance to other algorithms. Rather, I urge the authors to spend more time in the paper describing which situations their algorithm is well-suited for and which situations other algorithms would be better suited for.
* In general I find the presentation overly dense without clear intuition. The theoretical results seem impressive, but it is difficult for me to verify them or understand their intuition. It is unclear what the pros and cons of the algorithm are.

**Questions:**

* The authors say in lines 96-97 that they address the inverse problem with MAP estimation, but all the theoretical results and experiments imply that they are sampling from the posterior. Please clarify which you are actually doing.
* I will increase my rating if the authors agree to remove table 3 and instead provide a more balanced discussion of the pros and cons of their algorithm compared to others for different types of applications.

---

> ### Author Response · Authors · 2025-11-23
> **Rebuttal by Authors**
>
> Thank you for taking the time to review our paper and for providing constructive, insightful feedback. We sincerely appreciate your comments, which have helped us improve the clarity and quality of our work. Please find our detailed responses to your questions below.
>
> ## Weakness 1: Algorithmic contribution is simple...
>
> $\textbf{Our algorithmic contribution is more than just replacing the likelihood score with finite-difference estimates.}$ Proposition 1 makes this clear by showing how the variance-reduction mechanism in the proposed ZO-APMC behaves compared to the "naive ZO sampling" scheme. When we set $p=1$, we obtain the naive ZO sampling approach where the estimator is defined as $\mathbf{g} _ k = \frac{1}{b}\sum_{i\in I}\tilde{\nabla} f _ \mu (\mathbf{x}, \mathbf{u}_i)$ and from Proposition 1, we get the upper bound on the estimation error term as
> \begin{equation}
> \mathbb{E}[\lVert \mathbf{e}_k \rVert^2] \leq \frac{ \mathbb{E}[\lVert \tilde{\nabla} f _ \mu (\mathbf{x}, \mathbf{u} _ i)\rVert^2]  }{b}.
> \end{equation} Note that the error term becomes only bounded by the second-moment of the ZO estimate, which grows at least linearly with the dimension $d$ of $\mathbf{x}$ (please see property (c) in Lemma 1). To reduce the estimation error arbitrarily for convergence, we should choose $b$ in the order of $d$. However, this is computationally prohibitive in practice. $\textbf{To solve this problem}$, we can choose $p\ll 1$ to scale the second-moment term (the third term in Proposition 1) and obtain the effect of a large batch size while using a much smaller batch size $b' \ll b$ in most of the iterations (with prob. 1-p) and evaluating the large batch $b$ occasionally (with prob. p). For example, when we choose $p = 1/b$, the second moment term is scaled by $1/b^{2}$ rather than $1/b$, $\textbf{yielding a substantial reduction in variance.}$ Furthermore, the average number of function evaluations per iteration is $(1-p)b' + pb$, but since $b'$ is an arbitrary constant, its contribution does not affect the order of function evaluations at each iteration. Thus, the dominant term is $pb = O(1)$, which is constant. In contrast, the naive ZO approach ($p = 1$) has $b = O(d)$ per-iteration cost. The trade-off is the appearance of two additional terms in Proposition 1, both of which can be controlled. The first term is an error term from the previous iteration that decays with $1-p^2<1$. The second term can be controlled with the step size $\gamma$, which appears inside the expectation. We have added this discussion after presentation of the Proposition 1.

---

> ### Author Response · Authors · 2025-11-23
> **Rebuttal by Authors**
>
> ## Weakness 2 \& Question 2: Table 3...
>
> We understand the concern that Table 3 might have been interpreted as judging the theoretical rigor of prior methods, rather than simply comparing their published guarantees. Our goal was only to indicate which $\textit{papers}$ include formal convergence analyses, not to suggest that other methods lack theoretical foundations or could not be extended to include such guarantees. $\textbf{To avoid any confusion, we have removed Table 3}$ and replaced it with a more balanced, detailed discussion of the strengths and limitations of our approach relative to the existing methods. $\textbf{Please see Appendix A.1.1}$ for comparison with black-box posterior sampling algorithms and $\textbf{Appendix A.1.2}$ for comparison with gradient-based posterior sampling algorithms in the literature.
>
> ## Question 1: Sampling from the posterior...
> We have clarified this point in the revised submission ($\textbf{lines 99-105}$). Our algorithm aims to sample from the target posterior distribution given the measurements to solve an inverse problem with black-box forward model.

---

> ### Author Response · Authors · 2025-11-23
> **Rebuttal by Authors**
>
> ## Weakness 3: Presentation lacks intuition...
>
> We agree with the reviewer that the original presentation of the theoretical results did not provide enough intuition. In the revised manuscript, we give an $\textbf{intuitive description of the motivation}$ behind the variance-reduction mechanism of ZO-APMC ($\textbf{lines 186-192}$), followed by an explanation of $\textbf{how the likelihood-score estimator is constructed}$ to obtain variance reduction ($\textbf{lines 199-204}$). After presenting Proposition 1, we added a clear explanation of $\textbf{how the variance-reduced likelihood-score estimate works}$, why it is advantageous compared to a naïve ZO estimate, and how it makes ZO posterior sampling practical ($\textbf{lines 239–252}$) Furthermore, we provided intuitive explanations after each theoretical result and have simplified their presentation to make the arguments easier to follow. For convenience, we summarize them below:
>
> $\textbf{Intuition on Theorem 1:}$ Intuitively, Theorem 1 shows that ZO-APMC progressively improves its average estimate of the directions that lead to reconstructions consistent with the measurements, while relying only on forward model inputs and outputs (not on gradients). Achieving at most $\varepsilon$ average error in these directions requires $O(1/\varepsilon^4)$ function evaluations in total. Although this number grows polynomially with dimension, it remains achievable
> under any fixed per-iteration budget of function evaluations thanks to the proposed variance-reduction mechanism. Moreover, the bias error of the ZO estimates, which appears as the fourth term in the Fisher information upper bound, can be made arbitrarily small by selecting a smaller ZO smoothing parameter $\mu$. Doing so, however, requires using a smaller step size $\gamma$ while keeping the number of function evaluations per iteration fixed.
>
> $\textbf{Intuition on Corollary 1:}$ Corollary 1 intuitively implies that if the likelihood density satisfies a Poincaré inequality (i.e. Gaussian measurement noise with a linear forward model), meaning it has no flat valleys, widely separated modes, or heavy tails, then the samples produced by ZO-APMC are statistically indistinguishable from the images drawn from the true posterior conditioned on measurements, up to an $\varepsilon$ error, after $O(1/\varepsilon^4)$ function evaluations with arbtirary fixed per-iteration budget.
>
> $\textbf{Intuition on Theorem 2:}$ For solving ill-posed inverse problems, the most principled objective is to sample from the true posterior, since there are many possible reconstructions (solutions) consistent with the measurements. Intuitively, Theorem 2 shows that with very long ZO-APMC iterations under decaying parameters, schedules (i.e., geometric annealing used in practice [1]) and decaying score-approximation error, the ZO-APMC reconstructions converge to the exact target posterior. That is, each generated sample corresponds to a plausible underlying image or object consistent with the measured data, and this can be achieved with arbitrary per-iteration cost $pb=O(1)$. This property enables our method to provide reliable uncertainty quantification, an essential component in inverse problems [2]. In contrast, existing black-box approaches [3,4,5] do not offer such guarantee, as they are not proven to sample from the target posterior. Similarly, popular gradient-based methods do not offer such guarantee [6,7].
>
> References:
>
> [1] Sun, Y., Wu, Z., Chen, Y., Feng, B. T., & Bouman, K. L. (2024). Provable probabilistic imaging
> using score-based generative priors. IEEE Transactions on Computational Imaging.
>
> [2] Litvinenko, A., & Matthies, H. G. (2013). Inverse problems and uncertainty quantification. arXiv preprint arXiv:1312.5048.
>
> [3] Marco A Iglesias, Kody JH Law, and Andrew M Stuart. Ensemble kalman methods for inverse
> problems. Inverse Problems, 29(4):045001, 2013.
>
> [4] Haoyue Tang, Tian Xie, Aosong Feng, Hanyu Wang, Chenyang Zhang, and Yang Bai. Solving general noisy
> inverse problem via posterior sampling: A policy gradient viewpoint. In International Conference on
> Artificial Intelligence and Statistics, pp. 2116–2124. PMLR, 2024.
>
> [5] Yujia Huang, Adishree Ghatare, Yuanzhe Liu, Ziniu Hu, Qinsheng Zhang, Chandramouli S Sastry, Siddharth
> Gururani, Sageev Oore, and Yisong Yue. Symbolic music generation with non-differentiable rule guided
> diffusion. arXiv preprint arXiv:2402.14285, 2024.
>
> [6] H. Chung, J. Kim, M. T. Mccann, M. L.
> Klasky, and J. C. Ye, “Diffusion posterior sampling
> for general noisy inverse problems,” arXiv preprint arXiv:2209.14687, 2022
>
> [7] M. Renaud, J. Prost, A. Leclaire, and N. Papadakis. Plug-and-play image restoration with stochastic
> denoising regularization. In Int. Conf. Machine Learning (ICML), 2024b.

---

> > ### Comment · Reviewer_9C6Z · 2025-11-27
> >
> > Thank you to the authors for clarifying my questions and concerns. I am satisfied with the revisions and have increased my  score.

---

### Official Review · Reviewer_gjx9 · 2025-10-30

**Soundness:** 3
**Presentation:** 2
**Contribution:** 2
**Rating:** 4
**Confidence:** 3

**Summary:**

The paper proposes a Langevin inspired zeroth order annealing method for image restoration. The authors rigorously show that they approximate the ground truth posteriors in their setting, showing the scaling in number of forward evaluations. The approach is evaluated on a toy example, MRI, Black Hole imaging and a Navier Stokes example.

**Strengths:**

The paper is mostly well written, the motivation is relatively clear, and the math is proved rigorously. The evaluation is extensive.

**Weaknesses:**

1) I find the description in 3.1 of the algo to be insufficient for understand it, I would rather present non detailed theoretical statements in favor of describing the method properly. In particular the "accept-reject" like step is super unclear and needs elaboration. Also for the small-large batch size an intuitive explanation would help.

2) The theoretical restrictions are quite large, in particular Gaussians are usually not contained. I am also struggling with the main take away of the theory. If there was "informal" statements in the main paper, one could distill the statements into takeaways for practitioners, ignoring technical constants and looking at the main scaling behavior.

3) assumption 5 bites itself with the usual diffusion assumptions. At least in the diffusion case, one can show that the score needs to explode to "approximate" an empirical distribution. The authors now require that the learned score is bounded, and the difference to the true score is bounded, which I dont think is possible for empirical measures, and explodes for smoothing going to zero. (see for instance "Score based generative models detect manifolds")

4) I am also skeptical of the main motivation. The authors state examples like if the forward operator is a PDE solution, but calculating gradients (if the pde is already implemented) is in O(1). Of course there can incur instabilites, but they might also happen in the usual forward eval without the gradients. If the authors could elaborate on their specific setting, they would apply their algo in, I would be happy. The black hole example, I would be interested on the forward operator.

**Questions:**

see weaknesses.

---

> ### Author Response · Authors · 2025-11-23
> **Rebuttal by Authors**
>
> Thank you for taking the time to review our paper and for providing constructive, insightful feedback. We sincerely appreciate your comments, which have helped us improve the clarity and quality of our work. Please find our detailed responses to your questions below.
>
> ## Weakness 1: Clarification and intuition on the proposed method...
>
> We agree with the reviewer that the explanation in Section 3.1 of the initial manuscript lacked sufficient intuitive clarity. In the revised version, we added a short motivation section immediately before Section 3.1 (lines 185-192) to help the reader understand the $\textbf{intuitive motivation behind the variance-reduction mechanism}$ in the proposed ZO-APMC formulation. For convenience, we reproduce it here:
>
> "In a black-box setting, the negative likelihood score $\nabla f(\mathbf{x} _ k)$ is unavailable, so a natural choice is to approximate it using a naive ZO estimator, $\mathbf{g} _ k=(1/b) \sum_{i \in I} \tilde{\nabla}f_\mu (\mathbf{x} _ k, \mathbf{u} _ i)$. However, ZO estimates are known for their high variance, especially in high-dimensional settings [1], as they approximate a $d$-dimensional gradient using scalar function evaluations along random directions. This typically requires on the order of $d$ evaluations to obtain an accurate estimate. Consequently, convergence cannot be guaranteed with an arbitrary per-iteration budget; the batch size must increase with the dimension, which quickly becomes computationally prohibitive."
>
> Following this paragraph, we added another paragraph explaining $\textbf{how two branches of our estimator are constructed}$ intuitively at the beginning of Section 3.1 (lines 196-204):
>
> "We address this issue by introducing a variance-reduction mechanism for sampling, inspired by PAGE [2], which enables convergence under an arbitrary fixed per-iteration budget and thereby makes the algorithm practical. Intuitively, accurate ZO estimates require very large batch sizes per iteration, which is prohibitive, while small batches yield noisy estimates that degrade reconstruction quality. Our method addresses this by mixing both regimes: it computes a high-quality ZO estimate using a large batch $b$ occassionally (with prob. $p\ll1$); in most of the iterations (with prob. $1 - p$) it performs a cheap update using a small batch $b' \ll b$ to refine the existing high-quality estimate $\mathbf{g} _ k$ carried over from previous iterations, using the noisy estimate of the gradient change from $\mathbf{x} _ k$ to $\mathbf{x} _ {k+1}$."
>
> To clarify how the likelihood-score estimator (i.e., $\mathbf{g}_k$) with its two-branch $\textbf{``accept-reject like"}$ structure achieves variance reduction, we added intuitive explanation after Proposition 1. It is as follows
>
> "When $p = 1$, our method reduces to the posterior sampling with naive ZO estimator in equation (7), which estimates the negative likelihood score using a batch of $b$ perturbations. In this scenario, the error term is only bounded by the second-moment (or variance) term, which is the third term in (8). This variance grows at least linearly with the dimension (see property (c) of Lemma 1), so $b$ must scale with the dimension to keep the error manageable. This is computationally expensive and impractical to perform at every iteration. By choosing $p\ll 1$, we scale the variance by $p$ and obtain the effect of a large batch size, while using a smaller batch $b'\ll b$ in most of the iterations and evaluating the large batch $b$ occasionally. This reduces the variance term significantly while keeping the required average number of function evaluations per iteration smaller than the naive ZO approach. The trade-off is the introduction of two additional error terms from the "$1-p$ branch" of the estimator. The first is an estimation error propagated from previous iterations decaying with $1 -p^2<1$. The second is due to the estimation error of the gradient change between $\mathbf{x} _ k$ and $\mathbf{x} _ {k+1}$, which depends on the regularity of the forward model (Lipschitzness condition in Assumption 1), the ZO smoothing parameter $\mu$, and the discretization error of the Langevin diffusion (inside the expectation).
> These sources of error can all be mitigated by using a smaller discretization step size."
>
> Please see our next response for the intuitive explanation of the "small-large batch size".
>
> References:
>
> [1] Yurii Nesterov and Vladimir Spokoiny. Random gradient-free minimization of convex functions.
> Foundations of Computational Mathematics, 17(2):527–566, 2017.
>
> [2] Zhize Li, Hongyan Bao, Xiangliang Zhang, and Peter Richt´arik. Page: A simple and optimal prob-
> abilistic gradient estimator for nonconvex optimization. In International conference on machine
> learning, pp. 6286–6295. PMLR, 2021.

---

> ### Author Response · Authors · 2025-11-23
> **Rebuttal by Authors**
>
> ## Weakness 1: small-large batch size intuition
>
> The intuition behind "small-large batch size" is that as we increase the small batch size, ZO-APMC can tolerate less regular forward models (larger Lipschitz constant, allowing faster changes), larger discretization error, and more bias reduction due to the ZO approximation. However, as it increases, average per-iteration cost increases as well. For the large batch size, increasing it  reduces the variance inherent in ZO estimation, which depends on the dimension $d$, but it also increases the per-iteration cost. One should find the optimal $b$, $b'$, and $p$ to get the optimal solution, which we provided in Theorem 1 to bound the average Fisher information with $\varepsilon.$

---

> ### Author Response · Authors · 2025-11-23
> **Rebuttal by Authors**
>
> ## Weakness 2: Theoretical results lack intuition...
>
> We agree with the reviewer that the original presentation of the theoretical results did not provide enough intuition. In the revised manuscript, we provided one paragraph for explaining the intuition of each theoretical result to distill them into takeaways for practitioners. Here are the takeaways of our theorems and a corollary:
>
> $\textbf{Intuition on Theorem 1:}$ Intuitively, Theorem 1 shows that ZO-APMC progressively improves its average estimate of the directions that lead to reconstructions consistent with the measurements, while relying only on forward model inputs and outputs (not on gradients). Achieving at most $\varepsilon$ average error in these directions requires $O(1/\varepsilon^4)$ function evaluations in total. Although this number grows polynomially with dimension, it remains achievable
> under any fixed per-iteration budget of function evaluations thanks to the proposed variance-reduction mechanism. Moreover, the bias error of the ZO estimates, which appears as the fourth term in the Fisher information upper bound, can be made arbitrarily small by selecting a smaller ZO smoothing parameter $\mu$. Doing so, however, requires using a smaller step size $\gamma$ while keeping the number of function evaluations per iteration fixed.
>
> $\textbf{Practical procedure:}$ To obtain a reconstruction with an $\varepsilon$ error described above and low ZO bias under a fixed per-iteration budget, one should first select the ZO smoothing parameter and the batch size based on the target error and the problem dimension $d$. Next, setting $p = 1/b$ and $b'$ using the average-budget relation
> $\mathrm{budget} = (1-p)b' + pb.$ With these choices in place and a geometrically decaying annealing schedule [1], the algorithm is ready to run.
>
> $\textbf{Intuition on Corollary 1:}$ Corollary 1 intuitively implies that if the likelihood density satisfies a Poincaré inequality (i.e. Gaussian measurement noise with a linear forward model), meaning it has no flat valleys, widely separated modes, or heavy tails, then the samples produced by ZO-APMC are statistically indistinguishable from the images drawn from the true posterior conditioned on measurements, up to an $\varepsilon$ error, after $O(1/\varepsilon^4)$ function evaluations with arbitrary fixed per-iteration budget.
>
> $\textbf{Intuition on Theorem 2:}$ For solving ill-posed inverse problems, the most principled objective is to sample from the true posterior, since there are many possible reconstructions (solutions) consistent with the measurements. Intuitively, Theorem 2 shows that with very long ZO-APMC iterations under decaying parameters, schedules (i.e., geometric annealing used in practice [1]) and decaying score-approximation error, the ZO-APMC reconstructions converge to the exact target posterior. That is, each generated sample corresponds to a plausible underlying image or object consistent with the measured data, and this can be achieved with arbitrary per-iteration cost $pb=O(1)$. This property enables our method to provide reliable uncertainty quantification, an essential component in inverse problems [2]. In contrast, existing black-box approaches [3,4,5] do not offer such guarantee, as they are not proven to sample from the target posterior. Similarly, popular gradient-based methods do not offer such guarantee [6,7].
>
> [1] Sun, Y., Wu, Z., Chen, Y., Feng, B. T., & Bouman, K. L. (2024). Provable probabilistic imaging using score-based generative priors. IEEE Transactions on Computational Imaging.
>
> [2] Litvinenko, A., & Matthies, H. G. (2013). Inverse problems and uncertainty quantification. arXiv preprint arXiv:1312.5048.
>
> [3] Marco A Iglesias, Kody JH Law, and Andrew M Stuart. Ensemble kalman methods for inverse problems. Inverse Problems, 29(4):045001, 2013.
>
> [4] Haoyue Tang, Tian Xie, Aosong Feng, Hanyu Wang, Chenyang Zhang, and Yang Bai. Solving general noisy inverse problem via posterior sampling: A policy gradient viewpoint. In International Conference on Artificial Intelligence and Statistics, pp. 2116–2124. PMLR, 2024.
>
> [5] Yujia Huang, Adishree Ghatare, Yuanzhe Liu, Ziniu Hu, Qinsheng Zhang, Chandramouli S Sastry, Siddharth Gururani, Sageev Oore, and Yisong Yue. Symbolic music generation with non-differentiable rule guided diffusion. arXiv preprint arXiv:2402.14285, 2024.
>
> [6] H. Chung, J. Kim, M. T. Mccann, M. L. Klasky, and J. C. Ye, “Diffusion posterior sampling for general noisy inverse problems,” arXiv preprint arXiv:2209.14687, 2022
>
> [7] M. Renaud, J. Prost, A. Leclaire, and N. Papadakis. Plug-and-play image restoration with stochastic denoising regularization. In Int. Conf. Machine Learning (ICML), 2024b.

---

> ### Author Response · Authors · 2025-11-24
> **Rebuttal by Authors**
>
> ## Weakness 3: Assumption 5 and Gaussian noise modeling...
>
> $\textbf{The bounded score network norm assumption (update):}$ We appreciate the reviewer's constructive feedback and the reference to the literature regarding Assumption 5. We agree with the reviewer that the initial Assumption 5 was strong. Therefore, in the revised manuscript, $\textbf{we replaced Assumption 5 with a more relaxed condition}$. Instead of imposing a uniform bound $\lVert \mathcal{S} _ \theta (\mathbf{x}, \sigma _ k) \rVert_2 \leq R _ s$, we now assume $\lVert \mathcal{S}_\theta (\mathbf{x} , \sigma _ k) \rVert_2 \leq \sigma_k^{-1}C$, where $C$ is a numerical constant and $\sigma_k$ is standard deviation of the noise at $k ^ {th}$ iteration. Thus, as $\sigma _ k \rightarrow 0$, $\lVert \mathcal{S} _ \theta (\mathbf{x}, \sigma _ k) \rVert _ 2$ becomes unbounded  This is consistent with the well-known empirical findings in [1,2]: when a score network is trained via denoising score matching loss across multiple noise levels $\sigma_k$, its learned score scales as $\lVert \mathcal{S} _ \theta (\mathbf{x}, \sigma_k) \rVert \propto 1/\sigma_k$. We refer the reviewer to [1] for further details. After making this change, all theoretical results continue to hold. We made only minor revisions to the derivations, which have been updated accordingly.
>
> $\textbf{The bounded score network error assumption:}$ Assumption 5 also assumes that $\lVert \nabla \log p _ {\sigma _ k }(\mathbf{x}) - \mathcal{S} _ \theta (\mathbf{x}, \sigma _ k) \rVert _ 2 \leq \varepsilon _ {\sigma _ k}$. This is a reasonable assumption because, as proved in [3], the optimal score network that minimizes the denoising score-matching objective [1,2] satisfies $\mathcal{S} _ \theta ^* (\mathbf{x}, \sigma _ k) = \nabla \log p _ {\sigma _ k}(\mathbf{x})$ with probability 1. Our SGM priors were trained using the same loss function. Because the exact optimal score is not available, we made a more general assumption and assumed bounded approximation error between the learned score and the perturbed score, and this error depends on the perturbation level $\sigma_k$.
>
> $\textbf{Gaussian noise modeling example:}$ We want to clarify and correct the statement in Remark 1 regarding the applicability of Gaussian noise modeling. It does not apply to Lipschitz negative log-likelihood distributions $\textbf{for the $\textit{global}$ domain $\mathbb{R}^d$}$. However, in practice, image distributions are in between $[0,1]^d$ or $[0,255]^d$, so they are in a $\textbf{bounded domain $(\lVert \mathbf{x} \rVert _ 2 \leq R)$.}$
>
> Then, for a linear forward model (i.e. MRI reconstruction), we can show the Lipschitzness of the negative log-likelihood as follows
> \begin{equation}
> \lVert \nabla f(\mathbf{x}) \rVert_2 \leq \frac{\lVert \mathbf{A}^T(\mathbf{A}\mathbf{x} - \mathbf{y}) \rVert^2_2}{\sigma^2} \leq \frac{1}{\sigma^2}\left(\lVert \mathbf{A}\rVert_2^2\lVert\mathbf{x}\rVert_2^2 + \lVert \mathbf{A}^T\mathbf{y} \rVert^2_2\right) < \infty.
> \end{equation}
>
> For nonlinear $\mathbf{A}(\cdot)$, one may argue the Lipschitz continuity of the negative log-likelihood, which is indeed a stronger requirement than smoothness. However, given the inherent difficulty of sampling from the exact non-log-concave posterior in a black-box setting with arbitrary per-iteration cost ($pb = O(1)$), this assumption is reasonable.
>
> $\textbf{Additional theoretical result:}$ We have derived an additional theoretical result establishing convergence in Fisher information $\textbf{without assuming a Lipschitz-continuous log-likelihood}$, under a setting with noisy gradients. We provide the justification for this setting in our next response. With small modification in our estimator of the log-likelihood, we achieved $O(d^2\sigma_{\text{noise}} ^ 4  L _ \pi ^ 2 / \varepsilon^4)$ number of total gradient calculations or iterations with arbitrary fixed per-iteration budget $pb=O(1)$. The presentation of these results are in Appendix A.2.
>
>
> References:
>
> [1] Song, Y. and Ermon, S. Generative modeling by estimating gradients of the data distribution. In
> Advances in Neural Information Processing Systems, pp. 11918–11930, 2019.
>
> [2] Song and S. Ermon, “Improved techniques for training score-based generative models,” in Proc. Int. Conf. Neural Inf. Process. Syst., 2020, pp. 12438–12448.
>
> [3] Vincent. A connection between score matching and denoising autoencoders. Neural compu- tation, 23(7):1661–1674, 2011.

---

> > ### Author Response · Authors · 2025-11-24
> > **Rebuttal by Authors**
> >
> > ## Weakness 4: Main motivation and the setting...
> >
> > Our proposed ZO-APMC method is designed for solving general inverse problems with black-box forward models, where computing forward-model gradients accurately is not possible or inefficient. There are many applications where this setting applies. For example, many MRI vendors do not disclose their internal scanner settings [1,2], making the forward model proprietary and preventing the use of gradient-based method. In our experiments, however, we still include them (where applicable) to show that the proposed ZO-APMC achieves performance comparable to state-of-the-art gradient-based methods in MRI reconstruction and black hole imaging. We also compared the proposed ZO-APMC with the baselines on the black hole imaging inverse problem, as it is highly nonlinear and exhibits a non-log-concave distribution. In practice most of black-box forward models are nonlinear and non-log-concave.
> >
> > References:
> >
> > [1] Matt T Cashmore, Aaron J McCann, Stephen J Wastling, Cormac McGrath, John Thornton, and
> > Matt G Hall. Clinical quantitative mri and the need for metrology. The British Journal of Radiol-
> > ogy, 94(1120):20201215, 2021.
> >
> > [2] A Harbaugh, E Banta, M Hill, and M McDonald. The us geological survey modular groundwater
> > model [j]. User Guide to Modularization Concepts and the Groundwater Flow Process, 2000.

---

> > ### Comment · Reviewer_gjx9 · 2025-11-24
> > **boundedness of the score**
> >
> > I mean the argument that the optimal S needs to satisfy $S = \nabla log p_t$ does not cut it for me. Of course it does, but for this to work (for empirical samples), the $S$ needs to explode (which it cannot). If it is not bounded, then you basically describe a different target density, which would not be bad.

---

> ### Author Response · Authors · 2025-11-24
>
> ## Weakness 4: Main motivation and the setting cont'd...
>
> Regarding PDE-based inverse problems such as Navier–Stokes, which is our last experiment setting, no closed-form solution exists, and obtaining accurate numerical gradients via automatic differentiation is challenging because it requires back propagation through a very large and complex computation graph [1]. Furthermore, even if a forward operator could be solved and differentiated with PDE, it may be written in a \textbf{legacy programming language} and optimized only for forward execution, which makes it difficult to integrate with modern automatic-differentiation libraries [2]. A concrete example of this is discussed in the Introduction section of [4]. $\textbf{In addition}$, some forward operators are inherently non-differentiable due to $\textbf{discontinuous physics}$ and cannot be modeled accurately with differentiable simulators, such as turbulence (cloud modeling) [4,5] or fracture mechanics [6]. As the reviewer highlighted, in these scenarios, we can still have access to the unstable or noisy gradients. $\textbf{So, for these settings, we added new theoretical result with intuitive derivation of the modified algorithm}$ in Fisher information convergence in Appendix A.2. We modeled instability of gradients as additive Gaussian noise and, with small modification in the estimator, showed that the propose algorithm still converges with arbitrary fixed per-iteration budget $pb=O(1)$ with $O\left(d ^ 2 \sigma _ {\text{noise}}^4L _ \pi^2 / \varepsilon^4 \right)$.$\textbf{Lastly}$, there exist settings in which the forward model is a $\textbf{rule-based or expert-designed system}$, for which no meaningful derivative exists [7-9]. Our work can be applied to these inverse problems. We have included this discussion between $\textbf{lines 39-50.}$
>
> We provided the definition of the forward operator for black hole imaging in Appendix A.5.2. For more details about the inverse problem, we refer the reviewer to [1].
>
> [1] Hongkai Zheng, Wenda Chu, Bingliang Zhang, Zihui Wu, Austin Wang, Berthy Feng, Caifeng Zou, Yu Sun, Nikola Borislavov Kovachki, Zachary E Ross, et al. Inversebench: Benchmarking plug-and-play diffusion priors for inverse problems in physical sciences. In The Thirteenth International Conference on Learning Representations, 2025.
>
> [2] A Harbaugh, E Banta, M Hill, and M McDonald. The us geological survey modular groundwater
> model [j]. User Guide to Modularization Concepts and the Groundwater Flow Process, 2000.
>
> [3] Patel, D. V., Lee, J., Farthing, M. W., Kitanidis, P. K., & Darve, E. F. HIGH-DIMENSIONAL BAYESIAN INVERSION WITH BLACK-BOX SIMULATORS.
>
> [4] Ignacio Lopez-Gomez, Costa Christopoulos, Haakon Ludvig Langeland Ervik, Oliver RA Dunbar,
> Yair Cohen, and Tapio Schneider. Training physics-based machine-learning parameterizations
> with gradient-free ensemble kalman methods. Journal of Advances in Modeling Earth Systems,
> 14(8):e2022MS003105, 2022.
>
> [5] Zhihong Tan, Colleen M Kaul, Kyle G Pressel, Yair Cohen, Tapio Schneider, and Jo˜ao Teixeira. An
> extended eddy-diffusivity mass-flux scheme for unified representation of subgrid-scale turbulence
> and convection. Journal of Advances in Modeling Earth Systems, 10(3):770–800, 2018.
>
> [6] Nicolas Mo¨es, John Dolbow, and Ted Belytschko. A finite element method for crack growth without
> remeshing. International journal for numerical methods in engineering, 46(1):131–150, 1999.
>
> [7] Yujia Huang, Adishree Ghatare, Yuanzhe Liu, Ziniu Hu, Qinsheng Zhang, Chandramouli
> Shama Sastry, Siddharth Gururani, Sageev Oore, and Yisong Yue. Symbolic music generation
> with non-differentiable rule guided diffusion. In Proceedings of the 41st International Con-
> ference on Machine Learning, volume 235 of Proceedings of Machine Learning Research, pp.
> 19772–19797. PMLR, 21–27 Jul 2024.
>
> [8] Aosen Gong, Wei He, You Cao, Guohui Zhou, and Hailong Zhu. Interpretability metrics and op-
> timization methods for belief rule based expert systems. Expert Systems with Applications, pp.
> 128363, 2025.
>
> [9] Alexander P Rotshtein and Hanna B Rakytyanska. Inverse inference based on fuzzy rules. In Fuzzy
> Evidence in Identification, Forecasting and Diagnosis, pp. 193–233. Springer, 2012.

---

> ### Author Response · Authors · 2025-12-03
> **Clarification on bounded score error**
>
> We sincerely thank the reviewer for their insightful comments regarding the validity of Assumption 5, particularly in the context the findings in [1]. This deep observation has allowed us to significantly sharpen the theoretical positioning of our work. We provide below a rigorous justification for Assumption 5, clarifying the distinction between the empirical and population measures, and demonstrating mathematically that there is no conflict between our assumptions and the result of [1].
>
>
> To address the concern about score explosion, it is essential to distinguish between two distinct target distributions:
> 1. $\textbf{ The Empirical Measure ($ \hat{p} _ t $):}$ Let $\hat{p} _ {\text{data}} = \frac{1}{N} \sum _ {i=1} ^ N \delta _ {\mathbf{x} _ i}$ be the empirical training distribution. The perturbed marginal is $\hat{p} _ {\sigma _ t}(\mathbf{x}) = \hat{p} _ {\text{data}} * \mathcal{N}(0, \sigma _ t ^ 2 I)$. As $\sigma _ t \to 0$, the score $\nabla \log \hat{p} _ {\sigma _ t}(\mathbf{x})$ develops singularities of order $O(1/\sigma_t)$ or $O(1/\sigma_t^2)$ near the data points.
> 2. $\textbf{The Population Measure ($p _ {\sigma _ t}$):}$ Let $p _ {\text{data}}$ be the true underlying population distribution (supported on a manifold $\mathcal{M}$). The perturbed marginal is $p _ {\sigma_t}(\mathbf{x}) = p _ {\text{data}} * \mathcal{N}(0, \sigma _ t^2 I)$. Crucially, as $\sigma _ t \to 0$, the score $\nabla \log p _ {\sigma _ t}$ converges to the smooth, bounded score of the population distribution.
>
> Assumption 5 imposes two conditions on the learned score network $\mathcal{S} _ \theta(\mathbf{x}, \sigma_k)$:
> 1. $\textbf{Norm Bound:} \lVert\mathcal{S} _ \theta(\mathbf{x}, \sigma _ k)\rVert _ 2 \leq C\sigma _ k ^ {-1}$
> 2. $\textbf{Approximation Error:} \lVert \nabla \log p _ {\sigma _ k}(\mathbf{x}) - \mathcal{S} _ \theta(\mathbf{x}, \sigma_k)\rVert _ 2 \leq \epsilon_{\sigma_k}$, with $\epsilon_{\sigma_k} \to 0$
>
> The Norm Bound is consistent with the scaling of diffusion processes on manifolds (where the score scales as $1/ \sigma _ k$ to provide the restoring force). The Approximation Error bound further holds as, we interpret the target $p_{\sigma_k}$ as the $\textbf{population distribution, not empirical distribution}.$ A bounded neural network $\mathcal{S}_\theta$ can approximate the smooth population score with arbitrary precision ($\epsilon _ {\sigma _ k} \to 0$), whereas as pointed out by [1] and the reviewer, it cannot approximate the singular empirical score (where error would explode). Therefore, there is no conflict between our assumption and the results of [1]. In fact, $\textbf{our assumption mathematically enforces the condition for generalization derived in [1].}$
>
> To see this clearly, we recall that the influential work [1] proves that to avoid memorization (converging to $\hat{p} _ {\text{data}}$), the score error relative to the $\textbf{empirical measure}$ must be unbounded (specifically, the Girsanov energy must diverge). Our Assumption 5 requires that the score error relative to the $\textbf{population measure $p _ {\sigma _ k}$}$ is bounded. We prove consistency using the Triangle Inequality. Let $\mathbf{s} _ {\text{emp}} = \nabla \log \hat{p} _ {\sigma_k}$ be the singular empirical score, $\mathbf{s} _ {\text{pop}} = \nabla \log p_{\sigma_k}$ be the smooth population score, and $\mathcal{S}_\theta$ be the network. The error relative to the empirical measure is bounded from below by:
>
> $$
> \lVert \mathbf{s} _ {\text{emp}} - \mathcal{S} _ \theta \rVert _ 2 \geq \underbrace{\lVert\mathbf{s} _ {\text{emp}} - \mathbf{s} _ {\text{pop}} \rVert _ 2} _ {\text{Intrinsic Divergence}} - \underbrace{\lVert\mathbf{s}_{\text{pop}} - \mathcal{S} _ \theta\rVert _ 2} _ {\text{Assumption 5}}
> $$
>
> As $\sigma_k \to 0$, the difference between the singular empirical score and the smooth population score (i.e. the Intrinsic Divergence term) diverges to infinity: $\lVert \mathbf{s} _ {\text{emp}} - \mathbf{s} _ {\text{pop}}\rVert _ 2 \to \infty$. We assume the network approximates the population score well: $\lVert \mathbf{s} _ {\text{pop}} - \mathcal{S} _ \theta\rVert _ 2 \leq \epsilon _ {\sigma _ k} \approx 0$.
> Substituting these into the inequality implies $\lVert \mathbf{s} _ {\text{emp}} - \mathcal{S} _ \theta \rVert _ 2 \to \infty$, which is the result proved by [1].
>
> Thus, $\textbf{Assumption 5 serves as a rigorous quantification of the model's generalization capability.}$ We will clarify this distinction and add this rigorous justification to the final manuscript to resolve any ambiguity.
>
> Reference:
>
> [1] Score-Based Generative Models Detect Manifolds, Pidstrigach, NeurIPS 2022

---

### Official Review · Reviewer_2vHU · 2025-10-31

**Soundness:** 2
**Presentation:** 2
**Contribution:** 1
**Rating:** 4
**Confidence:** 3

**Summary:**

The paper proposes ZO-APMC, a derivative-free posterior sampling method for inverse problems that uses pre-trained score-based generative models (SGMs) as priors. The main claim is to provide the first formal convergence guarantees for such a "black-box" setting, where only forward-model evaluations are available. The authors combine a zeroth-order (ZO) gradient estimator with a variance reduction technique and an annealing schedule. Theoretical bounds on convergence are derived, and the method is empirically evaluated on several inverse problems, including MRI reconstruction, black-hole imaging, and Navier-Stokes equations.

**Strengths:**

1. The paper is overall well-written.
2. The author conducted extensive derivations, although most of the formula proofs felt over-engineered in an attempt to complete the proofs.

**Weaknesses:**

1. The central premise of this work is to establish theoretical guarantees for a highly specific, arguably contrived, problem setting: derivative-free posterior sampling with SGM priors. While technically challenging, the practical motivation for this exact combination feels weak. A significant portion of modern inverse problems, especially in imaging, do have differentiable forward models (e.g., via auto-differentiation through simulators). The community is increasingly moving towards differentiable programming. This paper overstates the prevalence of purely "black-box" scenarios where its method would be uniquely essential, making the direction feel more like a niche theoretical exercise than a solution to a pressing, widespread problem.
2. The paper attempts to tame the notoriously high variance of ZO gradient estimators within a stochastic sampling framework. This involves complex machinery like PAGE-style variance reduction and delicate annealing schedules, creating a "tricky" and highly engineered solution. The resulting algorithm is complex, laden with sensitive hyperparameters.
3. Despite the variety of tasks, the experimental setup feels superficial and fails to convincingly demonstrate that ZO-APMC represents a significant practical advancement over existing methods. The results (Table 1) show that ZO-APMC (PSNR 35.29) barely outperforms a simple baseline like DPG (PSNR 32.17) and is still noticeably worse than the gradient-based APMC (PSNR 36.55).
4. The paper's main selling point is its theoretical guarantees. However, the derived complexity bounds are astronomical (e.g., O(d⁷...)). Such bounds are often so loose that they provide little practical insight into the algorithm's real-world behavior or how to set its hyperparameters.

**Questions:**

I'm curious to know, besides publishing papers, what is the practical significance of researching inverse problem solving under black boxes?

---

> ### Author Response · Authors · 2025-11-24
> **Rebuttal by Authors**
>
> Thank you for taking the time to review our paper and for providing constructive, insightful feedback. We sincerely appreciate your comments, which have helped us improve the clarity and quality of our work. Please find our detailed responses to your questions below.
>
> ## Weakness 1 & Question 1: Practical motivation...
>
> While the community is increasingly moving toward differentiable simulators, many practically important settings still lack access to forward-model gradients. In addition to the applications discussed in our initial submission, we describe several further examples in what follows. $\textbf{First}$, even if a simulator is in principle differentiable, it may be implemented in a $\textbf{legacy programming language}$ and optimized exclusively for forward computations, which makes integration with modern automatic differentiation libraries challenging [1]. A concrete example of this is discussed in the Introduction section of [2]. $\textbf{Second}$, the forward operator may be a $\textbf{proprietary (closed-source) system}$, as is often the case in commercial MRI reconstruction pipelines [3,4]. $\textbf{Third}$, some forward operators are inherently non-differentiable due to $\textbf{discontinuous physics}$ and cannot be modeled accurately with differentiable simulators, such as turbulence (cloud modeling) [5,6] or fracture mechanics [7]. $\textbf{Lastly}$, there exist settings in which the forward model is a $\textbf{rule-based or expert-designed system}$, for which no meaningful derivative exists [8–10]. Notably, the importance of addressing inverse problems in fully black-box settings has been increasingly recognized with several recent 2024 works proposing SGM-based approaches, including SCG [7] and EnKG [10]. We added this additional literature review to the $\textbf{lines 39-50}$ in the revised submission.
>
> Additionally, while the forward model may be differentiable through automatic-differentiation simulators, the resulting gradients are often unstable or high-variance due to discretization errors or discontinuous physics [17]. In the revised manuscript, we now include this setting by $\textbf{assuming access to noisy, ``unstable'' forward model gradients corrupted by additive Gaussian noise}$ with variance $\sigma_{\text{noise}}^2$. We also $\textbf{remove the Lipschitzness assumption (Assumption 1)}$ for this setting on the forward model and modify our PAGE estimator by introducing a variance-reduced estimator suitable to the setting to guarantee convergence in FI under an arbitrary per-iteration computational budget. We prove that achieving $\varepsilon$-relative Fisher information requires $O\\left(\frac{d^2 \sigma_{\text{noise}}^4 L_\pi^2}{\varepsilon^4}\right)$
> total iterations (or gradient evaluations) with a fixed per-iteration cost of $pb = O(1)$. Additional details on this setting are provided in Appendix A.2. We are running the toy experiments for this setting and will share the results as soon as possible.
>
>
>
> References:
>
> [1] Harbaugh, A., et al. USGS Modular Groundwater Model: User Guide. USGS, 2000.
>
> [2] Patel, D. V., et al. High-dimensional Bayesian inversion with black-box simulators. (manuscript).
>
> [3] Karakuzu, A., et al. Rethinking MRI as a measurement device through modular and portable pipelines. Magn. Reson. Mater. Phy., 2025.
>
> [4] Cashmore, M. T., et al. Clinical quantitative MRI and the need for metrology. Br. J. Radiol., 2021.
>
> [5] Lopez-Gomez, I., et al. Training physics-based ML parameterizations with gradient-free ensemble Kalman methods. JAMES, 2022.
>
> [6] Tan, Z., et al. Extended EDMF scheme for subgrid turbulence and convection. JAMES, 2018.
>
> [7] Moës, N., Dolbow, J., & Belytschko, T. FEM for crack growth without remeshing. Int. J. Numer. Methods Eng., 1999.
>
> [8] Huang, Y., et al. Symbolic music generation with rule-guided diffusion. ICML, 2024.
>
> [9] Gong, A., et al. Interpretability metrics for belief-rule expert systems. Expert Syst. Appl., 2025.
>
> [10] Rotshtein, A. P., & Rakytyanska, H. Inverse inference based on fuzzy rules. In Fuzzy Evidence, Springer, 2012.
>
> [11] Zheng, H., et al. Ensemble Kalman diffusion guidance. TMLR, 2025.
>
> [12] Song, Y., & Ermon, S. Improved techniques for training score-based generative models. NeurIPS, 2020.
>
> [13] Sun, Y., et al. Provable probabilistic imaging using score-based generative priors. IEEE Trans. Comput. Imaging, 2024.
>
> [14] Tang, H., et al. Posterior sampling for noisy inverse problems via policy gradient. AISTATS, 2024.
>
> [15] Chung, H., et al. Diffusion posterior sampling for noisy inverse problems. arXiv:2209.14687, 2022.
>
> [16] Renaud, M., et al. Plug-and-play image restoration with stochastic denoising regularization. ICML, 2024.
>
> [17] Zheng, H., et al. InverseBench: Benchmarking PnP diffusion priors for inverse problems. ICLR, 2025.
>
> [18] Hueckelheim, J., et al. A review of automatic differentiation pitfalls in scientific computing. ICML Workshop on DAEs, 2023.

---

> ### Author Response · Authors · 2025-11-24
> **Rebuttal by Authors**
>
> ## Weakness 2: PAGE variance-reduction mechanism and annealing schedules...
>
> $\textbf{Incorporating PAGE}$ is not a highly engineered choice; it is $\textbf{necessary to guarantee convergence in FI under an arbitrary per-iteration computational budget.}$ This becomes clear by examining the recursive bound in Proposition 1:
>
> \begin{equation}
> \mathbb{E}[\lVert  e _ k\rVert^2] \leq (1 - p ^2) \mathbb{E}[\lVert  e _ {k-1}\rVert^2]  + \frac{4d(1-p)L _ {f _ 1}^2}{b'\mu^2} E[\lVert \mathbf{x} _ k - \mathbf{x} _ {k-1}\rVert^2] + \frac{p\mathbb{E}{[\lVert \tilde{\nabla} f _ \mu (\mathbf{x} _ k, \mathbf{u} _ i) \rVert]^2}}{b},
> \end{equation} where $b'$ is small batch size and constant/fixed, b is large batch size. When $p=1$, the update reduced to standard zeroth-order LMC sampling algorithm:
> \begin{equation}
> \mathbf{x} _ {k + 1} = \mathbf{x} _ {k + 1} - \gamma \left( \frac{1}{b} \sum _ {i \in I} \tilde{\nabla} f _ \mu (\mathbf{x _ k}, u _ i) - \alpha _ k \mathcal{S} _ \theta (\mathbf{x _ k}, \sigma _ k) \right) + \sqrt{2 \gamma} \mathbf{Z} _ k, \quad \mathbf{u} _ i, \mathbf{Z} _ k \sim \mathcal{N}(0, I),
> \end{equation}and the reconstruction simplies to
> \begin{equation}
> \mathbb{E}[\lVert \mathbf{e} _ k \rVert ^2\] \leq \frac{\mathbb{E} [\lVert \tilde{\nabla} f _ \mu (\mathbf{x} _ k, \mathbf{u} _ i) \rVert^2]}{b}.
> \end{equation}  Because this second-moment term grows with the variance of the ZO estimator $\tilde{\nabla} f_\mu(\mathbf{x}_k, \mathbf{u}_i)$, the estimator error term above does not vanish for an arbitrary batch size $b$. In fact, $b$ must be at least on the order of the dimension, i.e., $b = O(d)$ (see ZO Lemma 1 in Section A.3.1). This is computationally impractical, especially for high dimensions. $\textbf{We solved this problem by introducing PAGE variance-reduction}.$ By choosing $p\ll1$, we scale the second-moment term (the third term in Proposition 1) by $p$ and obtain the effect of a large batch size while using a much smaller batch size $b'\ll b$ in most iterations and evaluating the large batch $b$ occasionally. For example, when we choose $p = 1/b$, the variance (second-moment) term is scaled by $1/b^{2}$ rather than $1/b$, yielding a substantial reduction in variance. The average number of function evaluations per iteration is $(1-p)b' + pb$, but since $b'$ is an arbitrary constant, its contribution does not affect the order. Thus, the dominant term is $pb = O(1)$, which is constant and significantly smaller than the naive ZO approach where $p = 1$ and the per-iteration cost is $b = O(d)$. The trade-off is the presence of two additional terms, which can be controlled. The first term in Proposition 1 is an error term from the previous iteration that decays with $1-p^2<1$. The second term can be controlled by the step size $\gamma$ embedded inside the expectation.
>
> $\textbf{Setting an annealing schedule}$ for Langevin sampling is a well-established technique for improving the speed of sampling [1]. As shown in Theorem 1, the hyperparameters of this schedule are independent of $p$ and $b$, meaning that these choices do not influence one another with respect to convergence behavior. In [1], the authors present a principled and widely adopted method for selecting the annealing schedule parameters (i.e., $\alpha_k$, $\sigma_k$, $\xi$) based on properties of the dataset. We follow this procedure in our black hole and Navier-Stokes imaging experiments, while for the MRI reconstruction task we adopt the parameter settings used in the prior work [2]. In fact, similar scheduling parameters exist in most of the SGM-based posterior sampling algorithms including the DPG baseline [2-5].
>
> References:
>
> [1] Song, Y., \& Ermon, S. (2020). Improved techniques for training score-based generative models. Advances in neural information processing systems, 33, 12438-12448.
>
> [2] Sun, Y., Wu, Z., Chen, Y., Feng, B. T., \& Bouman, K. L. (2024). Provable probabilistic imaging using score-based generative priors. IEEE Transactions on Computational Imaging.
>
> [3] Haoyue Tang, Tian Xie, Aosong Feng, Hanyu Wang, Chenyang Zhang, and Yang Bai. Solving general noisy inverse problem via posterior sampling: A policy gradient viewpoint. In International Conference on Artificial Intelligence and Statistics, pp. 2116–2124. PMLR, 2024.
>
> [4] H. Chung, J. Kim, M. T. Mccann, M. L. Klasky, and J. C. Ye, “Diffusion posterior sampling for general noisy inverse problems,” arXiv preprint arXiv:2209.14687, 2022
>
> [5] M. Renaud, J. Prost, A. Leclaire, and N. Papadakis. Plug-and-play image restoration with stochastic denoising regularization. In Int. Conf. Machine Learning (ICML), 2024b.

---

> ### Author Response · Authors · 2025-11-24
> **Rebuttal by Authors**
>
> ## Weakness 3: Quantitative results...
>
> We respectfully disagree with the reviewer's assessment of our quantitative results. We think $\textbf{3.12 dB PSNR}$ improvement over DPG in brain MRI reconstruction task represents a clear and meaningful gain under the same black-box conditions. Moreover, the $\textbf{1.29 dB}$ gap to gradient-based APMC is relatively small given that APMC uses gradients while ZO-APMC relies solely on ZO information. Since APMC is not applicable to black-box settings, for example when forward model gradients are unavailable due to commerical proprietary tools (i.e. MRI reconstruction [1,2]), ZO-APMC offers a practical alternative. In these scenarios, ZO-APMC remains close to the APMC (gradient-based method) and clearly outperforms the current black-box state-of-the-art DPG method in all reported metrics except standard deviation.
>
> References:
>
> [1] Agah Karakuzu, Nadia Blostein, Alex Valcourt Caron, Arnaud Bor´e, Franc¸ois Rheault, Maxime
> Descoteaux, and Nikola Stikov. Rethinking mri as a measurement device through modular and
> portable pipelines. Magnetic Resonance Materials in Physics, Biology and Medicine, pp. 1–17,
> 2025.
>
> [2] Matt T Cashmore, Aaron J McCann, Stephen J Wastling, Cormac McGrath, John Thornton, and
> Matt G Hall. Clinical quantitative mri and the need for metrology. The British Journal of Radiol-
> ogy, 94(1120):20201215, 2021.

---

> > ### Author Response · Authors · 2025-11-24
> > **Rebuttal by Authors**
> >
> > ## Weakness 4: The bound seem to scale badly...
> >
> > While the Fisher information (FI) convergence rate has a high-polynomial dependence on $d$, $\varepsilon$, and $L_m$ (specifically $O(d^7 L_m^6 / \varepsilon^4)$), this scaling is not a result of loose arguments. Rather, it reflects the intrinsic difficulty of the underlying posterior estimation problem without calculating the gradient of the forward model. $\textbf{The difficulty arises from the need to jointly bound two inherent sources of error:}$ (1) the discretization error introduced when approximating the Langevin diffusion, and (2) the zeroth-order (derivative-free) error in estimating the likelihood score ($\nabla \log \ell(\mathbf{y}|\mathbf{x})$). We can see the challenge in jointly bounding them in Proposition 1:
> >
> > \begin{equation}
> > \mathbb{E}[\lVert  e _ k\rVert^2] \leq (1 - p ^2) \mathbb{E}[\lVert  e _ {k-1}\rVert^2]  + \frac{4d(1-p)L _ {f _ 1}^2}{b'\mu^2} E[\lVert \mathbf{x} _ k - \mathbf{x} _ {k-1}\rVert^2] + \frac{p\mathbb{E}{[\lVert \tilde{\nabla} f _ \mu (\mathbf{x} _ k, \mathbf{u} _ i) \rVert]^2}}{b},
> > \end{equation} where $b'$ is small batch size and constant/fixed, b is large batch size. Note that achieving an $\varepsilon$-relative FI solution requires reducing the ZO smoothing parameter $\mu$ to lower the estimator bias due to ZO estimates. However, decreasing $\mu$ increases the second term in the bound, which includes the discretization error. Unlike optimization, Langevin sampling also introduces a discretization error that scales with the dimension $d$ and affects the convergence directly. Thus, reducing the bias amplifies the Langevin diffusion discretization error by a factor of $d$. This trade-off leads to a convergence rate that grows as a higher-order polynomial in $d$. In the following, we present some examples from the literature to demonstrate the difficulty of the problem.
> >
> > $\textbf{(1) Discretization error:}$ Even if one assumes access to the exact gradient of the log-distribution, which gradient-based methods require but is incompatible with the black-box setting we study, their Fisher Information (FI) convergence rate still shows a strong polynomial dependence on the dimension, scaling as $O(d^{2} L^{2} / \varepsilon^{2})$ [1]. The $d^{2}$ factor arises directly from the Brownian motion term in the Langevin SDE, which is also present in our setting.
> >
> > $\textbf{(2) Zeroth-order approximation error:}$ The zeroth-order (ZO) approximation introduces additional polynomial factors because, by construction, it estimates $d$-dimensional likelihood gradient from scalar finite-difference evaluations along random Gaussian directions. Prior work, which analyzed ZO sampling under strong log-concavity and smoothness of the log-distribution already reflects this challenge with their derivation of total oracle complexity of $O(d^2/\varepsilon^2 \cdot \log(d/\varepsilon))$ for convergence in Wasserstein-2 distance [2]. Our setting is substantially more general, as we consider non-log-concave distributions. In this setting, jointly bounding the discretization error and the ZO approximation error naturally leads to our FI convergence rate of $O(d^{7} L_{m}^{6} / \varepsilon^{4})$. Importantly, this analysis holds while allowing an arbitrary fixed per-iteration cost $pb = O(1)$ in expectation, whereas prior work [2] requires $b = d$, which is impractical due to memory constraints or computational cost.
> >
> > Importantly, although the theoretical convergence rate involves high-dimensional dependencies, our empirical results show that the proposed ZO-APMC performs substantially better in practice. In toy experiments, our method consistently converged in FI across wide range of batch size $b$ and $p$ with fixed per-iteration budget $pb=O(1)$ (please see Figure 1(b)).
> >
> > References:
> >
> > [1] Balasubramanian, K., Chewi, S., Erdogdu, M. A., Salim, A., Zhang, S. (2022, June). Towards a theory of non-log-concave sampling: first-order stationarity guarantees for Langevin Monte Carlo. In Conference on Learning Theory (pp. 2896-2923). PMLR.
> >
> > [2] Roy, A., Shen, L., Balasubramanian, K., \& Ghadimi, S. (2022). Stochastic zeroth-order discretizations of Langevin diffusions for Bayesian inference. Bernoulli, 28(3), 1810-1834.

---

### Author Response · Authors · 2025-12-03
**Message to AC**

Dear Area Chair,

Thank you for handling our submission, $\textit{``Provable Derivative-Free Inference with Score-Based Generative Priors.''}$
Reviewers gjx9, 9C6Z, and U8Hg recognized the work as principled, mathematically rigorous, and supported by extensive empirical results. The primary concerns raised relate to the assumptions, lack of intuitive explanations, the benefit of the variance-reduction mechanism, and the convergence rate derived in Theorem 1. We have addressed all of these points in our detailed rebuttal, and we respectfully direct the AC to those responses for the full technical discussion and references. All revisions made in response to the reviews are highlighted in blue in the updated manuscript for clarity.


1. $\textbf{Assumptions 1 and 5:}$ Reviewers raised concerns about the Lipschitzness and score network assumptions.
- Assumption 1: In the original submission, we noted that the log-likelihood under Gaussian noise is not globally Lipschitz, whereas     under Laplace noise it is. Reviewers gjx9 and U8Hg questioned this point and, in particular, Reviewer U8Hg asked for experiments with Laplace noise. We conduct the experiments and clarified that, for image distributions with bounded support, the log-likelihood under Gaussian noise can still satisfy a (local) Lipschitz condition, making our assumption reasonable in this setting.

- Assumption 5: We revised the boundedness condition on the score network to align with a well-known empirical observation $\lVert \mathcal{S} _ \theta (\mathbf{x}, \sigma _ k) \rVert_2 \propto 1/ \sigma _ k$ [1]. Reviewers gjx9 and U8Hg were concerned with the bounded score approximation error. We clarified that it is a reasonable assumption to enable rigorous quantification of the generalization ability of SGMs (please see $\textbf{our last response to Reviewer gjx9}$).

2. $\textbf{Motivation for derivative-free sampling.}$ Reviewers 2vHU and gjx9 questioned the need for a derivative-free posterior sampling approach, arguing that modern automatic differentiation tools can already be used to solve PDE-based inverse problems. To address this, we added concrete examples where differentiation tools may not be applicable such as commercial softwares in medical imaging ($\textbf{Weakness 1 response to Reviewer 2vHU }$). We explained that, in practice, PDE solvers with automatic differentiation can be unstable due to the physics of the forward operator $\mathbf{A}$. To address this issue, we proposed another posterior sampling algorithm $\textbf{NG-APMC}$ for noisy gradients, which uses different variance-reduction mechanism and $\textbf{does not require the log-likelihood to be Lipschitz.}$ NG-APMC has $O(L_{\pi}^2 \sigma_{\mathrm{noise}}^4 d^2 / \varepsilon^4)$ total gradient calculation with a fixed per-iteration budget $pb = O(1)$. We demonstrated the benefit of NG-APMC’s variance-reduction strategy through both numerical and statistical validations; please see Appendix A.2 for details.

3.  $\textbf{Intuitive explanations}:$ In response to Reviewers gjx9, 9c6Z, and U8Hg, we added intuitive explanations of the variance-reduction mechanism and why it is needed (please see $\textbf{Weakness 2 response to Reviewer gjx9}$). We also added intuitive explanations after each theoretical result to improve clarity.

4. $\textbf{Variance-reduction mechanism}:$ In addition to our intuitive explanation of the variance-reduction mechanism, we conducted additional numerical and statistical experiments (as requested by Reviewer U8Hg) comparing the naive ZO-APMC with $p=1$, which does not use variance reduction, to ZO-APMC with $p<1$, which incorporates our variance-reduction strategy. Please see Appendix A.4 for the results. Furthermore, in our response to Reviewer U8Hg on $\textbf{``Clarification on $p=1$ (Prop. 1)''}$, we argued that the naive estimator ($p=1$) lies in a bounded-variance regime with poor dimension scaling and established the corresponding lower bounds, whereas our PAGE estimator ($p<1$) achieves tighter bounds. This complements Proposition 1, providing aligned upper- and lower-bound perspectives on the benefit of variance reduction, $\textbf{an issue raised by all reviewers.}$


5. $\textbf{Convergence rate for Fisher information:}$ Reviewer 2vHU and U8Hg raised concerns about the high-order polynomial convergence rate in Theorem 1. We explained that this rate reflects the inherent difficulty of the problem, as it originates from the coupling between the zeroth-order approximation error and the discretization error in Langevin dynamics, an effect that is unavoidable in SDE-based posterior sampling methods. Please see our $\textbf{Weakness 2 response to U8Hg}.$


We appreciate your consideration in the final decision.

Warm Regards,

Anonymous Authors


Reference:

[1] Song, Y. and Ermon, S. Generative modeling by estimating gradients of the data distribution. In Advances in Neural Information Processing Systems, pp. 11918–11930, 2019.

---

### Meta-Review · Area_Chair_D7Af · 2026-01-12

**Summary:**

The paper proposes ZO-APMC, a derivative-free posterior sampling method for inverse problems that uses pre-trained score-based generative models (SGMs) as priors. The main claim is to provide the first formal convergence guarantees for such a "black-box" setting, where only forward-model evaluations are available.

Reviewers raised several concerns about the assumptions, theoretical analysis, practical relevance, and experiments. Some of the main concerns are listed below.

- The main contribution of the paper lies in theoretical guarantees, but the bounds are quite large and loose.
- The paper addresses a niche setting in which they only use forward operator and approximate the derivative using a zero-order method. The authors argue that such a problem is especially relevant when backpropagation through forward operator provides unstable gradient.
- The presentation of the proposed method and theoretical analysis lack intuitive explanation.

**Reviewer Concerns:**

Authors have provided pretty lengthy responses for each comment. I read as many of them as I could. While I sympathize with the authors, I feel their explanation did not offer direct and convincing responses to reviewer concerns.

- Practical motivation concern seems valid to me. While authors provided multiple examples, they all seem rather niche and a bit disconnected from the experiments in the paper.
- The large bounds on the forward model evaluation further diminishes the practical relevance of the proposed method. Authors explained that the large bound is not because of loose inequalities, but due to fundamental difficulty in the estimation. While I appreciate the difficulty, I also think this large bound makes the proposed method less appealing for any practical problem.
- Authors revised the description of the method in Sec 3.1 but the notions of accept-reject and small-large batch remain a bit confusing.
- One reviewer additionally commented that, according to authors, although theoretical convergence rate involves high-dimensional dependencies, the empirical results show that ZO-APMC performs substantially better in practice. If empirical results are much better, what is the relevance of the theoretical results which cannot be improved ?

Overall, I think this is a nice paper with theoretical analysis of a potentially useful problem but it will require a major revision.

**Reviewer Scores:**

Reviewer scores are 6,4,4,4

From the comments and one discussion I see, the scores would have likely remained the same.

---

### Decision · Program_Chairs · 2026-01-26

Reject